

# Deeply subducted continental fragments: II. Insight from petrochronology in the central Sesia Zone (Western Italian Alps)

Francesco Giuntoli[1], Pierre Lanari[1], Marco Burn[1], Barbara Eva Kunz[1] & Martin Engi[1]

[1]Institute of Geological Sciences, University of Bern, Baltzerstrasse 1+3, 3012 CH-Bern

*Correspondence to*: Francesco Giuntoli (francesco.giuntoli@plymouth.ac.uk)

**Abstract.** Continental granulite terrains subducted to mantle depths commonly show only partial and localized eclogitization. The Sesia Zone (NW Italy) is exceptional as eclogitic micaschists predominate in large parts of this terrane, and Alpine high-pressure (HP) assemblages almost completely replaced the Permian granulite protoliths. This study

documents when and at what conditions this extensive HP-equilibration took place. Results constrain the main stages of mineral growth and deformation, associated with fluid influx that occured during continental subduction. Our study comprises both main complexes of the Sesia terrane and covers some of the recently recognized tectonic subunits involved in its assembly, hence our data constrain the HP-tectonics that formed the Sesia Zone.

In the Internal Complex (IC), pulses of fluid percolated at eclogite facies conditions, between 77 and 55 Ma with the HP-

conditions reaching ~2 GPa and 600-670 °C. By contrast the External Complex (EC) records a lower pressure peak of ~0.8 GPa for 500 °C, at ~63 Ma. The juxtaposition of the two complexes occurred during exhumation, at ~0.8GPa and 350°C; the timing is constrained between 46 and 38 Ma. Mean vertical exhumation velocities are constrained between 0.9 and 5.1 mm/year for the IC, up to its juxtaposition with the EC. Exhumation to the surface occurred before 32 Ma, as constrained by the overlying Biella Volcanic Suite, at a mean vertical velocity between 1.6 and 4 mm/year. These findings constrain the

processes responsible for the exhumation and assembly of high pressure continental units.

## 1 Introduction

The behaviour of continental crust subducted to high-pressure (HP) conditions is far from being entirely understood (Kylander-Clark et al., 2008;Gerya et al., 2002;Rubatto and Hermann, 2001;Brun and Faccenna, 2008;Malusà et al., 2011;Angiboust et al., 2016). Seismic tomography beneath collisional orogens shows that large slab parts reached mantle

depths (e.g. Lippitsch et al., 2003;Replumaz et al., 2010;Zhao et al., 2015), but the source of these remnants is hard to assess because the rocks cannot be directly investigated. Conversely, where such orogens contain exhumed continental HP-rocks, these offer opportunities to investigate some of the tectono-metamorphic processes involved, notably those responsible for the return flow of continental fragments back to the surface.



Some of the crucial questions related to continental HP-domains are: how rapidly were they subducted and to what depth? When and how fast were they exhumed? Which pressure-temperature-time-deformation (P-T-t) trajectories did they experience? When and how did fluids affect these continental fragments? To shed light on these questions, the sequence of metamorphic stages recorded in suitable samples needs to be analysed in detail (Engi et al., 2017). P-T-t paths promise

insight into these details of fundamental tectonic processes.

In recent studies on the Western Alps, the combination of tectonic and numerical model studies (e.g. Yamato et al., 2008;Faccenda et al., 2009) with the reconstruction of P-T-t paths (e.g. Regis et al., 2014;Rubatto et al., 2011) has led to the recognition of two possible end-members scenarios for the subduction of continental domains to HP conditions. In the first one, the units essentially experienced tectonic slicing and accretion to overlying continental units, thus building up

complexes composed of different tectonometamorphic slices (e.g. Regis et al., 2014;Manzotti et al., 2014;Vitale Brovarone et al., 2013). Alternatively, units were eroded or ablated in the subduction channel, lost coherence and experienced substantial mixing, leading to diverse and complex P-T-t paths (e.g. Stöckhert and Gerya, 2005 Warren et al., 2008;Keppie et al., 2009;Roda et al., 2012). The distinction between these two end-members scenarios is important to understand the metamorphic evolution of deeply subducted continental domains and possibly their paleogeographic provenance.

Petrochronological studies have attempted to constrain exhumation rates of Alpine HP and UHP continental domains. Calculated rates vary from a few mm (e.g. Zucali et al., 2002 for the internal part of the Sesia Zone) to a few centimetres per year (e.g. Rubatto and Hermann, 2001 for Dora Maira UHP massif). In detail, some studies found that after a first phase of rapid exhumation up to lower crustal levels, a marked decrease in exhumation rates is observed down to a few millimetres per year (Rubatto and Hermann, 2001;Yamato et al., 2008).

A fundamental question is how much of the subduction history is recorded in the studied samples, and what triggered mineral recrystallization or equilibration. Several studies (e.g. Erambert and Austrheim, 1993;Austrheim, 1987;Rubie, 1986;Oliver, 1996;Etheridge et al., 1983;Engi et al., in preparation) suggest a strong link between deformation and fluid influx triggering mineral reactions in deeply subducted high-grade (granulite and amphibolite facies) domains. Again, it is important to know when this happened in the P-T-t evolution, especially if it occurred during subduction or only upon

exhumation (e.g. Konrad-Schmolke et al., 2011).

To respond to such first order questions we present a field-based study emphasizing P-T-t data and their implications. A key point in this approach is to establish reliable links between age data (t) and the P-T conditions of mineral formation (e.g. Schenk, 1989;Buick and Holland, 1989;Scott and St-Onge, 1995;Liati and Gebauer, 1999;Rubatto and Hermann, 2001;Foster et al., 2004;Janots et al., 2009;Gasser et al., 2012;Donaldson et al., 2013;Rubatto et al., 2011;Regis et al.,

2014;Mottram et al., 2014). We used mutual inclusions of the main mineralogical phases in the datable accessory minerals along with microstructural analyses to bracket the age data to P-T conditions.  The study area is the central Sesia Zone, located in the Western Alps (Valle d'Aosta, Italy). P-T-t data are reconstructed for the eclogite facies IC and, for the first time, for the epidote-blueschist facies EC. These data allow us to constrain the juxtaposition of the two complexes of having



occurred under greenschist facies conditions and to calculate exhumation rates for the IC as well as for the assembled Sesia Zone.

## 2 Geological setting

The Alps are an orogen developed since the Cretaceous due to the subduction of the European Plate below the Adriatic Plate
and their subsequent continental collision (e.g. Dewey et al., 1989;Rosenbaum et al., 2002;Handy et al., 2010). The Sesia Zone (SZ) is located in the Western Alps (Fig. 1a) and regarded as a rifted portion of the Adriatic Margin that experienced subduction to HP conditions (e.g. Dal Piaz, 1999). At present, it is bounded by two tectonic lineaments: the Insubric Line, to the SE, and the Gressoney Shear Zone, to the NW. The Insubric line is a major fault system of Oligocene-Neogene age that separates the SZ from the Southern Alpine, which represents the Adriatic Margin with a weak Alpine imprint at sub-
greenschist facies. An important complex of the Southern Alps is the Ivrea Zone, consisting of amphibolite to granulite facies meta-sediments, mantle peridotites and mafic intrusives; it is proposed to represent a cross section through the lower pre-Alpine crust (e.g. Bertolani, 1959, 1954;Rivalenti et al., 1975;Zingg, 1983;Quick et al., 2003). Analogies between the pre-Alpine metamorphism of the Internal Complex and the Ivrea Zone have been emphasized by several authors (e.g. Dal Piaz et al., 1972;Compagnoni et al., 1977). The Gressoney Shear Zone is a greenschist shear zone (e.g. Wheeler and Butler,
1993;Babist et al., 2006) that separates the SZ from units derived from the Piedmont-Ligurian Ocean that experienced from blueschist to eclogite facies metamorphism during the Alpine Orogeny (e.g. Bucher et al., 2005;Negro et al., 2013).

Recently, the Central Sesia Zone has been subdivided into an Internal Complex (IC) and an External Complex (EC; Giuntoli and Engi, 2016; Fig. 1b-c). The IC corresponds, as a whole, to the Eclogitic Micaschist Complex of Williams and Compagnoni (1983) and Passchier et al. (1981). In detail, the IC comprises several eclogitic sheets with a thickness of few
kilometres; they all are characterized by an alternance of bands and elongate bodies of micaschist, eclogite, ortho- and paragneisses, and leucogneiss. The main foliation of this complex is of eclogite facies. Several deformation phases locally affect this foliation at retrograde blueschist and greenschist facies conditions (details in Giuntoli and Engi, 2016).

The EC corresponds to the Gneiss Minuti and the 2DK complexes of the previous authors. It comprises km-thick tectonic sheets of leuco- to mesocratic orthogneiss with minor paragneiss, calcmicaschist, impure quartzite and marble separated by
lenses preserving Permian HT metamorphic relics (2DK; Giuntoli and Engi, 2016;Carraro et al., 1970;Dal Piaz et al., 1971). The main composite foliation reflects HP greenschist, at most epidote blueschist facies; these are the highest Alpine metamorphic conditions recorded by this complex. This foliation is affected by subsequent greenschist facies deformation. Juxtaposition of the two complexes involved a greenschist facies shear zone (Barmet Shear Zone in Giuntoli and Engi, 2016); the subunits in each complex are thin sheets that range in thickness from 0.5 to 3 km.

Based on P-T-t paths determined by petrochronological techniques, Rubatto et al. (2011) and Regis et al. (2014) recognized two tectonometamorphic subunits in the IC: A more internal unit, called Druer slice, experienced eclogite facies condition with pressure of 1.9-2 GPa and temperatures of ~550 °C at around 85 Ma, then followed by exhumation. A more external



unit, called Fondo slice, experienced a first stage of eclogite facies condition, with pressures of 1.7-1.8 GPa and temperatures between 520 and 550°C at around 75 Ma, then followed by an intermediate lower pressure stage (P<1.6 GPa and T<520 °C) at around 68 Ma, and a second high pressure stage (P=1.4-2 GPa and ~550 °C) at around 65 Ma, followed by a retrograde decompression path. For the EC in the central Sesia Zone, no P-T-t data are available so far.

A number of studies have produced additional age data for the Sesia Zone, using various methods, notably for the U-Th-Pb, Rb-Sr, and Ar-Ar systems; results are summarized in Table 1 of Compagnoni et al. (2014). These age data span from 85 to 62 Ma for the eclogite facies metamorphism in the IC, but apart from those detailed above, none of the datasets have been linked in detail to petrogenetic conditions. In the EC few data are available, ranging from 46 to 38 Ma, generally linked to the greenschist facies imprint (Compagnoni et al., 2014 and references therein).

## 3 Sampling strategy and petrochronological approach

To document the polyphase history of the Sesia Zone, we reconstructed detailed P-T-t paths for five samples taken in the IC and one sample in the EC. Of over a hundred samples checked, a very small percentage fulfilled the requirements for such a study, and in the EC, in particular, suitable material to quantify P-T-t conditions by the present method turned out to be very rare. The samples analysed nevertheless provide constraints to derive a P-T-t path also for the EC, allowing us to determine

when and at what conditions the two complexes were juxtaposed.

The samples were taken in key areas of the mapped structures and were collected oriented, in order to keep the link between the meso- and microstructural evidence. Samples were carefully studied by optical and (where needed) scanning electron microscopy (SEM) to reconstruct their microstructural and metamorphic evolution. Once a relative chronology was established, selected growth zones of mineral phases were analysed by electron probe micro-analyser (EPMA) as a basis to

perform thermodynamic modelling. P-T data were linked with fabric elements, using textural criteria. *In situ* age dating of specific growth zones of accessory phases was then performed by LA-ICP-MS to link the age (t) to a metamorphic stage responsible for mineral growth. Observations and P-T-t data derived from each sample were then compared and correlated within the sample series and then related to observations made in the field data and in microscopy.

This process, called petrochronology (e.g. Engi et al., 2017), is particularly effective if the study area is mapped and

structurally characterized in detail, as in the present case.

## 4 Petrography and mineral chemistry

### 4.1 Methods

#### 4.1.1 SEM

Back-scattered electron images (BSE) were acquired using the Zeiss EVO50 SEM at the Institute of Geological Sciences

(University of Bern) using an accelerating voltage of 15 to 25 KeV, a beam current of 500 pA and a working distance of 10





mm. Cathodoluminescence (CL) pictures where obtained with the same operative conditions, but with 10 KeV accelerating voltage and a working distance of 9.5 mm.

### 4.1.2 EPMA analyses

EPMA analyses were performed using a JEOL JXA-8200 superprobe at the Institute of Geological Sciences (University of

Bern). Point mode analyses and X-ray compositional maps were acquired both using wavelength dispersive spectrometers (WDS). For X-ray mapping the procedure described in Lanari et al. (2013) was followed. It consists in measuring point mode analyses first and then acquiring X-ray compositional maps on the same area. For point analyses, analytical conditions were 15 KeV accelerating voltage, 10 to 20 nA specimen current, 40 s dwell times (including 2×10 s of background measurement) and a beam ø from 1 to 5 μm. Lower current and higher beam size were used for minerals containing Ca, Na

and K such as phengite and plagioclase. Nine oxide compositions were measured, using synthetic and natural standards: wollastonite / orthoclase / almandine ($SiO_2$), anorthite / almandine ($Al_2O_3$), anorthite (CaO), almandine (FeO), forsterite / spinel (MgO), orthoclase / phlogopite ($K_2O$), albite ($Na_2O$), ilmenite ($TiO_2$), and tephroite (MnO). For X-ray maps, analytical conditions were 15 KeV accelerating voltage, 100 nA specimen current and dwell times of 150-250 ms. Nine elements (Si, Ti, Al, Fe, Mn, Mg, Na, Ca and K) were measured at the specific wavelength in two passes. Intensity X-ray

maps were standardized to concentration maps of oxide weight percentage using spot analyses as internal standard. X-ray maps were processed using XMapTools 2.2.1 (Lanari et al., 2014).

### 4.2 Sample description

Five samples were analysed from the IC and two samples from the EC. These were collected from internal (SE) to external areas (NW) of the Sesia Zone and are described in detail, also their GPS location, in the Supplement S1. A brief account is

given here, with characteristic images shown in Fig. 2.

The studied samples of the IC (FG1324, FG1315, FG12157, FG1347, and FG1249) are micaschists with eclogite facies assemblages comprising quartz, phengite, garnet, ± paragonite ± glaucophane ± omphacite ± chloritoid, with accessory allanite ± rutile. The main fabric visible in all these samples is an eclogite facies foliation (Fig. 2a). Evidence of several deformation stages occurring before or after the main eclogite facies foliation is recorded in several samples as microlithons,

commonly of phengite, omphacite, glaucophane or chloritoid oriented at high angle relative to the main foliation, which wraps around them or is overgrown by them (Fig. 2b). A further evidence of several metamorphic stages occurring at eclogite facies conditions is reflected in the growth zones of garnet (Giuntoli et al., submitted). Pre-Alpine relics include garnet cores (Fig. 2c) and zircon (cores ± first rims).

Retrograde stages of blueschist facies or greenschist facies assemblages related to decompression are locally recorded in the

samples. The blueschist facies stage produced pleochroic crossite rims around glaucophane (Fig. 2d). The greenschist facies stage produced symplectites of actinolite ± albite ± chlorite around glaucophane and omphacite, chlorite grew at the expense of garnet, and titanite rims formed around rutile (Fig. 2e-f).



Sample FG1420, collected in the EC, is a garnet orthogneiss that shows a HP greenschist foliation marked by phengite, chlorite, and titanite; the foliation wraps garnet porphyroblasts that preserve a relic internal foliation (Fig. 2g). Permian magmatic relics of pleochroic allanite are surrounded by an Alpine corona of epidote grains (Fig. 2h). Some hundred meters to the north, another sample (FG12107) of the EC was collected. This is a leucogneiss characterized by the same metamorphic history and paragenesis as the previous sample, except that garnet and magmatic allanite are missing.

## 4.3 Growth zones of garnet and phengite

Garnet and phengite display features in the IC samples that differ from those in the EC samples. To highlight and describe these, a comparison between sample FG1249 (IC) and FG1420 (EC) is proposed in the following paragraphs. More documentation of garnet textures and mineral inclusions in the IC is available in Giuntoli et al. (submitted).

In the IC samples, garnet consists of a core followed by several rims with a grain size up to several millimeters (Fig. 3a, b). The compositional map of the grossular end-member fraction ($X_{Grs}$) in sample FG1249 shows a porphyroclastic core ($Alm_{72}Prp_{18}Grs_5Sps_5$; Table 1) with internal fractures sealed by garnet of higher $X_{Grs}$ (Fig. 3b). A first rim (rim1-$Alm_{76}Prp_{15}Grs_9$) overgrows the core and displays higher grossular contents. This rim1 is followed by rim2, which again records an increase of $X_{Grs}$ ($Alm_{62}Prp_{20}Grs_{18}$), and both internally and externally resorbs parts of rim1 and core. Rim3 is peripheral and shows the highest Ca contents ($Alm_{58}Prp_{19}Grs_{23}$).

Sample FG1315 is characterized by a porphyroclastic core ($Alm_{69}Prp_{28}Grs_4$) with lobate edges and resorption features (details in Giuntoli et al. submitted and Engi et al. in prep.) surrounded by several rim generations: rim1 ($Alm_{61}Prp_{21}Grs_{19}$), rim2 ($Alm_{65}Prp_{24}Grs_{11}$) and rim3 ($Alm_{70}Prp_{24}Grs_6$). Atoll garnets, a few hundred microns in size, are observed in this sample. The shells of the atoll garnet have similar zoning patterns and compositions as the rim generations just described. In sample FG12157 the garnet core ($Alm_{70}Prp_{26}Grs_4$) is rimmed by two growth zones: rim1 ($Alm_{64}Prp_{20}Grs_{16}$) and rim2 ($Alm_{59}Prp_{24}Grs_{17}$). In sample FG1347, the garnet core ($Alm_{69}Prp_{28}Grs_3$) is enclosed by three rims (rim1-$Alm_{66}Prp_{23}Grs_{11}$, rim2- $Alm_{68}Prp_{26}Grs_6$, rim3-$Alm_{70}Prp_{26}Grs_4$). The exception is sample FG1324, in which garnet shows a single growth zone of homogeneous composition ($Alm_{70}Prp_{18}Grs_{11}Sps_1$).

In the EC, sample FG1420 shows garnet with completely different features. As shown in Fig. 3d, the $X_{Sps}$ map highlights concentric zoning (values of $Alm_{54}Prp_2Grs_{36}Sps_8$ for the core, $Alm_{57}Prp_3Grs_{37}Sps_4$ for rim1, $Alm_{60}Prp_3Grs_{35}Sps_2$ for rim2 and $Alm_{63}Prp_4Grs_{32}Sps_1$ for rim3), with no visible resorption features.

To link the growth zones of garnet to the main assemblage observed in the mineral matrix, microstructural relations, overprinting criteria and mutual inclusions were employed, based on optical microscopy, SEM, and compositional maps. In particular, garnet in sample FG1249 displays inclusions of paragonite, phengite, and quartz between rim1 and rim2 (Fig. 3a). Rutile inclusions of few microns are present in rim2 and 3. Late chlorite fractures dissect the entire garnet. Garnet in sample FG1420 is wrapped by the main external foliation and includes an internal foliation marked by quartz, epidote, and titanite (Fig. 3c).



Phengite crystals in the IC display a uniform composition except along grain boundaries, where lower Si and $X_{Mg}$ contents are found, indicating retrograde overprinting (e.g. Fig. 3e; Group1 Si ~3.36 apfu, $X_{Mg}$ ~0.83; Group2 Si ~3.3 apfu, $X_{Mg}$ ~0.68 in sample FG1249; Table 2).

In the EC, phengite shows two distinct generations based on their microtextural position: the first one describes the main foliation and is characterized by high silicon values (Fig. 3f; Group1 Si ~3.4 apfu, $X_{Mg}$ ~0.61 in sample FG1420). The second phengite generation (Group2 Si ~3.32 apfu, $X_{Mg}$ ~0.61) rims the first one and occurs in fold hinges that deform the main foliation.

## 4.4 Allanite textures and their microstructural relations

In the samples of the IC, allanite prisms are elongate in the eclogite foliation, showing mutual intergrowth relations with other minerals defining this main foliation. Based on these observations (and details given below for each sample), we infer coeval growth of allanite and the minerals marking the eclogite facies foliation in all of the samples analysed from the IC.

In samples FG1324, FG1315, and FG1347 allanite crystals are characterized by one main growth zone. Some thin (< 20-30 µm) allanite rims are observed as well as epidote or clinozoisite rims that appear dark in BSE photos (Fig. 4); where present, these mark a retrograde greenschist overprint. In sample FG12157 and FG1249, BSE pictures show one or more allanite rims characterized by lower brightness (Fig. 4c-e). These rims may reflect minor retrograde stages that weakly altered the eclogite facies parageneses as well. Again, a peripheral epidote rim is present. Monazite is present as a relic in some allanite cores in samples FG1324, FG1315, FG1347, and FG1249. Monazite shows lobate edges and is surrounded by symplectites (µm in size) of allanite and apatite or by discrete crystals of apatite and allanite (Fig. 4f). These features suggest prograde growth of allanite and apatite at the expense of monazite, a common allanite-forming reaction (e.g.Janots et al., 2008).

Various mineral inclusions are found in allanite grains, as summarized in Table 5. In detail, allanite in sample FG1324 shows intergrowths with garnet (Fig. 4a) suggesting synchronous growth of the two minerals. Phengite inclusions analysed in allanite show the same chemical composition as those marking the main foliation (representative mineral analyses are available in the Supplement S3). Based on these features and the alignment in the foliation, allanite is interpreted to have grown syn-kinematically in the foliation and at the same time as garnet. In sample FG1315 allanite includes phengite, paragonite, and garnet (Fig. 4b); the latter is similar in composition to Alpine HP-rims and atoll garnet. Phengite inclusions have the same composition as phengite marking the main foliation and as phengite included in atoll garnet. Due to the relationships of these mutual inclusions in this sample, allanite growth again appears to be related to the development of the main foliation. In samples FG1347 and FG1249 allanite includes phengite and paragonite, in FG12157 only phengite; these micas have the same composition as those defining the eclogite foliation (Fig. 4c, d, f). In the case of FG1249 and FG1347, allanite also shows intergrowths with both white micas (Fig. 4e).

In the EC, allanite is rare and, where present, is typically a magmatic allanite that appears dark brown and pleochroic in the optical microscope, with a grain size of some millimetres (Fig. 2h; Giuntoli and Engi, 2016). In only two samples



metamorphic allanite was found, preserved in the core of epidote crystals (FG1420, FG12107). The metamorphic allanite has a grain size less than 50 μm, is colourless or pale yellow in polarized light, with low interference colour and undulose extinction in crossed polarized light. Sample FG1420 shows both magmatic and metamorphic allanites (Fig. 4g, h). Allanite includes paragonite, with phengite and titanite occurring both in the epidote rim and at the allanite-epidote boundary (Fig. 4g). Very few tiny monazite grains (few μm) are found in the core of metamorphic allanite. Sample FG12107 also shows similar epidote crystals preserving metamorphic allanite in their core, as in sample FG1420. The magmatic allanite preserved in sample FG1420 occurs as mm-size grains that are fractured and appear much brighter in BSE pictures than metamorphic allanite (Figs. 2h, 4h). Epidote crystals form satellites around magmatic allanite, suggesting partial breakdown (Fig. 4h). Note that these epidote crystals have a BSE-bright core of newly grown (Alpine) metamorphic allanite.

## 4.5 Zircon textures

The internal textures of zircon from the IC reveal complex zoning in CL-images (Fig. 5) with detrital cores and several phases of resorption and (metamorphic) overgrowth. Zircon cores commonly preserve a variety of internal textures, most commonly oscillatory zoning (Fig. 5e) which is typical of zircon grown form melt. Many cores are affected by resorption, obliterating earlier features, but in some grains show sharp boundaries between core and rims, and these preserve features of sediment transport such as broken, rounded or pitted surfaces. The number of metamorphic rim varies between and within samples, from one rim in sample FG1257 to a maximum of five different metamorphic rims in FG1315. Most zircon grains show a first metamorphic rim with a light grey to bright CL-response, followed by a rim with dark CL-appearance. The third rim typically has again medium grey to bright white CL-response. In sample FG1315, a forth (dark CL) and fifth (light grey) rim follow, while sample FG1249 has a forth rim with a bright CL-appearance and FG1347 occasionally shows a forth rim which is dark in CL-images. The internal textures of the different rims are not always clear to distinguish, either because of the limited width or, in case of very dark or bright CL-response, due to limited contrast. The metamorphic rim in FG12157 is either uniform or shows fir-tree zoning, in sample FG1347 the metamorphic rims are often too thin to properly distinguish textures or, in case of the bright-CL rim, no texture is recognizable. The first two rims in sample FG1315 have cloudy textures, the third rim with the bright-CL shows no further internal textures but sometimes has inclusions and the two outermost rims commonly are uniform or cloudy in texture. The innermost and outermost metamorphic rim in sample FG1249 and FG1324 are bright and featureless, the second dark rim in sample FG1249 shows sector zoning as well as many inclusions. The third (medium grey) rim in FG1249 has a cloudy texture.




# 5 Thermobarometry

## 5.1 Methods

### 5.1.1 Whole rock major element compositions

Major element compositions were analysed by X-ray fluorescence (XRF) spectrometry at the University of Lausanne (Switzerland). Representative quantities of samples were crushed and then pulverized in a tungsten carbide mill. The powder was dried for two hours at 105°C. Loss of ignition was then determined by weight difference after heating to 1050°C for 3 hours.

### 5.1.2 Garnet thermobarometry using GrtMod

To model the complex garnet textures present in these samples, the program GRTMOD (Lanari et al., 2017) was used. In essence, the code refines the local bulk composition used in free energy minimization. This offers an inverse modeling approach to obtain P-T conditions from garnet growth zones from samples that may have experienced garnet fractionation and possibly also resorption of previously formed garnet. For each inversion, corresponding to a growth stage, a solution was deemed acceptable if the residual value ($C_0$) was <0.05, reflecting a sufficiently close match between the modeled and observed garnet compositions. In the IC samples, resorption and fractionation were constrained according to the volumetric proportion of each growth zone, as estimated from the thin section and the compositional maps. In sample FG1420 no resorption was allowed, as garnet textures show no evidence for it in the compositional maps. To model the rims generation in the IC samples the "go fast mode" function was used (Lanari et al., 2017) with initial starting PT-guess of 650 °C and 1.6 GPa. The MnO component was used in the thermodynamic computations of the relatively Mn-garnet in sample FG1420. In sample FG1249, MnO was used to model the garnet core but was ignored in the models of the following rims because the concentration found in garnet is low (< 1 wt% MnO). In the remaining samples, MnO was ignored (< 1 wt%), and the system considered in modelling was simplified to $SiO_2$-$TiO_2$-$Al_2O_3$-FeO-MgO-CaO-$Na_2O$-$K_2O$-$H_2O$. The thermodynamic database used was the same as to compute the isochemical phase diagrams.

### 5.1.3 Isochemical phase diagrams (pseudosections)

To compute isochemical equilibrium phase diagrams, we used the Gibbs free energy minimization algorithm THERIAK/DOMINO (de Capitani and Petrakakis, 2010;De Capitani and Brown, 1987). The thermodynamic database of Berman (1988) with subsequent updates collected in JUN92.bs (distributed with Theriak-Domino 03.01.2012; Supplement S4) was used together with the following solution models: Berman (1990) for garnet; Fuhrman and Lindsley (1988) for feldspar; Meyre et al. (1997) for omphacite; Keller et al. (2005) for white mica, and ideal mixing models for amphibole (Mäder and Berman, 1992;Mäder et al., 1994), epidote, and chlorite (Hunziker, 2003). All Gibbs free energy minimizations were carried out assuming an excess in pure $H_2O$ fluid. The amount of $H_2O$ component predicted at high-pressure is in line with the measured LOI (1.4-2.7 wt-%) in the present-day samples. Note that for the pre-Alpine HT computations no melt



model was used. $Fe^{3+}$ was ignored because of the lack of analytical data and suitable ferric end-members in solid solution models.

### 5.1.4 Chlorite and white mica multi-equilibrium

To constrain the P-T conditions of retrograde stages, multi-equilibrium computations of the high-variance assemblages

involving chlorite and white mica were carried out using the standard state properties and solid solution models of Vidal et al. (2005; 2006) for chlorite, Dubacq et al. (2010) for phengite and the program CHLMICAEQUI (Lanari, 2012). The activity of $H_2O$ was set to unity. Three methods were successively employed:

(1) *Chlorite+Quartz+$H_2O$ thermometry*: The chlorite formation temperature and $XFe^{3+}$ were estimated at a fixed pressure of 0.1 GPa from the convergence of four equilibria involving five chlorite end-members, quartz and $H_2O$

(Lanari et al., 2012; Vidal et al., 2016).

(2) *White-Mica+Quartz+$H_2O$ barometry*: A divariant P-T equilibrium line was estimated for each white mica analysis (assuming $XFe^{3+} = 0$) from the convergence of three equilibria involving five phengite end-members, quartz and $H_2O$ (Dubacq et al., 2010).

(3) *Chlorite+White-Mica+Quartz+$H_2O$ thermobarometry*: The formation P and T of each chlorite and white mica

couple as well as their respective $XFe^{3+}$ were estimated by minimizing the square root of the sum of $(\Delta G_{reaction})^2$ for 6 equilibria (Supplement S5).

Note that for clarity, only 64 equilibria (excluding the Pyrophyllite·$1H_2O$ end-member) are shown in the P-T diagrams. The starting guess for T and P was taken from the result of Chlorite+Quartz+$H_2O$ thermometry and White-Mica+Quartz+$H_2O$ barometry. This multi-equilibrium approach relies on the assumption of local thermodynamic

equilibrium between the selected chlorite and white mica compositions at given P-T conditions of convergence. Chlorite and white mica couples were chosen if a sharp contact was observed between them (textural equilibrium) or where microtextural evidence suggested equilibrium.

### 5.2 Bulk rock and reactive bulk composition

For samples FG1324 and FG1420 the original bulk rock compositions obtained by XRF were used to compute the

isochemical phase diagrams. In the samples FG1315, FG12157, FG1347 and FG1249 however, the unmodified bulk rock composition cannot be used for modeling because a significantly high volume fraction of garnet is present (5 − 10 vol%), including a pre-Alpine core and Alpine rims generations. Garnet is known to fractionate the reactive bulk rock composition, and this process affects the validity of the P-T estimates (Lanari and Engi, 2017). To compute these diagrams properly, the reactive bulk rock composition was approximated using the program GRTMOD, which substracts the previously crystallized

garnet composition from the original XRF bulk rock composition (details in Lanari et al., 2017). Thus each isochemical phase diagram is valid for a single P-T stage only. To select the reactive bulk composition of this specific stage, a link must be established between the particular garnet generation that formed in equilibrium with the mineral phases present in the

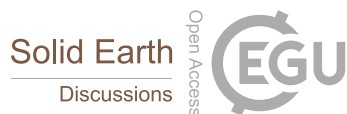

matrix. We established this link using petrographic observations, including textural equilibrium criteria, compositional zoning (visible in X-ray maps) and inclusions relationships. In detail, we thus determined that the garnet growth zones that coexisted with the mineral matrix are garnet rim3 for sample FG1315, rim1 for sample FG12157, rim3 for sample FG1347, and rim2 for sample FG1249. The corresponding reactive bulk compositions used for the modeling are contained in the

Supplement S3.

### 5.3 Garnet thermobarometry and phase diagrams

In Fig. 6 isochemical P-T isochemical phase diagrams are shown for each sample. On top of these diagrams are plotted the results of garnet thermobarometry (GRTMOD), the garnet isopleths ($X_{Grs}$, $X_{Alm}$, and $X_{Prp}$ in sample FG1324), and the $X_{Mg}$ and Si (in apfu) isopleths for phengite. Furthermore, for sample FG1420, the results of chlorite and white mica thermobarometry

are displayed. A summary of the mineral compositional data for the main phases, the modelling method used, the XRF analyses of major elements of each sample, and details of the GrtMod results are available as Supplement S3, S6, S7 and S8 respectively. Each sample from the IC and EC is presented separately below.

### 5.3.1 IC - FG1324 Omphacite, garnet, glaucophane and rutile micaschist

Garnet isopleths (Alm, Grs, Prp) intersect in two areas of the PT-diagram: at 1.65-1.75 GPa and 600-650 °C and at 1.9-2

GPa and ~550 °C. Assuming equilibrium between garnet and phengite, these conditions can be further constrained by matching this result with the Si and $X_{Mg}$ isopleths of phengite (measured values: 3.38-3.46 and 0.76-0.81 respectively). Phengite isopleths match the garnet composition at 1.65-1.75 GPa and 600-650 °C (Fig. 6a). Minor discrepancies between observations and modelled assemblages are noted at these conditions: paragonite instead of glaucophane is predicted to be stable in the model, probably because no solid solution model for sodic amphiboles is available in the thermodynamic

database used in this study. Furthermore, 6 vol% of biotite is also predicted to be stable, whereas it is not observed in thin section.

### 5.3.2 IC - FG1315 Garnet, epidote and rutile quartz-micaschist

The garnet core is found to be stable at 0.82 GPa and 750 °C, with a crystallization of ~14 vol% (details available in Supplement S3 and S8). The first garnet rim is predicted stable at ~1.5 GPa and 650 °C, with a resorption of ~3.7 vol%

garnet core and crystallization of ~5.5 vol% rim1. The second and third garnet rims are found to be stable at similar conditions: 1.9 GPa, 650°C and 1.8 GPa (resorption of 2.3, 2.5 vol% core and rim1, crystallization of 5.6 vol% rim2), and at 670°C (resorption of 5, 0.4, and 3.1 vol% core, rim1, and rim2, plus crystallization of 8.6 vol% rim3). PT-conditions of rim2 and 3 match with the intersection of Si (3.37-3.40) and $X_{Mg}$ (0.80-0.82; Fig. 6b) isopleths of phengite. The predicted assemblage agrees with the minerals observed in thin section (Supplement S1, S3).



### 5.3.3 IC - FG12157 Garnet, glaucophane, epidote and rutile micaschist

In this model, ~7 vol% of garnet is modelled stable at 0.6 GPa and 900 °C; its composition corresponds to the garnet core. The first garnet rim is predicted stable at 1.6 GPa - 650 °C (resorption of ~1.4 vol% garnet core and crystallization of ~10.8 vol% rim1), in agreement with the intersection of phengite isopleths for the observed values of Si (3.34-3.38) and $X_{Mg}$ (0.76-0.82; Fig. 6c). Crossite rims around glaucophane in this sample mark a decompression stage ; this may correlate with the second rim in garnet, which is found to be stable at 1.4 GPa and 650 °C (resorption of 1.5 vol% core, 7.3 vol% rim1, and crystallization of 10.6 vol% rim2). Biotite is predicted stable (4-6%) but none is observed in thin section.

### 5.3.4 IC - FG1347 Chloritoid, garnet and rutile micaschist

The garnet core is modelled stable at 0.8 GPa and 780 °C, with a crystallization of ~17.7 vol%. The three garnet rims show similar PT-conditions: 1.9 GPa, 590 °C (resorption of ~1.6 vol% garnet core and crystallization of ~2.8 vol% rim1); 1.8 GPa, 600 °C (resorption of 3.2 vol% core, 1.4 vol% rim1, and crystallization of 5 vol% rim2); then at 2.0 GPa , 600°C (crystallization of 2 vol% rim3). PT-estimates for the garnet rims are in perfect agreement with the intersection of phengite isopleths at Si apfu (3.29-3.33) and $X_{Mg}$ (0.78-0.82; Fig. 6d). The predicted assemblage matches with the minerals observed, except for kyanite (3 vol% predicted) that was not detected in thin section.

### 5.3.5 IC - FG1249 Garnet, epidote and rutile micaschist

The garnet core is modelled stable at 0.6 GPa, 730 °C, with crystallization of ~10 vol% garnet. rim1 is modelled stable at 0.6 GPa, 620 °C (resorption of ~2 vol% garnet core and crystallization of ~2 vol% rim1). The second rim is related to the peak pressure recorded by this sample (1.63 GPa, 615 °C with resorption of 0.2 vol% core and 0.05 vol% rim1, plus crystallization of 10 vol% rim2), conditions are in agreement with the Si apfu (3.35-3.4) and $X_{Mg}$ (0.77-0.83; Fig. 6e) compositions of phengite. The last rim records may mark the thermal peak at lower pressure (1.56 GPa, 660 °C with garnet resorption of 6.9 vol% rim2 and crystallization of 8.6 vol% rim3).

### 5.3.6 EC - FG1420 Garnet orthogneiss

The garnet core is found stable at 0.48 GPa and 490 °C, rim1 at 0.67 GPa and 500 °C, rim2 at 0.73 GPa and 510 °C and rim3 at 0.8 GPa and 520 °C, with a total amout of 4.8 vol% garnet. Note that no resorption of garnet was taken into consideration in this model.

Two phengite generations are presents (chapter *Growth zones of garnet and phengite*; Fig. 3f): the phengite describing the main foliation, with higher silica content, displays Si apfu and $X_{Mg}$ isoplets itersection at ~1.4 GPa, 550 °C. These conditions are not substantiated by the mineral assemblage predicted to contain omphacite and rutile, which are not observed in thin section. Also, at these PT-conditions further garnet should have crystallized, with a modal increase from 6.5 to >7.5 vol%, but is no garnet of a chemistry compatible with these P-T conditions is observed. We suspect that phengite grew at lower PT-



conditions, as the appropriate Si apfu values intersect the P-T results derived from White mica + quartz + H$_2$O barometry at 0.6-0.8 GPa, 350-400 °C (Fig. 6f; more details in the following chapter).

The second generation of phengite, post-dating the main foliation, displays an intersection of Si apfu values with the results derived from White mica + quartz + H$_2$O barometry at 0.55-0.75 GPa, 300-350 °C.

**5.4 Multi-equilibrium thermobarometry**

**5.4.1 IC - FG1315 Garnet, epidote and rutile quartz-micaschist**

Chlorite in this sample are retrograde and recorded formation temperatures decreasing from 450 °C to 300 °C (Fig. 7a). White mica + quartz + H$_2$O barometry suggest pressures comprised between 1.5 and 0.4 GPa for the temperature range of chlorite. Chlorite and white mica grains in textural equilibrium are used to constrain the equilibrium conditions at 0.8 ± 0.2

GPa and 340 ± 50 °C for the retrograde stage (Fig. 7b; Tables 3 and 4).

**5.4.2 IC - FG12157 Garnet, glaucophane, epidote and rutile micaschist**

Chlorite records formation temperatures decreasing from 430 °C to 310 °C (Fig. 7c). White mica + quartz + H$_2$O barometry suggests pressures comprised between 0.02 and 0.1 GPa for the temperature range shown by chlorite. Chlorite and white mica grains in textural equilibrium are used to constrain the equilibrium conditions at 0.54 ± 0.2 GPa and 394 ± 50 °C for the

retrograde stage (Fig. 7d).

**5.4.3 IC - FG1347 Chlorithoid, garnet and rutile micaschist**

Chlorite records formation temperatures decreasing from 370 °C to 250 °C (Fig. 7e). White mica + quartz + H$_2$O barometry suggests pressures comprised between 0.02 and 0.1 GPa for the temperature range of chlorite. Chlorite and white mica grains in textural equilibrium are used to constrain the equilibrium conditions at 0.78 ± 0.2 GPa and 341 ± 50 °C for the retrograde

stage (Fig. 7f).

**5.4.4 EC - FG1420 Garnet orthogneiss**

The chlorite and white mica multi-equilibrium technique was used to constrain the equilibrium conditions of three successive stages of retrogression using couples linked to different microstructural positions developed after the main foliation (fold hinges, pressure shadows, static overgrowth over the main foliation). The first couple shows equilibrium conditions at 0.87 ±

2 GPa, 354 ± 50 °C; the second at 0.60 ± 2 GPa, 331 ± 50 °C; the third at 0.42 ± 2 GPa, 231 ± 50 °C (Fig. 8).



# 6 Geochronology

## 6.1 Methods

### 6.1.1 Allanite geochronology

Single grains of allanite were separated using high voltage pulsed power discharges (Selfrag device at University of Bern;
(e.g. Rudashevsky et al., 1995) followed by magnetic separation and heavy liquids, hand-picked, mounted in acryl/epoxy and polished to equatorial section. The grains were imaged using SEM (BSE), to document the internal texture and compositional zoning. Allanite dating was performed using a LA-ICP-MS Geolas Pro 193 nm ArF excimer laser coupled to an Elan DRC-e ICPMS (Institute of Geological Sciences, University of Bern). The analytical procedure followed is described in detail in Burn et al. (2017). In particular, pre-ablation was performed for 4-5 pulses using an energy density of
2.5 J/cm$^2$, a repetition rate of 1 Hz and spot sizes of 40 and 32 μm. Ablation was performed using an energy density of 2.5 J/cm$^2$, a repetition rate of 9 Hz and spot sizes of 32 and 24 μm. He (1 l/min) and H2 (0.08 l/min) were used as aerosol transport. Instrument setting was optimised for heavy masses and oxide production (ThO$^+$/Th$^+$) was decreased to be lower than 0.5 %. NIST SRM 610 measurements were performed for quantification of U- and Th-concentrations. Plešovice (Sláma et al., 2008) zircon was used as primary standard. Cima d'Asta Pluton (CAP) allanite was used as secondary standard. The
acquisition series were approximately 1h to minimise instrumental drift including between 8 and 12 unknowns analyses bracketed by 8 analyses of the primary standard Plešovice used for U-Th-Pb ratio calibration; SRM610 for trace element calibration, as well as 3 analyses of the secondary reference material CAP[b]. Data reduction was performed with the in-house program TRINITY (Burn et al., 2017).

### 6.1.2 Zircon geochronology

Individual grains of zircon were separated as described for allanite and investigated by SEM, using cathode luminescence (CL) imaging to document their internal textures. Zircon dates were obtained using the same LA-ICP-MS instrument, following the measurement procedures described by Kunz et al. (2017). Ablation was conducted with an energy density of 2.5 J/cm$^2$, a repetition rate 9 Hz and spot sizes of 32 or 16 μm. Samples were bracketed by zircon standard GJ-1 (Jackson et al. 2004) and NIST SRM 612 measurements for quantification of U, Th and other trace element concentrations, using $^{29}$Si as
an internal standard. For accuracy and long-term reproducibility, Plešovice was measured as secondary zircon standard giving a $^{206}$Pb/$^{238}$U weighted mean age of 339.2 ± 1.6 Ma (n=34; error 2σ). Acquisition series took approximately one hour to minimise instrumental drift and were composed of the following: 2 SRM glasses, 3 GJ-1 zircon, 6-8 zircon unknowns, 3-4 Plešovice zircon, 6-8 zircon unknowns, 3 GJ-1 zircon, 2 SRM glasses. Data reduction was performed with Iolite 2.5 (Paton et al., 2011;Paton et al., 2010) with DRS 'Visual age' (Petrus and Kamber, 2012). All dates reported in this study are
concordant within error, no common Pb correction was applied.





## 6.2 Allanite U-Th-Pb dating

For both IC and EC, we were able to date allanite cores only, as the rims turned out to have to high common lead ($Pb_c$) fractions $f_{206}$ and $f_{208}$, hence the common lead correction (Gregory et al., 2007;Burn et al., 2017) would lead to large uncertainties in the age calculation.

In the IC and EC, the Tera-Wasserburg and $^{232}Th/^{206}Pb_c$ -$^{208}Pb/^{206}Pb_c$ isochron diagrams are concordant, within the range of error, in all the analyzed samples (Figs. 9, 10). These range between 77 and 56 for the allanite cores of the IC. In the EC, the magmatic allanite cores in sample FG1420 yielded ages of ~290 Ma, the metamorphic cores of 73.7±8.2 Ma. In sample FG12107, the metamorphic allanite cores were dated to 62.8±3.3 Ma. A summary of the Alpine allanite age data of each sample is available in Table 5.

In detail, in the IC Tera-Wasserburg diagrams show $^{207}Pb/^{206}Pb$ y-intercepts of 0.84 ± 0.01, 0.823 ± 0.018 and 0.83 ± 0.004 for samples FG1324, FG1315 and FG12157 respectively, with an MSWD on the regression comprised between 1.2 and 2.1 (Fig. 9). The $^{206}Pb_c$-isochron diagrams display a $^{208}Pb/^{206}Pb$ y-intercept of 2.081 ± 0.027, 2.077 ± 0.066 and 2.085 ± 0.028 for samples FG1324, FG1315 and FG12157 respectively, with an MSWD on the regression comprised between 0.46 and 0.98. These values are close to the predicted values of predicted values of model lead evolution of Stacey and Kramers (1975) for this time range (Fig.1 in Burn, 2016). The exception is sample FG1347, in which the Tera-Wasserburg diagram shows a $^{207}Pb/^{206}Pb$ y-intercept of 0.787 ± 0.04 (MSWD on the regression of 2.5) and displays a $^{208}Pb/^{206}Pb$ y-intercept on the $^{206}Pb_c$-isochron diagram of 1.98 ± 0.082 (MSWD on the regression of 0.4). These values differs from the predicted values of Stacey and Kramers (1975).

In the EC, for the magmatic allanite of sample FG1420, Tera-Wasserburg diagram shows $^{207}Pb/^{206}Pb$ y-intercept of 0.85 ± 0.28 and displays a $^{208}Pb/^{206}Pb$ y-intercept on the $^{206}Pb_c$-isochron diagram of 2.09 ± 12 (Fig. 10). These large uncertainties are related to the few (4) spots analyses. Tera-Wasserburg diagrams of metamorphic allanites show $^{207}Pb/^{206}Pb$ y-intercepts of 0.825 ± 0.01 and 0.811 ± 0.014 (MSWD on the regression of 0.7 and 1.2) and displays $^{208}Pb/^{206}Pb$ y-intercepts on the $^{206}Pb_c$-isochron diagrams of 2.061 ± 0.029 and 2.03 ± 0.031 (MSWD on the regression of 0.4) for sample FG1420 and FG12107, respectively. These values are close to the predicted values of Stacey and Kramers (1975) for these time ranges (Fig.1 in Burn, 2016).

## 6.3 Zircon U-Pb dating

The range of Alpine U-Pb zircons dates for each sample is summarized in Table 5. Details on the pre-Alpine dates are available in Kunz et al. (2017). The Supplement S9 gives an overview of range of pre-Alpine dates as well as individual analyses of Alpine dates. Detrital zircon cores in samples from the IC range from ~793 to 353 Ma and partially overlap with the first rim in sample FG1324 and FG1249 (Fig. 5); dates range between ~450–420 Ma. In most samples, at least two rims are found, yielding Carboniferous to Triassic dates from ~313 to 222 Ma, as discussed in detail by Kunz et al. (2017). FG1324 is the only sample that yielded no Carboniferous to Triassic dates. Alpine rims, with a range between 77 to 56 Ma,



were measured in all samples except FG1249, where the third rim was too thin (10µm) to be analysed. No correlation was found between rim types and ages within samples or amongst them.

Th/U ratios in detrital zircon cores range between 0.15 and >3, the rim generation with dates between 450–420 Ma have low Th/U ratios in sample FG1324 (0.006–0.14) whereas those in sample FG 1249 are between 0.1 and 0.5. The Carboniferous

to Triassic rims show Th/U ratios between 0.002– 0.4. The Alpine metamorphic rims have low Th/U ratios between 0.001– 0.01.

## 7 Discussion

### 7.1 Linking equilibrium conditions with time constraints

The age data deriving form *in situ* dating of allanite were linked to P-T results obtained from thermodynamic modeling using

several criteria, notably: (i) textural evidence, (ii) the presence of distinctive minerals as inclusions, and (iii) the presence of intergrowths of allanite with other phases (see section 4.4 Allanite textures and their microstructural relations; Table 5; Figs. 4, 6). Furthermore, the compositions of mineral inclusions in allanite were compared to those found in the matrix and to those predicted by thermodynamic modeling (representative chemical analyses in Supplement S3). Importantly, in four samples of the IC, the ages obtained for allanite cores and Alpine zircon rims overlap within error, suggesting coeval growth

of these two accessory minerals.

### 7.2 P-T-t paths of samples

The history of the studied samples is summarized in the P-T-t diagrams of Fig. 11 and compared with relevant data from the literature. The first metamorphic stage recorded in our samples from the IC is of Permian age and reflects granulite facies conditions. Porphyroclastic garnet cores (present in samples FG1315, FG12157, FG1347 and FG1249) are the only major

mineral relics of this HT stage. Among accessory minerals, late Carboniferous to Upper Triassic zircon rims (between 313 and 222 Ma) are observed, and monazite relics in allanite most probably are Paleozoic as well. The ages of the Permian zircon cores fit well with those reflecting HT metamorphism in the Ivrea Zone (Vavra et al., 1999;Vavra et al., 1996;Kunz et al., 2017;Ewing et al., 2013), thus supporting the long held view that the Sesia Zone is closely related to the Ivrea Zone (e.g. Compagnoni et al., 1977). The single Upper Triassic date (222 ± 13 Ma) in sample FG1315 is similar to ages reported from

the Ivrea Zone by Vavra et al. (1999), which have been related to fluid alteration. The PT-conditions for the pre-Alpine HT stage are constrained between 0.6-0.8 GPa and 700-900 °C from the garnet cores of the IC samples. These results confirm earlier data for the IC (0.6-0.9 GPa, ~850 °C; Lardeaux and Spalla, 1991;Rebay and Spalla, 2001; Fig. 11). A retrograde pre-Alpine path through amphibolite facies conditions is evident in sample FG1249, where the first garnet rim generation yields ~0.6 GPa and 600 °C (Fig. 11b), again in fair agreement with previous data (0.3-0.5 GPa, 570-670 °C; Lardeaux and Spalla,

1991;Rebay and Spalla, 2001). A sketch of the chronology and thermodynamic modelling of one of these garnets (sample FG1249) is contained in Giuntoli et al. (submitted).





For the Alpine history, our samples from the IC indicate higher P and T than in the EC, based on detailed garnet compositional modelling (Lanari et al. 2017). The IC recorded a range of maximum pressures between 1.6 and 2 GPa at temperatures of 600-670 °C; the EC yields a maximum pressure of 0.8 GPa for 500 °C. Age constraints provided by allanite dating of our five samples from the IC fall into three groups:

- Group 1 with ages of ~73 Ma (from 77.2 ± 7.3 Ma to 72.4 ± 1.4 Ma, based on samples FG1324 and FG1249 respectively)

- Group 2 at ~65 Ma (from 65.4 ± 3.5Ma to 64.5 ± 4.3Ma, based on samples FG1315 and FG12157)

- Group 3 at 55.7 ± 4.5Ma (sample FG1347).

Metamorphic conditions for Group 1 show 1.6-1.75 GPa and 580-650 °C for the earliest stage. In Group 2, samples FG1315 and FG12157 yield not only the same age (within error), but also similar metamorphic conditions; the pressure difference of 0.1-0.2 GPa is probably within the uncertainty of the model, as shown by error bars in Fig. 11b. Notably, the same PT-conditions were derived from garnet rim1 of FG12157 in Group 2 and from garnet rim3 of sample FG1249 in Group 1. Summarizing, Groups 1 and 2 experienced similar P-T conditions but Group 2 attained these some 5-10 Myr later. Furthermore, the outermost garnet rim in sample FG12157 preserves evidence of a retrograde stage at 1.4 GPa and 650 °C. Sample FG 1347 (Group 3) shows the youngest allanite ages for a similar pressure range as Groups 1 and 2, but lower temperature conditions (~580-600°C). It thus appears that the samples from the IC reflect several stages of allanite growth, probably because rocks of slightly different bulk composition produced allanite by different metamorphic reactions (Engi, 2017). The three growth stages captured by our samples are at ~73 Ma, ~65 Ma, and ~56 Ma. The different P-T-t paths of Group1, 2 and 3 are interpreted to represent different continental sheets (Giuntoli and Engi, 2016) that experienced similar PT conditions but at different times (further discussed in section 7.3 Assembly and exhumation of the Sesia Zone).

In the EC, sample FG1420 shows pre-Alpine allanite ages of ~290 Ma for the magmatic allanites. These age data probably reflect magmatic crystallization ages of the granitoids; they are in good agreement with several age data of magmatic zircon and monazite for the Sesia-Dent Blanche nappe (Bussy et al., 1998 Fig. 1). In the same sample, metamorphic allanites yield ages of 73.7 ± 8.2 Ma associated with metamorphic conditions of 0.8 GPa and 350 - 500°C, a stage constrained by the growth of garnet and phengite, the mica marking the main foliation. Sample FG12107 yields an allanite growth age of 62.8 ± 3.3 Ma (Fig. 11b-c highlighted by the star). Chlorite and white mica data from this sample give thermobarometric results in agreement with FG1420. The proximity of the two samples in the field, their similarity in PT-conditions and in the textural features of allanite lead us to suggest that allanite grew at the same metamorphic stage in these two samples. Their (nominal) ages seem discrepant, but considering the overlapping (2 sigma) uncertainties, we view them as a single age group (Group 4: ~63±3 Ma).

Comparing P-T-t data for the IC and EC, we note that Group 2 (in the IC) and Group 4 (in the EC) recorded the same age data of ca. 65 Ma, but very different metamorphic conditions. This implies that the IC and EC at that time were at completely different structural position in the subduction zone. Assuming lithostatic pressures, a difference of 1 GPa





translates to a vertical distance of ~30 km between the two complexes. Note that Group 4 also displays ~150 °C lower temperatures compared to Group 2.

The retrograde PT-trajectories of the two complexes are similar, as suggested by the chlorite –white mica equilibria: ~0.8 GPa and 340 °C for samples FG1315, FG1347 (IC) and sample FG1420 (EC; Fig. 11b). The EC also records two further

retrograde stages at 0.60 GPa, 330 °C and 0.42 GPa, 220°C respectively. Sample FG12157 recorded a slightly lower pressure and higher temperature (0.54 GPa and 393 °C) compared to the two other samples of the IC.

Regis et al. (2014; Druer slice) and Konrad-Schmolke and Halama (2014) suggested similar P-T-t paths for different parts of the IC (Fig. 11c), with the highest pressure reached at ~85 Ma. Halama et al. (2014) constrained the development of the retrograde blueschist facies Tallormo Shear Zone (Konrad-Schmolke et al., 2011) at 65.0 ± 3.0 Ma using Ar-Ar data on

phengite. This shear zone was related to external fluid influx occurring approximately at 1.35 GPa and 550 °C (Konrad-Schmolke and Halama, 2014). These two P-T paths are similar to Group 3 of the present study, but the latter are up to 30 Ma younger. Groups 1 and 2 consistently display similar P-T paths, but temperatures are 50 – 100°C higher than those determined by Regis et al. (2014) and Konrad-Schmolke and Halama (2014). In our samples from the IC, we detected no evidence of pressure cycling such as documented by Rubatto et al. (2011) further south, in the Fondo slice (Regis et al.,

15    2014).

Inger et al. (1996) and Reddy et al. (1999) constrained the greenschist facies overprint of the EC between 46 and 38 Ma (Rb-Sr using phengite). In the light of our data, we interpret these ages to reflect the greenschist facies conditions documented in both complexes (constrained between ~0.8 GPa and 400 °C to 0.4 GPa and 300 °C). The allanite core ages in Group 4 would then be related to a previous deformation event, as suggested by the epidote rim marking the greenschist stage (Figs. 4g,

11c). Zircon fission track ages for the Sesia Zone range from 42 to 26 Ma (Hurford and Hunziker, 1985;Hurford et al., 1989;Hurford et al., 1991;Berger et al., 2012;Wagner and Reimer, 1972;Kapferer, 2010), suggesting that the terrain crossed the ~250°C isotherm during exhumation during this age range. For the most internal section of the Sesia terrain, age constraints for the final exhumation to the surface are provided by the overlying Biella Volcanic Suite dated at 32.5 Ma (Kapferer et al., 2012;Kapferer, 2010).

**7.3 Assembly and exhumation of the Sesia Zone**

The IC shows several tectonic sheets, from several hundred meters to a few kilometres in thickness (Giuntoli and Engi, 2016), some of which may have moved independently (Rubatto et al., 2011;Regis et al., 2014) at some stages of the evolution. Some of the samples studied, while taken at most a few kilometres apart in the field, recorded similar P-T paths at different times, as reflected by the three age groups identified. This age difference may reflect relative mobility between such

sheets, which are notoriously difficult to delimit in this terrain (Giuntoli and Engi, 2016).

The differences in the PT-conditions (at eclogite facies) documented above do not indicate trajectories as tortuous as those suggested by numerical models for ablative subduction (e.g. Stöckhert and Gerya, 2005;Roda et al., 2012). Our P-T-t data are more indicative of progressive accretion at depth of ~60 km, juxtaposing a series of continental sheets (Vitale Brovarone





et al., 2013;Regis et al., 2014). Eclogite facies conditions in the IC evidently prevailed for an extended period, at least ~15 Myr.

In the IC, the main deformation stages and mineral reactions were related to pulses of external fluid passing through the rocks, as reported by Engi et al. (in preparation) and Giuntoli et al. (submitted). Repeated fluid influx occurred at eclogite

facies conditions, as shown by resorption and growth features in garnet and zircon, and these hydrous fluid pervasively rehydrated rocks that had previously been almost completely dehydrated by Upper Paleozoic metamorphism (Engi et al., in preparation). Thus Permian Kinzigites (granulite facies metapelites) were transformed back to micaschists during Alpine subduction at eclogite facies conditions. As allanite and zircon ages from these samples are identical within analytical uncertainties, it appears that this fluid influx also triggered the coeval crystallization of several accessory phases. Later and

more localized fluid influx has also been documented at blueschist facies conditions (Konrad-Schmolke et al., 2011), and this probably continued to greenschist facies conditions, as reflected by the partial overprint of the main eclogite assemblages, which is locally observed in the IC. Furthermore, our samples in the IC indicate ~50° C higher temperatures than those previously proposed in the literature (Fig. 11). This observation can be related to a temperature increase in external areas (NW) of the IC and may be linked to effect of (re)hydration of these pre-Alpine HT rocks, since this process is

exothermic (e.g. Peacock, 1987;Lyubetskaya and Ague, 2009).

A counterclockwise PT-path is proposed here for the EC. This trajectory is based on the results of garnet and chlorite + white mica modeling. The path as shown in Fig. 11 implies that, at nearly isobaric conditions, this area of the EC experienced cooling by 100-150 degrees, possibly related to the entry of cold material into the subduction channel, as already proposed by Pognante (1989) for the southern Sesia Zone. Piemonte-Liguria oceanic units are a likely source of such material, which

would be in line with age data of 58-40 Ma for the HP metamorphism in the Zermatt-Saas Zone (e.g. Rubatto et al., 1998;de Meyer et al., 2014;Weber et al., 2015) and with the kinematic model for the evolution of the Sesia–Dent Blanche nappes (Manzotti et al., 2014).

Summarizing, the major differences between the eclogite facies conditions recorded in the IC and the epidote blueschist facies condition in the EC are now quantitatively constrained by the P-T-t data presented in this study. Their juxtaposition

appears to have occurred at 0.7-0.9 GPa and 350-400 °C around 46-38 Ma (age data from Inger et al., 1996 and Reddy et al., 1999). These data confirm the interpretation of Williams and Compagnoni (1983) and Giuntoli and Engi (2016) of a first order tectonic contact existing between the two complexes. For the first time, we quantify the discontinuity in pressure and temperature at the contact between the two complexes to be of ~1 GPa and 100-180 °C. Note that in the same time range (75-60 Ma) the IC and EC may both have reached their subduction climax but at completely different P and T conditions.

Their juxtaposition only occurred during exhumation, some 20-30 Ma later.

The data shown in Fig. 11 allow us to derive average exhumation rates for the IC (Table 6). Using Group 1 and Group 3 as the two end members, i.e. the oldest and the youngest groups, a first calculation (Stage1) considers the interval from the highest pressures recorded in these two groups (~1.6 GPa at ~73 Ma and ~2 GPa at ~55 Ma, respectively) to the greenschist conditions at the juxtaposition with the EC (~0.6 GPa at ~38 Ma, age from Inger et al., 1996). The mean vertical exhumation





velocity obtained ranges from 0.9 to 2.7 mm/year respectively (Table 6). If we take the juxtaposition to be at ~46 Ma (Reddy et al., 1999) for the same P conditions, the mean vertical exhumation velocity increases to values of 1.2 and 5.1 mm/year, respectively. In these calculations we assumed a thickness of 20 km for the upper crust, 10 km for the lower crust, followed by upper mantle; densities of these were taken as 2.7, 3.0, and 3.3 g/cm$^3$, respectively (Rubatto and Hermann, 2001), with

subduction angles of ~90°, 60° and 45°. The rate of plate convergence between Adria and Europe in the time span of 68-38 Ma was ~15 mm/year (Handy et al., 2010).

For the final stage of exhumation of the IC and EC (Stage2), we computed mean vertical exhumation velocities from the juxtaposition condition, at ~0.6 GPa and ages as in the previous computation, to the surface, at 32.5 Ma. This age of the Biella Volcanic Suite (Kapferer, 2010;Kapferer et al., 2012) is supported by zircon fission track ages (Berger et al., 2012).

The mean vertical exhumation velocity ranges from 4 mm/year for the first case (juxtaposition at ~38 Ma), to 1.6 mm/year for the second case (juxtaposition at ~46 Ma; Table 6). For this late stage, only the vertical exhumation velocity was computed, and an average density of 2.7 g/cm$^3$ was assumed. The rate of convergence in the 35-20 Ma period was ~13 mm/year (Handy et al., 2010). This second set of data (Stage2) must be taken as approximate values of the exhumation rates, as the last part of the exhumation was not uniform everywhere in the Sesia Zone; notably local differences may be major

owing to brittle structures (e.g. Berger et al., 2012;Malusà et al., 2006). Furthermore, these values must be seen as minimum exhumation velocities, being the age of the Biella Volcanic Suite a maximum temporal constrain for the exhumation of the Sesia Zone to the surface.

Our first data set for the exhumation rate of the IC (Stage1) are in agreement with exhumation velocities proposed for the same complex by Zucali et al. (2002) and the mean exhumation rates for the Sesia Zone proposed by Roda et al. (2012).

However, our rates are up to an order of magnitude lower than those proposed by Rubatto et al. (2011) and the maximum values of Roda et al. (2012). Furthermore, the mean exhumation velocities for the final exhumation stage of the Sesia Zone (Stage2) are in the same range (some mm/year) as those for the exhumation of Groups 1 and 3. These data show no decrease in exhumation velocity from early exhumation (i.e. mantle to deep crustal positions) to late stages (i.e. crustal levels to the surface), as on the contrary had been proposed for ultra-high pressure terranes by Rubatto and Hermann (2001).

**8 Conclusions**

The present paper provides P-T-t data for the central the Sesia Zone, documenting in particular when and at what conditions the two main complexes of this HP-terrain were juxtaposed during the Alpine orogenic cycle.

In particular our data indicate that:

- In the Internal Complex the main stages of mineral growth, and probably the attendant deformation are related to
30        eclogite facies condition occurring between 77 and 55 Ma. Triggered by repeated influx of external fluid, it is during this subduction phase that most of the Permian HT-assemblages were replaced, leaving only sparse relics of essentially dry granulites.



- The Internal Complex comprises three groups of samples, probably reflecting different tectonic sheets (Giuntoli and Engi, 2016). These experienced similar internal deformation and P-T paths, but at different times, reflecting minor (km-scale) adjustments in the subduction channel at 50-60 km depth.

- Comparing the two main complexes, separate metamorphic evolutions emerge between 77 and 55 Ma, with conditions of 1.6-2 GPa and 600-670 °C in the Internal Complex, whereas only 0.7-0.9 GPa and ~500°C were reached in the External Complex.

- The two complexes were juxtaposed between 46 and 38 Ma (Inger et al., 1996;Reddy et al., 1999) at ~0.8GPa and 350°C, so at mid-crustal levels during exhumation.

- The Internal Complex was exhumed with mean vertical velocities of 0.9 – 5.1 mm/year, from the highest pressures recorded to the greenschist facies conditions attained upon juxtaposition with the External Complex (Stage1)

- The final exhumation of the entire Sesia Zone (Stage2) occurred with mean vertical velocities between 1.6 and 4 mm/year, but the late stages of exhumation in the area are fairly complex (Malusà et al., 2006;Berger et al., 2012)

This study shows that petrochronology,can be a powerful tool to quantify processes and unravel the tectonometamorphic evolution in complex geological settings, provided that detailed analytical work at the microscale can be linked to solid field evidence.

## Data availability

Original data underlying the material presented care available by contacting the authors.

## Competing interests.

The authors declare that they have no conflict of interest.

## Acknowledgments

This work was financially supported by the Swiss National Science Foundation (Project 200020-146175). We thank Roberto Compagnoni, Daniele Regis, and Jörg Hermann for fruitful discussions.

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

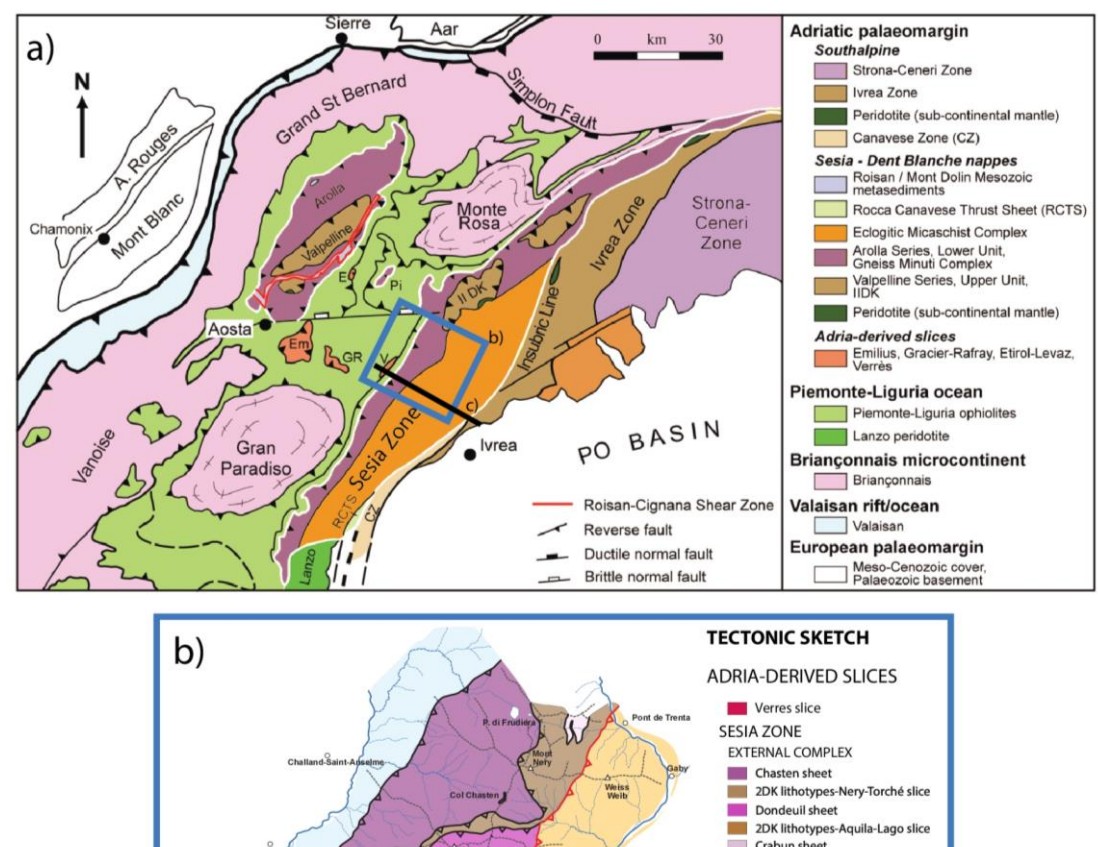

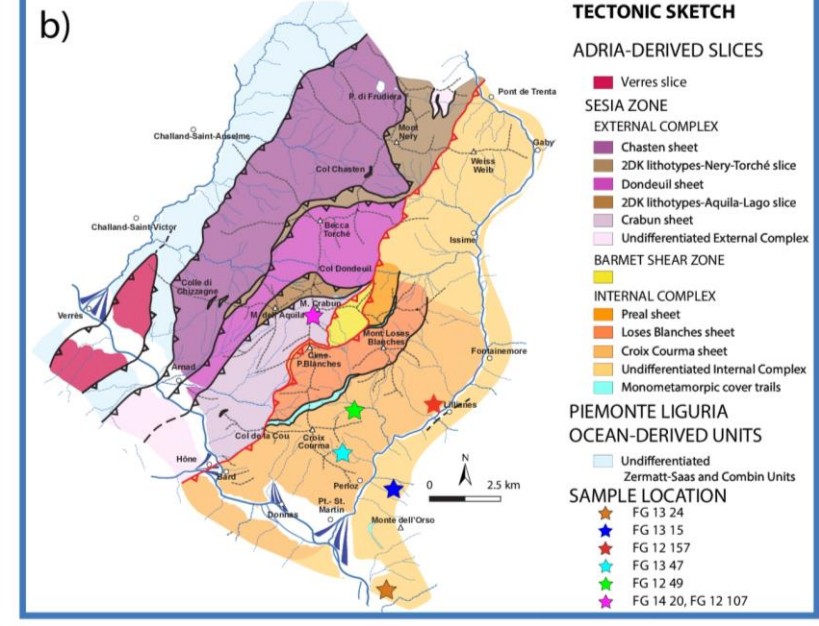

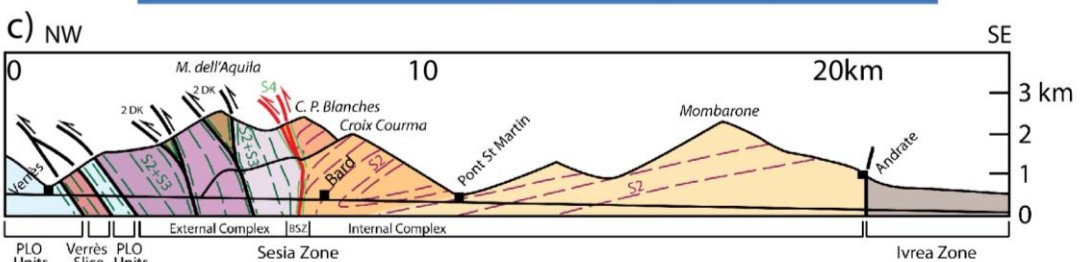



**Figure 1: (a) Simplified tectonic map of the Western Alps (modified from Manzotti et al., 2014). (b) Tectonic sketch of the study area (modified from Giuntoli and Engi, 2016). (c) Cross section through the study area (location show in a) with foliation traces: violet indicates the eclogite facies foliation (S2) of the IC, dark green indicates the composite epidote blueschist-greenschist facies foliation (S2+S3) of the EC, dark green indicates the greenschist facies mylonitic foliation (S4) at the contact IC-EC (modified from Giuntoli and Engi, 2016).**







**Figure 2: Thin section photos illustrating the metamorphic evolution of the studied samples. End-member mineral abbreviations used throughout text and figures are from Whitney and Evans (2010). (a) Eclogite foliation ($S_{ecl}$) marked by subparallel phengite, omphacite and glaucophane (plane-polarized light). (b) Eclogite foliation marked by the preferred orientation of chloritoid and phengite. Note the chloritoid crystal (left) is oriented perpendicular to $S_{ecl}$ and overgrowing it (plane-polarized light). (c) Garnet crystal with a bright pre-Alpine core and darker Alpine rims visible due to fine rutile inclusions (plane-polarized light). (d) Glaucophane crystal with a core displaying pale blue absorption colour, rimmed by darker blue crossite that marks a retrograde blueschist stage (plane-polarized light). (e) Colorless to pale green amphibole growing at the expense of omphacite during a retrograde greenschist stage (plane-polarized light). (f) Open folds with chlorite crystallizing in the hinge zone, marking a retrograde greenschist stage (plane-polarized light). (g) Garnet porphyroclast wrapped by the main greenschist foliation ($S_{gr}$); note the inner foliation inside the garnet ($S_{int}$; crossed-polarized light). (h) Brown magmatic allanite surrounded by finer grain epidote crystals (plane-polarized light).**









**Figure 3: (a) Mineral phases in sample FG1249 (compare with Fig. 2c) . (b) $X_{Grs}$ map highlights the porphyroclastic cores showing fractures sealed by higher $X_{Grs}$ garnet and three rims (Further garnet end-members maps for the samples are available in Giuntoli et al. submitted). (c) Mineral phases in sample FG1420 (compare with Fig. 2g). (d) Garnet displaying concentric zoning related to a decrease in $X_{Sps}$. (Further garnet end-members maps for the samples are available in the Supplement S2). (e) Phengite groups in sample FG1249 (details in text). (f) Phengite groups in sample FG1420 (details in text). Note larger phengite flakes of Group2 localized in fold hinges.**





**Figure 4: Back-scattered electron photos illustrating some of the allanite crystals in the studied samples. (a) Allanite intergrown with garnet and wrapped by a darker epidote rim. (b) Allanite including garnet, rutile and phengite. (c) Allanite displaying several growth zones and a dark outermost rim of epidote. Note the phengite inclusion and laser ablation pits (32 μm). (d) Allanite with numerous phengite and paragonite inclusions rimmed by darker epidote. (e) Allanite displaying several growth zones, the innermost of which are intergrown with paragonite. The tiny dark inclusions are fine grained phengite and paragonite. (f) Monazite with lobate edges preserved at the core of an allanite grain. (g) Allanite preserved at the core of epidote. Inclusions of paragonite and phengite are present at the boundary allanite-epidote rim. (h) Magmatic allanite with epidote crystals as satellites; some of the latter display cores of metamorphic allanite.**







**Figure 5: CL-images of zircon textures in the samples from the Internal Complex. Typically detrital cores show more or less resorption and are followed up by one to five metamorphic rim generations of different CL responses. The scale bar in all images is 50 μm.**







**Figure 6: Equilibrium phase diagrams of the studied samples computed with Theriak/Domino, assuming a free H$_2$O fluid, between 1.3-2 GPa and 400-700°C for (a), (b), (c), (d), (e) and between 0.3-1.5 GPa and 200-600°C for (f).**







**Figure 7: Multi-equilibrium thermobarometry results for the IC samples. (a), (c), and (e): intersection of the results deriving from** *Chlorite+Quartz+H₂O thermometry* **with** *White-Mica+Quartz+H₂O barometry*. **(b), (d), and (f): results of** *Chlorite+White-Mica+Quartz+H₂O thermobarometry*.

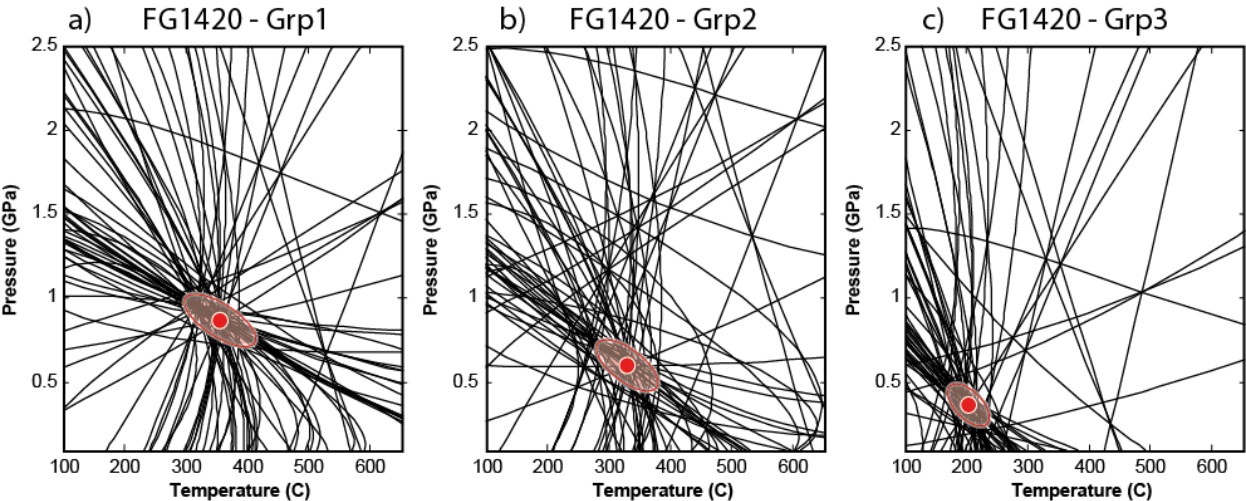

**Figure 8:** *Chlorite+White-Mica+Quartz+H₂O thermobarometry* **results for the EC sample.**





## AGE DATA-Internal Complex

**Tera-Wasserburg diagram**  **Sample**  **Th–isochron diagram**

**FG 13 24**

initial $^{207}Pb/^{206}Pb$:0.8398 ± 0.0097
Intercept Age:77.1693 ± 7.2294
MSWD:2.2428

isochron age: 77.7694 ± 11.9377
ini $^{208/206}Pb$: 2.0815 ± 0.0274
MSWD: 0.4580

**FG 13 15**

initial $^{207}Pb/^{206}Pb$:0.8229 ± 0.0181
Intercept Age:65.0300 ± 3.6123
MSWD:2.1307

isochron age: 65.8526 ± 4.4563
ini $^{208/206}Pb$: 2.0774 ± 0.0663
MSWD: 0.9825

**FG 12 157**

initial $^{207}Pb/^{206}Pb$:0.8290 ± 0.0088
Intercept Age:64.5110 ± 4.3763
MSWD:1.2168

isochron age: 64.5327 ± 6.3285
ini $^{208/206}Pb$: 2.0851 ± 0.0280
MSWD: 0.5501

**FG 13 47**

initial $^{207}Pb/^{206}Pb$: 0.7870 ± 0.0399
Intercept Age:55.6831 ± 4.4989
MSWD:2.5376

isochron age: 59.3781 ± 5.0689
ini $^{208/206}Pb$: 1.9796 ± 0.0817
MSWD: 0.39698

**FG 12 49**

initial $^{207}Pb/^{206}Pb$:0.8301 ± 0.0036
Intercept Age:72.5305 ± 1.4113
MSWD:1.7768

isochron age: 75.1703 ± 2.9426
ini $^{208/206}Pb$: 2.0771 ± 0.0137
MSWD: 0.5126



**Figure 9: Results of the IC samples plotted in the Tera-Wasserburg and $^{232}$Th/$^{206}$Pb$_c$ -$^{208}$Pb/$^{206}$Pb$_c$ isochron diagram with intercept and isochron age respectively and initial common lead composition estimations.**







**Figure 10: Results of the EC samples plotted in the Tera-Wasserburg and $^{232}$Th/$^{206}$Pb$_c$ -$^{208}$Pb/$^{206}$Pb$_c$ isochron diagram with intercept and isochron age respectively and initial common lead composition estimations.**








**Figure 11: P-T-t paths of the study area (Fig. 1). (a) Data and paths from previous studies: proposed pre-Alpine evolution from the data of Lardeaux and Spalla (1991), and Rebay and Spalla (2001); Alpine evolution of the Internal Complex from Rubatto et al. (2011), Regis et al. (2014), Konrad-Schmolke and Halama (2014), and Halama et al. (2014); ages for the External Complex from Inger et al. (1996) and Reddy et al. (1999), related to greenschist facies conditions. (b) Data presented in this study. Ellipses show P-T constraints for the age data, rectangles refer to retrograde stages. (c) Interpreted P-T-t paths. Full lines = Internal Complex; dashed lines = External Complex.**

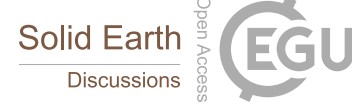



| Sample | FG1324 | | | FG1315 | | | | FG12157 | | | FG1347 | | | | | FG1249 | | | |
|---|---|---|---|---|---|---|---|---|---|---|---|---|---|---|---|---|---|---|---|
| Grt | | | | CORE | RIM 1 | RIM 2 | RIM 3 | CORE | RIM 1 | RIM 2 | CORE | RIM 1 | RIM 2 | RIM 3 | RIM 4 | CORE | RIM 1 | RIM 2 | RIM 3 |
| | Spot analyses (wt%) | | | Average composition (wt%) | | | | Average composition (wt% | | | Average composition (wt%) | | | | | Average composition (wt%) | | | |
| SiO2 | 37.75 | 37.93 | 37.74 | 37.70 | 37.96 | 37.81 | 37.44 | 38.17 | 38.43 | 38.74 | 37.75 | 37.88 | 38.18 | 37.75 | 37.33 | 36.91 | 36.62 | 37.80 | 37.98 |
| TiO2 | 0 | 0.01 | 0.04 | 0.01 | 0.01 | 0.01 | 0.01 | 0.07 | 0.07 | 0.17 | 0.02 | 0.02 | 0.02 | 0.03 | 0.03 | 0.05 | 0.16 | 0.27 | 0.04 |
| Al2O3 | 21.59 | 21.62 | 21.53 | 21.27 | 21.44 | 21.46 | 21.32 | 21.07 | 21.20 | 21.24 | 21.25 | 21.28 | 21.09 | 21.21 | 21.33 | 20.83 | 21.36 | 21.18 | 21.77 |
| FeO | 31.16 | 31.12 | 31.77 | 31.46 | 27.94 | 30.03 | 32.07 | 31.23 | 29.05 | 26.83 | 31.30 | 30.19 | 30.82 | 31.96 | 32.07 | 33.06 | 34.23 | 28.79 | 27.12 |
| MnO | 0.65 | 0.63 | 0.60 | 0.61 | 0.41 | 0.42 | 0.31 | 1.00 | 0.57 | 0.56 | 0.92 | 0.37 | 0.39 | 0.35 | 0.32 | 2.31 | 0.37 | 0.66 | 0.56 |
| MgO | 4.5 | 4.73 | 4.35 | 7.19 | 5.33 | 6.18 | 6.30 | 6.41 | 5.08 | 4.22 | 7.05 | 5.76 | 6.60 | 6.62 | 7.30 | 4.65 | 3.86 | 5.18 | 4.86 |
| CaO | 4.1 | 3.94 | 3.87 | 1.31 | 6.72 | 4.17 | 2.08 | 1.38 | 5.59 | 8.39 | 1.14 | 3.96 | 2.13 | 1.49 | 0.96 | 1.77 | 3.27 | 6.35 | 8.38 |
| Cr2O3 | 0 | 0.00 | 0.00 | 0.00 | 0.00 | 0.00 | 0.00 | 0.04 | 0.04 | 0.04 | 0.00 | 0.00 | 0.00 | 0.00 | 0.00 | 0.03 | 0.04 | 0.03 | 0.03 |
| Total | 99.75 | 99.97 | 99.90 | 99.56 | 99.81 | 100.08 | 99.52 | 99.37 | 100.03 | 100.18 | 99.41 | 99.47 | 99.24 | 99.41 | 99.33 | 99.62 | 99.90 | 100.27 | 100.73 |
| **Formulae based on 12 oxygens** | | | | | | | | | | | | | | | | | | | |
| Si | 2.99 | 3.00 | 2.99 | 2.97 | 2.97 | 2.96 | 2.96 | 3.02 | 3.02 | 3.03 | 2.98 | 2.99 | 3.02 | 2.98 | 2.94 | 2.95 | 2.93 | 2.96 | 2.95 |
| Ti | 0 | 0.00 | 0.00 | 0.00 | 0.00 | 0.00 | 0.00 | 0.00 | 0.00 | 0.01 | 0.00 | 0.00 | 0.00 | 0.00 | 0.00 | 0.00 | 0.01 | 0.02 | 0.00 |
| Al | 2.02 | 2.01 | 2.01 | 1.97 | 1.98 | 1.98 | 1.99 | 1.97 | 1.96 | 1.96 | 1.97 | 1.98 | 1.96 | 1.98 | 1.98 | 1.96 | 2.01 | 1.96 | 1.99 |
| Fe | 2.07 | 2.06 | 2.12 | 2.07 | 1.83 | 1.96 | 2.12 | 2.07 | 1.91 | 1.76 | 2.06 | 1.99 | 2.04 | 2.11 | 2.11 | 2.21 | 2.29 | 1.89 | 1.76 |
| Mn | 0.04 | 0.04 | 0.05 | 0.04 | 0.03 | 0.03 | 0.02 | 0.07 | 0.04 | 0.04 | 0.06 | 0.03 | 0.03 | 0.02 | 0.02 | 0.16 | 0.03 | 0.04 | 0.04 |
| Mg | 0.53 | 0.56 | 0.49 | 0.84 | 0.62 | 0.72 | 0.74 | 0.76 | 0.59 | 0.49 | 0.83 | 0.68 | 0.78 | 0.78 | 0.86 | 0.56 | 0.46 | 0.61 | 0.56 |
| Ca | 0.35 | 0.33 | 0.33 | 0.11 | 0.56 | 0.35 | 0.18 | 0.12 | 0.47 | 0.70 | 0.10 | 0.34 | 0.18 | 0.13 | 0.08 | 0.15 | 0.28 | 0.53 | 0.70 |
| Σ cations | 8.00 | 8.00 | 7.99 | 8.00 | 8.00 | 8.00 | 8.00 | 8.00 | 8.00 | 8.00 | 8.00 | 8.00 | 8.00 | 8.00 | 8.00 | 8.00 | 8.00 | 8.00 | 8.00 |
| **Molecular proportions of garnet end members** | | | | | | | | | | | | | | | | | | | |
| Alm | 0.69 | 0.69 | 0.70 | 0.68 | 0.60 | 0.64 | 0.69 | 0.69 | 0.63 | 0.59 | 0.68 | 0.66 | 0.67 | 0.69 | 0.69 | 0.72 | 0.75 | 0.62 | 0.58 |
| Prp | 0.18 | 0.19 | 0.17 | 0.28 | 0.20 | 0.24 | 0.24 | 0.25 | 0.20 | 0.16 | 0.27 | 0.22 | 0.26 | 0.26 | 0.28 | 0.18 | 0.15 | 0.20 | 0.18 |
| Grs | 0.12 | 0.11 | 0.11 | 0.04 | 0.19 | 0.11 | 0.06 | 0.04 | 0.16 | 0.24 | 0.03 | 0.11 | 0.06 | 0.04 | 0.03 | 0.05 | 0.09 | 0.17 | 0.23 |
| Sps | 0.01 | 0.01 | 0.01 | 0.01 | 0.01 | 0.01 | 0.01 | 0.02 | 0.01 | 0.01 | 0.02 | 0.01 | 0.01 | 0.01 | 0.01 | 0.05 | 0.01 | 0.01 | 0.01 |

9    **Table 1: Representative analyses of garnet of the samples**




| Sample Ph | FG1324 | | | FG1315 | FG12157 | FG1347 | FG1249 | FG1420 | |
|---|---|---|---|---|---|---|---|---|---|
| | Spot analysis (wt%) | | | Average | Average | Average | Average | Average | |
| wt% | | | | | | | | High Si | Low Si |
| SiO2 | 51.34 | 49.24 | 51.99 | 50.89 | 49.61 | 49.43 | 49.57 | 50.82 | 48.54 |
| TiO2 | 0.41 | 0.53 | 0.53 | 0.26 | 0.31 | 0.23 | 0.24 | 0.15 | 0.13 |
| Al2O3 | 26.67 | 26.82 | 26.83 | 28.15 | 27.58 | 29.78 | 27.92 | 27.45 | 30.17 |
| FeO | 1.67 | 1.39 | 1.64 | 1.49 | 2.09 | 1.40 | 1.30 | 3.21 | 2.63 |
| MnO | 0.00 | 0.00 | 0.00 | 0.06 | 0.19 | 0.06 | 0.05 | 0.08 | 0.08 |
| MgO | 3.00 | 3.37 | 3.45 | 3.68 | 3.65 | 3.29 | 3.45 | 2.85 | 2.30 |
| CaO | 0.00 | 0.00 | 0.00 | 0.00 | 0.19 | 0.01 | 0.01 | 0.00 | 0.00 |
| Na2O | 0.62 | 0.86 | 0.74 | 0.77 | 0.66 | 0.71 | 0.87 | 0.44 | 0.32 |
| K2O | 10.16 | 10.14 | 10.33 | 10.24 | 9.97 | 10.35 | 9.97 | 10.56 | 10.77 |
| Total | 93.87 | 92.35 | 95.51 | 95.54 | 94.26 | 95.25 | 93.37 | 95.56 | 94.93 |

**Formulae based on 11 oxygens**

| | | | | | | | | | |
|---|---|---|---|---|---|---|---|---|---|
| Si | 3.46 | 3.38 | 3.44 | 3.37 | 3.35 | 3.29 | 3.36 | 3.40 | 3.27 |
| Ti | 0.02 | 0.03 | 0.03 | 0.01 | 0.02 | 0.01 | 0.01 | 0.01 | 0.01 |
| Al | 2.12 | 2.17 | 2.09 | 2.20 | 2.19 | 2.34 | 2.23 | 2.16 | 2.39 |
| Fe | 0.09 | 0.08 | 0.09 | 0.08 | 0.12 | 0.08 | 0.07 | 0.18 | 0.15 |
| Mn | 0.00 | 0.00 | 0.00 | 0.00 | 0.01 | 0.00 | 0.00 | 0.00 | 0.01 |
| Mg | 0.30 | 0.35 | 0.34 | 0.36 | 0.37 | 0.33 | 0.35 | 0.28 | 0.23 |
| Ca | 0.00 | 0.00 | 0.00 | 0.00 | 0.01 | 0.00 | 0.00 | 0.00 | 0.00 |
| Na | 0.08 | 0.11 | 0.10 | 0.10 | 0.09 | 0.09 | 0.11 | 0.06 | 0.04 |
| K | 0.87 | 0.89 | 0.87 | 0.87 | 0.86 | 0.88 | 0.86 | 0.90 | 0.93 |
| ∑ cations | 6.94 | 7.01 | 6.97 | 7.00 | 7.01 | 7.02 | 7.00 | 6.99 | 7.02 |
| XMg | 0.76 | 0.81 | 0.79 | 0.82 | 0.76 | 0.81 | 0.83 | 0.61 | 0.61 |

10    **Table 2: Representative analyses of phengite of the samples**





| Sample | FG1315 | | FG12157 | | FG1347 | | FG1420 | | |
|---|---|---|---|---|---|---|---|---|---|
| Chl | | | | | | | *Grp1* | *Grp2* | *Grp3* |
| | Chl-35093 | Chl-35092 | Chl-204162 | Chl-204794 | Chl-214479 | Chl-215870 | Chl-910650 | Chl-547806 | Chl-881538 |
| SiO2 | 25.64 | 26.22 | 25.81 | 25.91 | 24.91 | 25.62 | 25.11 | 25.91 | 25.45 |
| Al2O3 | 19.62 | 19.71 | 20.64 | 20.1 | 21.7 | 20.97 | 20.09 | 21.63 | 20.34 |
| FeO | 26.29 | 26.68 | 24.53 | 27.34 | 24.96 | 26.45 | 26.87 | 28.98 | 25.53 |
| MnO | 0.05 | 0.05 | 0.21 | 0.33 | 0.11 | 0.19 | 0.28 | 0.31 | 0.27 |
| MgO | 15.32 | 15.82 | 15.87 | 14.32 | 13.51 | 13.49 | 13.01 | 12.61 | 12.61 |
| CaO | 0.05 | 0.06 | 0.03 | 0.03 | 0.13 | 0.09 | 0.08 | 0.09 | 0.17 |
| Na2O | 0.05 | 0.03 | 0.04 | 0.03 | 0 | 0.03 | 0.02 | 0.03 | 0.03 |
| K2O | 0.07 | 0.06 | 0.02 | 0.03 | 0.15 | 0.22 | 0.03 | 0.03 | 0.06 |
| **Atom site distribution (14 anhydrous-oxygen basis including Fe3+)** | | | | | | | | | |
| Si(T1+T2) | 2.68 | 2.71 | 2.70 | 2.69 | 2.65 | 2.72 | 2.73 | 2.70 | 2.78 |
| Al(T2) | 1.32 | 1.29 | 1.30 | 1.31 | 1.34 | 1.28 | 1.27 | 1.30 | 1.22 |
| Al(M1) | 0.32 | 0.29 | 0.30 | 0.31 | 0.34 | 0.28 | 0.27 | 0.30 | 0.22 |
| Mg(M1) | 0.28 | 0.33 | 0.35 | 0.27 | 0.24 | 0.31 | 0.30 | 0.27 | 0.30 |
| Fe2+(M1) | 0.20 | 0.26 | 0.27 | 0.21 | 0.21 | 0.32 | 0.33 | 0.32 | 0.32 |
| V(M1) | 0.20 | 0.12 | 0.09 | 0.21 | 0.21 | 0.10 | 0.10 | 0.11 | 0.16 |
| Mg(M2+M3) | 2.10 | 2.10 | 2.13 | 1.95 | 1.90 | 1.83 | 1.80 | 1.69 | 1.75 |
| Fe(M2+M3) | 1.46 | 1.61 | 1.64 | 1.57 | 1.62 | 1.87 | 1.94 | 2.03 | 1.85 |
| Al(M2+M3) | 0.41 | 0.26 | 0.19 | 0.44 | 0.43 | 0.24 | 0.21 | 0.24 | 0.34 |
| Al(M4) | 0.36 | 0.56 | 0.76 | 0.41 | 0.60 | 0.84 | 0.83 | 0.82 | 0.84 |
| XMg | 0.59 | 0.57 | 0.56 | 0.55 | 0.54 | 0.49 | 0.48 | 0.45 | 0.49 |
| Fe3+(M4) | 0.64 | 0.44 | 0.24 | 0.59 | 0.40 | 0.16 | 0.17 | 0.18 | 0.16 |

13    **Table 3: Representative analyses of chlorite used in the multi-equilibrium calculations**



| Sample | FG1315 | | FG12157 | | FG1347 | | FG1420 | | |
|---|---|---|---|---|---|---|---|---|---|
| Wm | | | | | | | *Grp1* | *Grp2* | *Grp3* |
| | Wm-14731 | Wm-14031 | Wm-232431 | Wm-231796 | Wm-245987 | Wm-245295 | Wm-883652 | Wm-558805 | Wm-851517 |
| SiO2 | 46.39 | 43.65 | 48.23 | 47.06 | 49.56 | 49.29 | 50.45 | 50.56 | 49.03 |
| Al2O3 | 27.05 | 27.06 | 30.09 | 29.98 | 31.01 | 29.62 | 28.75 | 28.95 | 27.97 |
| FeO | 1.92 | 1.62 | 2.5 | 2.32 | 2.64 | 2.83 | 2.48 | 3.13 | 3.28 |
| MnO | 0.07 | 0.06 | 0.27 | 0.2 | 0.06 | 0.06 | 0.09 | 0.1 | 0.13 |
| MgO | 2.87 | 2.56 | 2.67 | 2.71 | 2.89 | 2.98 | 2.27 | 2.52 | 2.44 |
| CaO | 0.03 | 0.02 | 0.27 | 0.23 | 0.01 | 0.01 | 0 | 0 | 0 |
| Na2O | 0.42 | 0.42 | 0.47 | 0.6 | 0.49 | 0.58 | 0.37 | 0.28 | 0.26 |
| K2O | 9.39 | 8.93 | 10.04 | 10.17 | 10.14 | 10.5 | 10.17 | 11.01 | 11.04 |
| **Atom site distribution (11 anhydrous-oxygen basis including Fe3+)** | | | | | | | | | |
| Si(T1+T2) | 3.33 | 3.26 | 3.24 | 3.22 | 3.25 | 3.28 | 3.37 | 3.34 | 3.33 |
| Al(T2) | 0.67 | 0.74 | 0.76 | 0.78 | 0.75 | 0.72 | 0.63 | 0.66 | 0.67 |
| V(M1) | 0.96 | 0.97 | 0.98 | 0.97 | 0.94 | 0.95 | 1.00 | 0.99 | 1.00 |
| Mg(M1) | 0.03 | 0.03 | 0.03 | 0.03 | 0.04 | 0.03 | 0.00 | 0.01 | 0.01 |
| Fe2+(M1) | 0.01 | 0.01 | 0.01 | 0.01 | 0.02 | 0.02 | 0.00 | 0.01 | 0.00 |
| Al(M2+M3) | 1.62 | 1.65 | 1.63 | 1.64 | 1.64 | 1.60 | 1.64 | 1.60 | 1.58 |
| Mg(M2+M3) | 0.28 | 0.26 | 0.24 | 0.25 | 0.24 | 0.26 | 0.22 | 0.24 | 0.24 |
| Fe(M2+M3) | 0.10 | 0.09 | 0.11 | 0.12 | 0.12 | 0.14 | 0.12 | 0.14 | 0.14 |
| XMg | 0.73 | 0.74 | 0.68 | 0.68 | 0.66 | 0.65 | 0.65 | 0.64 | 0.64 |
| K(A) | 0.86 | 0.85 | 0.86 | 0.89 | 0.85 | 0.89 | 0.87 | 0.93 | 0.96 |
| Na(A) | 0.06 | 0.06 | 0.06 | 0.08 | 0.06 | 0.07 | 0.05 | 0.04 | 0.03 |
| V(A) | 0.08 | 0.09 | 0.06 | 0.02 | 0.09 | 0.03 | 0.08 | 0.04 | 0.01 |

15    **Table 4: Representative analyses of white mica used in the multi-equilibrium calculations**





| | SAMPLE | AGE Aln (2σ) | AGE Zrn | P | T | CONSTRAINTS | NAME |
|---|---|---|---|---|---|---|---|
| IC | FG1324 | 77.2±7.3 | 77-63 | 1.65-1.75 Gpa | 600-650 | Inclusion of Ph, Grt | Omp, Grt, Gln and Rt micaschist |
| | FG1315 | 65.4±3.5 | 68-58 | 1.75-1.9 Gpa | 650-670 | Inclusion of Ph, Grt (Rim1-2-3), Pg | Grt, Ep and Rt Qz-micaschist |
| | FG12157 | 64.5±4.3 | 65-60 | 1.55-1.65 Gpa | 630-670 | Inclusion of Ph | Grt, Gln, Ep and Rt micaschist |
| | FG1247 | 55.7±4.5 | 56±4 | 1.8-2 Gpa | 580-610 | Inclusion of Ph, Pg, Rt | Cld, Grt and Rt micaschist |
| | FG1249 | 72.5±1.4 | - | 1.55-1.65 GPa | 580-630 | Inclusion of Ph, Pg | Grt, Ep and Rt micaschist |
| EC | FG1420 | 73.7±8.2 | - | 0.5-0.8 Gpa | 500-400 | Inclusion of Ph, Pg,Ttn | Grt orthogneiss |
| | FG12107 | 62.8±3.3 | - | no P | no T | Inclusion of Ph, Pg | Ep, Ph and Ab leucogneiss |

17 **Table 5: Constraints linking age data to pressure and temperature based on mineral inclusions in allanite; uncertainties**

18 **of age data are 2σ. Zircon dates are given as the minimum and maximum range of individual analysis.**



|  |  | Age at ~0.6Gpa | vertical velocity mm/year | along sub. inter. 60° mm/year | along sub. inter. 45° mm/year |
|---|---|---|---|---|---|
| **Stage1** | **Group 1** | **~38 Ma** | 0.9 | 1.1 | 1.3 |
|  | **Group 1** | **~46 Ma** | 1.2 | 1.4 | 1.7 |
|  | **Group 3** | **~38 Ma** | 2.7 | 3.1 | 3.8 |
|  | **Group 3** | **~46 Ma** | 5.1 | 5.9 | 7.2 |
| **Stage2** | **INT-EXT** | **~38 Ma** | 4 | - | - |
|  | **INT-EXT** | **~46 Ma** | 1.6 | - | - |

21 **Table 6: Exhumation velocities for Groups 1 and 3 (IC) from the highest pressures recorded to the greenschist**

22 **juxtaposition with the EC (Stage1) and for the late exhumation to the surface of the Sesia Zone (INT-EXT; Stage 2).**

