# Peer review of "Deeply subducted continental fragments: II. Insight from petrochronology in the central Sesia Zone (Western Italian Alps)"

_Solid Earth, 2017_

## Referee Comment (RC1) · S. Angiboust (Referee) · 2 Oct 2017

General comments

The manuscript "deeply subducted continental fragments: II. Insight from petrochronology in the central Sesia zone (W. Italian Alps)" from Giuntoli et al. deals with reconstructing P-T-t paths of the individual tectonic elements building up the Sesia zone. 40 yrs of geological investigations in this region enabled the identification of three main units, which according to recent work (e.g. Rubatto et al., 2011), may have more complex individual P-T-t histories than earlier thought. The authors combine modern thermobarometric tools together with allanite in situ dating to yield new P-T-t paths for

various samples across the Sesia zone. The paper is well written, clear, in good English and concise. This contribution, which constitutes one step forward to improve our knowledge about this tectonic zone, presents an interesting dataset and should for this reason be published in Solid Earth after moderate revisions. Yet, I don't fully agree with the tectonic interpretation and I have some comments about the proposed P-T estimates which strongly differ from previous work. These points should be carefully addressed before re-submission.

1. It is a quite risky and challenging task to date allanite and conclude about the meaning of the obtained age since we understand so little about its P-T stability field as well as the effect of retrogressive fluids on its replacement and/or re-equilibration. One important finding of this paper comes from the nice agreement between Aln and Zrn ages as shown in Table 5. This match does not mean that both methods date the metamorphic peak (I see four Zrn overgrowths on figure 5d: which one corresponds to the peak and which one you dated?). Retrogression and fluid-rock interaction can easily reset these geochronometers and the agreement between these two methods would, in that case, just mean that both minerals have been affected in the same way by the same event. In other words, a blueschist-facies retrogression event accompanied by fluid influx (Konrad-Schmolke et al., 2011) would easily explain the 56 Ma age data obtained for sample FG1247. No need for complex and chaotic thrusting inside the Sesia zone to explain this. I thought this point of view was important to mention and I would like to see some words about that in the discussion to show the reader the existing debate on the meaning of such ages.

2. I have been very surprised by the very high temperatures proposed by the authors. These estimates strongly depart from the previous estimates (c.100°C warmer on average). The re-hydration heat production is not enough to explain such enormous amount of heat needed. In the companion paper submitted to Solid earth, the authors report glaucophane inclusions in Grt 2 from sample FG12157 and propose P-T conditions of 650°C and 1.4 GPa for this event. Glaucophane is not stable at these

conditions. I would rather expect a barroisite. And we would have staurolite everywhere in all metapelitic rocks which is not the case in Sesia zone rocks. Last, the Raman thermometer on organic matter estimates by Giuntoli & Engi (2016) yields 520-615°C which is clearly less than what is presented here and in better agreement with previous works (Regis et al., 2014). This suggests some disequilibrium problems in the thermobarometric approach and makes me doubt about the robustness of some of these estimates. Maybe pre-alpine garnet resorption yields an artificial Mg-enrichment in the vicinity of the dissolution front and artificially "boosts" the temperatures towards higher values? Further discussion is needed here. I also suggest that the authors take a look on the recent paper by Angiboust et al. (2016) about Mt Emilius eclogitized granulites. Similar X-ray maps on similar rocks have been already published and these results should be used for comparison and discussed in both manuscripts. Note that the P-T conditions (obtained by conventional THERMOCALC average P-T mode: 500-550°C, 2.1-2.4 GPa) are sensibly different. Peak ages for Emilius-like slivers above the Zermatt-Saas unit are in the range 50-60 Ma (Weber et al. 2015; Fassmer et al., 2016). 3. There is a lack of field constrains to support the alleged P-T-t differences reported between the different group of samples (1, 2 and 3) inside the Internal Complex. I recommend to make a better use of the extensive and high-quality field dataset from Giuntoli & Engi (2016) to better highlight the link between P-T-t gaps and individual structural sub-units (if any).

Specific comments

There are a number of important references relevant for your study area which should be cited. P.2,L.10 and P.19,L.22: Angiboust et al. (2014). This paper proposes a vision significantly different than yours on the geodynamics and emplacement of the internal nappes (in line with previous works from Pognante, 1987 and Polino et al., 1990). This model should not be neglected in your work and some words presenting these models and comparing them to your results are needed here (in particular in section 7.3). I also believe that the paper from Beltrando et al., 2010 (Gond. Res.) should be

acknowledged in the geological setting section.

P.8: no Zrn found in the External Complex? Please state this explicitly.

P.17: why initial starting PT guess of 650°C? maybe this is the reason why your P-T estimates went astray...

P.11, L.7: isochemical twice

P.11, L.20: 6 vol.% of biotite is a lot for a biotite-free sample. I would not use phengite silica isopleths to constrain the peak with so much biotite predicted. I would rather consider this attempt as a fail and try with another sample or with a different micro-chemical domain. If you take the other intersect at 550°C (Fig.6a), the biotite amount would surely be lower – and thus closer from actual petrological observations.

P.13, L.7: "chlorite IS retrograde and recordS"

P.13, L.16: chloritoid

P.13, L.25: I am very puzzled by the meaning of 0.6 +/- 2.0 GPa pressure estimates...

P.19, L.28: in the same time range (75-60 Ma)? But 15 Ma is a lot of time! these two units may easily have been subducted diachronously (with the EC entering the subduction zone much later than the IC; see my comment above and the attached references).

P.20, L.30: I see no evidence here for peak, eclogite-facies metamorphism at 55 Ma. This allanite/zrn ages could date the blueschist-facies overprint associated with ex-humation and fluid ingression (Pognante 1987, Halama et al., 2014). I would advise to follow the same "petrochronological" strategy in the Tallorno shear zone and see what Aln and Zrn tell you.

Fig.1: FG1347 and FG1249 are very spatially close but have very different P-T esti-mates. Have you noticed any tectonic boundary between them? See my comment #3.

Fig.2: External complex (not ENTERNAL)

Fig.4: which ones are EC / IC ?

Fig.5: sure about the reference FG1347? (or rather FG1247?) Please provide spot location where the ablation holes have been made. This is very important to understand the meaning of your Zrn ages. Fig.6: EC/IC should be given (in the title, close to the sample number)

Fig.9: Th-isochron diagram for FG1420: why so much uncertainty? Lead loss or fluid-rock interaction?

Table 1: please provide EC garnet composition as well

Table 2: why you give the average composition? If so, you should mention the associated standard deviation

Table 3: please give totals for chlorite composition

Table 4: please give totals for phengite composition

I hope these comments helped. There is a still a long way until we really understand how the Sesia zone formed. This contribution does not solve all the problems but it provides some elements of the puzzle and raises important questions for future works.

Paris, 30/09/2017

---

## Referee Comment (RC2) · Anonymous Referee #2 · 13 Oct 2017

Review of Giuntoli et al., 'Deeply subducted continental fragments: II. Insight from petrochronology in the central Sesia Zone (Western Italian Alps)'

Summary

This paper uses the Sesia Zone (NW Italy) to understand both the conditions and timing of high pressure (HP) metamorphism and the nature of the assembly and exhumation of this preserved subduction zone. The authors use petrochronology to quantify the timing and conditions of metamorphism. Textural characterisation of samples followed by conventional thermobarometry, pseudosection and multi-equilibrium thermometry is used to define the both pre-Alpine high-temperature and Alpine HP metamorphic

events. Laser Ablation U-Pb geochronology on allanite and zircon is used to place timing constraints on the pre-Alpine and Alpine events and to produce P-T-t paths for the separate blocks of the Sesia Zone. The authors conclude that the two studied blocks of the Sesia zone experienced different HP histories, with the internal zone having a more prolonged period at depth as material was accreted into the subduction zone. The external zone experienced a shorter period at depth later in the history. The two units were subsequently juxtaposed during exhumation, which occurred at rates of between 1.6-4 mm/yr.

General comments:

This paper represents a worthwhile study of a topic of interest: understanding the rates and timescales over which high pressure metamorphic processes happen in continental subduction zones. The paper provides some insight into the prolonged accretion process that occurred in the Sesia zone subduction zone and demonstrates how blocks with distinct deformational and metamorphic histories were juxtaposed during exhumation. This paper is therefore of interest to a variety of scientists concerned with processes occurring in subduction zones and those concerned with the use of geochronology and metamorphic petrology to explain large-scale tectonic problems. Although I am not an expert on Alpine geology or HP metamorphic processes, I think this manuscript is of merit and with some edits could be a useful contribution for knowledge about processes occurring in continental subduction zones. The authors could better highlight the significance of the findings- particularly the accretionary processes into the subduction zone at depth.

Reviewer questions:

1. Does the paper address relevant scientific questions within the scope of SE? Yes although the significance and importance of the question could be better highlighted in the abstract, introduction and conclusions.

2. Does the paper present novel concepts, ideas, tools, or data? I am not an expert in

Alpine Geology so it is difficult to fully assess this as regards the topic. Methodologically the paper does not use new methods, however it does provide a solid dataset of allanite data- a relatively underused and novel geochronometer. The paper also uses the relatively new field of petrochronology, which with some edits to the discussion could be more effectively used to link the age and P-T data.

3. Are substantial conclusions reached? The paper is able to draw together a relatively large P-T-t dataset to explain the tectonic evolution of the area, however I think that the wider novelty and significance of the paper could be better highlighted, particularly for non-experts.

4. Are the scientific methods and assumptions valid and clearly outlined? Yes the methods are outlined, there are a few minor improvements for the presentation of the geochronology methods that could be done. The authors do not adequately outline the limitations of their study, particularly due to the fact that allanite crystals were not analysed in-situ but separated from the rocks.

5. Are the results sufficient to support the interpretations and conclusions? Yes largely, although the conclusions about the external zone are based on only one sample.

6. Is the description of experiments and calculations sufficiently complete and precise to allow their reproduction by fellow scientists (traceability of results)? yes

7. Do the authors give proper credit to related work and clearly indicate their own new/original contribution? Yes. Authors cite a lot of work that is 'in preparation' and 'submitted' which is not that useful, the authors may consider limiting the scope of the paper to mainly discuss the results presented in this manuscript.

8. Does the title clearly reflect the contents of the paper? Yes

9. Does the abstract provide a concise and complete summary? Yes, but I suggest adding something about the methods employed and the wider significance of the research question.

10. Is the overall presentation well structured and clear? Some improvements could be made by restructuring the discussion.

11. Is the language fluent and precise? Yes

12. Are mathematical formulae, symbols, abbreviations, and units correctly defined and used?

13. Should any parts of the paper (text, formulae, figures, tables) be clarified, reduced, combined, or eliminated? The paper could be shortened by making the text more concise, particularly by restructuring the discussion

14. Are the number and quality of references appropriate? Yes

15. Is the amount and quality of supplementary material appropriate? Yes, the authors may consider including concordia diagrams of the zircon data.

Good points

• The paper does a very good and thorough job exploring the P-T conditions of formation of the samples with a very solid investigation of the P-T conditions using multiple thermobarometry methods. The authors are clear in the limitations of the P-T work and it is clear that this is the strongest part of the manuscript, particularly the consideration of garnet growth. • This paper is appropriate for the journal, although the wider significance of the findings could be better discussed for a non-Alpine audience. • The authors employ sound methodology to produce a solid dataset to support the interpretations and conclusions. However, some of the interpretation and discussion of the data could be expanded to better discuss the tectonic implications and the potential impact of the study.

Suggestions for improvement

1. Impact of the research The manuscript needs to more clearly and simply state the importance, novelty and impact of the research question and results for the fields of

Alpine geology, HP metamorphism and petrochronology in the abstract, introduction and discussion/ conclusions.

2. Petrochronology The paper makes a good effort to highlight the need and methods for careful Petrochronological analysis of samples in order to adequately link ages to metamorphic processes. However there seems to be a discord between the aims and ideals of the authors and the reality of the data they present and the discussion of that data. This is not an unrepairable flaw- the authors just need to think about how to better organise their discussion to more adequately link the allanite ages with the careful P-T work that they have undertaken. This is possible but the manuscript needs to be more clearly written so that that reader can easily link together the ages, P-T data and interpretation

• The authors should explain why the allanite was separated from the rock rather than analysing the grains in-situ in thinsections- it would have been much easier to link the ages to the textures and therefore P-T data if they had been analysed in situ. The authors could think about maybe doing some extra analyses of the different textures observed to make sure of the ages of the interpreted textures, or if that extra work is not feasible, I recommend that the different inclusions in allanite are very clearly explained so that the reader can easily link the geochron and P-T work- perhaps with a table? • The authors may consider moving the section describing the allanite textures closer to the geochronology results so that they can be more effectively discussed with respect to the data. • It would be useful to have a petrochology discussion section separate from the main discussion- here you could carefully, clearly and systematically describe how the allanite ages, textures and P-T conditions fit together for each sample. • Did the authors collect any trace element data either from the allanite or the zircon, monazite or major phases such as garnet that could be used to link the geochronology to the P-T work. Could allanite be included in the pseudosection to show where these crystals were growing in the P-T path?

3. Development of discussion • As discussed above, it would be useful if you could

separate out the petrochronological and geological discussions. If you clearly took the reader through step by step how all the allanite data is linked to the P-T data and then in a separate section explain how that data fits with previously published data. You could then discuss the wider geological implications of the data. • It would be useful to present a diagram- perhaps a schematic cross section or cartoon explaining the spatial correlation between units and age- what is the relationship in time and space between the IC and EC and could you show this in a figure? • There are lots of references throughout the manuscript to other manuscripts in preparations or submitted by the authors, I am not sure the journal policy on this, but I would recommend that the authors restrict the discussion to based on evidence presented in this manuscript rather than lots of other data that is not evident to the reader. This is particularly true in the section about fluids in the crust- if the authors would like to discuss this they should present more evidence for the interaction of fluids in the discussion, particularly of the accessory phase textures.

Other suggestions for improvement

1. Make sure you outline the reproducibility of your secondary reference material and explain if the quoted ages include propagated uncertainties taking into account the long term reproducibility of these secondary standards. 2. Quote how many analyses make up each age in the results section. 3. The discussion of the External complex interpretations is only based on one sample- perhaps consider more clearly stating the limitations of your study- and perhaps how viable interpretations based on one sample are. 4. Add grid references of samples somewhere in the manuscript or supplementary data 5. Data tables- where is the allanite data presented in a data table? 6. Include the concordia plots of the zircon data.

Also see annotations on attached PDF.

Please also note the supplement to this comment:
https://www.solid-earth-discuss.net/se-2017-88/se-2017-88-RC2-supplement.pdf

**Supplement:**

[revised manuscript text omitted]

---

## Author Response (AR1)

Here we reply to the comments of **Angiboust-Referee #1**. Below, we show these *in italic*, our responses to them in straight text.

**Note**: at the end of this document are available the modified figures and the Concordia diagrams, according to our replies to the two referees.

**Summary statement**

*1. It is a quite risky and challenging task to date allanite and conclude about the meaning of the obtained age since we understand so little about its P-T stability field as well as the effect of retrogressive fluids on its replacement and/or re-equilibration. One important finding of this paper comes from the nice agreement between Aln and Zrn ages as shown in Table 5. This match does not mean that both methods date the metamorphic peak (I see four Zrn overgrowths on figure 5d: which one corresponds to the peak and which one you dated?). Retrogression and fluid-rock interaction can easily reset these geochronometers and the agreement between these two methods would, in that case, just mean that both minerals have been affected in the same way by the same event. In other words, a blueschist-facies retrogression event accompanied by fluid influx (Konrad-Schmolke et al., 2011) would easily explain the 56 Ma age data obtained for sample FG1247. No need for complex and chaotic thrusting inside the Sesia zone to explain this. I thought this point of view was important to mention and I would like to see some words about that in the discussion to show the reader the existing debate on the meaning of such ages.*

We certainly agree with Angiboust that linking the allanite age to the PT data of crystal growth is essential. For this purpose, we documented the microstructural context of allanite and all diagnostic mineral inclusions (section **4.4 Allanite textures and their microstructural relations** pg. 7, Table 5 and **7.1 Linking equilibrium conditions with time constraints** pg. 16). We note that, in all the dated samples, phengite inclusions in allanite cores invariably show the same composition as the phengite marking the eclogite facies foliation (Pg.7 lines 20-30). Both show the highest Si-contents (hence pressure) recorded by the sample. Phengite inside allanite also marks the same high-pressure foliation as in the matrix, and allanite grains are aligned with this same fabric. Allanite in samples FG1324 and FG1315 also contains inclusions of garnet the composition of which corresponds to the eclogite facies garnet (Pg.7 lines 20-30). For all these reasons, we conclude that allanite grew at eclogite facies conditions.

We also agree with Angiboust that "*Retrogression and fluid-rock interaction can easily reset these geochronometers*". Therefore, the allanite grains chosen for age dating were first BSE-imaged (Pg. 4 lines 29-30; Fig. 4). BSE pictures are well suited to reveal allanite growth zones and replacement textures, where present, such as epidote overgrowths. We reported "These rims may reflect minor retrograde stages that weakly altered the eclogite facies parageneses as well. Again, a peripheral epidote rim is present." (Fig. 4; lines 12-16 Pg. 7). Except for these local rims, our allanite crystal cores display no trace or evidence of replacement or re-equilibration. This is in line with the age data for allanite cores (Fig. 9) being very consistent and showing no trend that would indicate re-equilibration. Moreover, we rarely found evidence of a localised retrograde blueschist imprint – a clear difference to some previously reported samples in the literature – such as in sample FG12157 (pg.5 line 29). In that sample, allanite grains do have a rim (visible in Fig. 4c. noted on pg. 7 line 14), but no ages were obtained from such growth zones (as explained on pg. 15 lines 2-4).

Regarding zircon, we have no constraints to link the age data to the eclogite facies stages, but we note the agreement with the allanite age data, as pointed out by Angiboust. Regarding the particular question "*I see four Zrn overgrowths on figure 5d: which one corresponds to the peak and which one you date?*" we will update figure 5 and indicate the laser spots in the revised version. Our intent behind this figure is to show texturally most representative grains for each sample – that was the main criterion used to select each image. However, our descriptions of the rim types in the text were based on observations made in all of the grains of a particular sample.

Concerning the interpretation of the young ages obtained by FG1347, we consider the possibility illustrated on Pg 17 lines 16-18, on Pg 18 lines 27-33, and on Pg 19 lines 1-4, but we do not suggested "*complex and chaotic thrusting inside the Sesia zone to explain this*".

*2. I have been very surprised by the very high temperatures proposed by the authors. These estimates strongly depart from the previous estimates (c.100°C warmer on average). The re-hydration heat production is not enough to explain such enormous amount of heat needed.*

Angiboust correctly remarks on the substantially higher temperatures proposed by our study, even though his assertion is at odds with well established estimates of the thermal effect of (de)hydration. That effect was documented as early as 1982 (e.g. Walther & Orville, Contrib Mineral Petrol, v.79, p.253). However, the reviewer's comment made us realize that this is not common knowledge. Hence we propose (a) to refer to the Walther & Orville (1982) results, and (b) to quantify the estimated thermal effect for our specific case, roughly as follows:

> Walther & Orville (1982) analysed the thermal effect of (de-)hydration reactions during regional metamorphism and found it to be substantial. When applied to the present case of (re-)hydration, heat capacity data indicate that heating the Permian protolith requires ~1.0 kJ/K per kg of leucogneiss. The enthalpy released upon partial hydration of this protolith (producing the water content typical of these micaschists, 1.5 wt-% $H_2O$) adds ~77 kJ/kg in enthalpy. Such hydration should thus result in a temperature increase of some 80 °. This estimate lends credibility to the P-T estimates obtained here, which indeed are 60-90 °C higher than some maximum temperatures recently reported from other parts of the Sesia Zone, e.g. 575°C  (Konrad-Schmolke & Halama 2014), 570-630°C for the Druer Slice (Regis et al. 2014), or >600°C for the Ivozio Complex (Zucali and Spalla 2011). In addition, Zr-in-rutile data reported by Kunz et al. (2017, their Table 3) gave 640 °C for one of the present samples (FG1249), confirming our results by an entirely independent method.

For these reasons, we see no compelling reason to doubt the P-T results presented or the technique used, which – as explained in Lanari et al. (2017) – is based on the intersection of garnet isopleth, a well-established method.

3. *In the companion paper submitted to Solid earth, the authors report glaucophane inclusions in Grt 2 from sample FG12157 and propose PT conditions of 650°C and 1.4 GPa for this event. Glaucophane is not stable at these conditions. I would rather expect a barroisite. And we would have staurolite everywhere in all metapelitic rocks which is not the case in Sesia zone rocks. Last, the Raman thermometer on organic matter estimates by Giuntoli & Engi (2016) yields 520-615°C which is clearly less than what is presented here and in better agreement with previous works (Regis et al., 2014). This suggests some disequilibrium problems in the thermobarometric approach and makes me doubt about the robustness of some of these estimates. Maybe pre-alpine garnet resorption yields an artificial Mg-enrichment in the vicinity of the dissolution front and artificially "boosts" the temperatures towards higher values? Further discussion is needed here. I also suggest that the authors take a look on the recent paper by Angiboust et al. (2016) about Mt Emilius eclogitized granulites. Similar X-ray maps on similar rocks have been already published and these results should be used for comparison and discussed in both manuscripts. Note that the P-T conditions (obtained by conventional THERMOCALC average P-T mode: 500- 550°C, 2.1-2.4 GPa) are sensibly different. Peak ages for Emilius-like slivers above the Zermatt-Saas unit are in the range 50-60 Ma (Weber et al. 2015; Fassmer et al., 2016).*

This is a good comment but to model such polyorogenic garnets with drastic compositional variations between the pre-Alpine core and the Alpine rims, it was not possible to use a conventional approach, as extensively discussed in Lanari et al., (2017). For this reason we developed a new approach and software (GRTMOD) that accounted for fractionation of the previous growth zones and dissolution and precipitation.

We also applied the method (GrtMod) used in this study with the same thermodynamic database to samples from the Mont Emilius (Burn, 2016) and we found conditions of 500-550°C in samples with similar bulk rock compositions of Sesia Internal Complex. The same temperature is also found with the same method in samples from the Zermatt-Saas unit (work in progress). This argument thus cannot be used to explain the temperature difference we obtained for the samples of this study. Any T estimate not taking into account garnet fractionation and resorption would produce shifted P-T conditions (e.g. Lanari & Engi 2017).

We discussed these higher T predicted by our thermodynamic modelling: at lines 12-15 pg. 19 we wrote "Furthermore, our samples in the IC indicate ~50° C higher temperatures than those reported so far from parts of the Sesia Zone (Fig. 11). This observation indicates a temperature increase in at least some of the external (i.e. north-westerly) units of the IC, which may be linked to effects such as shear-heating and (re)hydration in the pre-Alpine HT rocks (see also reply to the previous comment). All our T estimates are around 600°C, considering the uncertainties, as shown in Fig. 11b by the lines departing from the ellipses, except sample FG12157. Such variation of T probably reflects different tectonic sheets that experienced somewhat different P-T-t paths, as discussed in chapter **7.3 Assembly and exhumation of the Sesia Zone** (see further details in our reply to comment 3)**.**

Glaucophane may not be stable at these temperatures in mafic rocks, but how much of a calcic component appears in amphibole at higher temperatures strongly depends on bulk composition, and barroisite may not develop in low-Ca rocks, such as our sample.

Regarding the absence of staurolite remarked by Angiboust, we note that (a) its stability field is not extensive at high pressures, and (b) in equilibrium phase diagrams calculated for the compositions of our samples no staurolite is predicted to occur between 1.3-2 GPa and 400-700°C, except for FG1347, where staurolite is predicted to be stable below 1.6 GPa above 600°C (Fig. 6d), but our PT estimates are outside that stability field of staurolite – and the prediction is in agreement with the assemblage we observed.

For the above arguments, we disagree that such higher T are caused by *disequilibrium problems in the thermobarometric approach*. To extract compositions for modelling, garnet growth zones were carefully selected after detailed microstructural analysis of the compositional maps; such areas where chemically homogeneous and uniform amongst different garnet crystals, they displayed no evidence of enrichment or depletion in the major elements. We should probably stress this point, adding few lines in the section **5.1.2 Garnet thermobarometry using GrtMod**.

We are aware of the recent paper by Angiboust et al. (2016) on Mt Emilius, but here we consider just the Sesia Zone *sensu stricto* (see pg. 3, lines 5-8). These two units occupy different structural levels in the Alpine nappe stack (more detail in Giuntoli and Engi, 2016 page 3): Emilius is located below the Combin Unit, the Sesia Zone *sensu stricto* is located above it. We agree with previous studies and consider both of them as fragments of the Adriatic passive margin (e.g. Dal Piaz 1999; Beltrando et al. 2014), but a direct correlation of their Alpine tectonometamorphic history is hazardous.

4. *There is a lack of field constrains to support the alleged P-T-t differences reported between the different group of samples (1, 2 and 3) inside the Internal Complex. I recommend to make a better use of the extensive and high-quality field dataset from Giuntoli & Engi (2016) to better highlight the link between P-T-t gaps and individual structural sub-units (if any).*

As we wrote on pg 17 lines 16-20 "It thus appears that the **samples from the IC reflect several stages of allanite growth, probably because rocks of slightly different bulk composition produced allanite by different metamorphic reactions** (Engi, 2017). The three growth stages captured by our samples are at ~73 Ma, ~65 Ma, and ~56 Ma. The different P-T-t paths of Group1, 2 and 3 are interpreted to represent different continental sheets (Giuntoli and Engi, 2016) that experienced similar PT conditions but at different times (further discussed in section 7.3 Assembly and exhumation of the Sesia Zone)." and in section **7.3 Assembly and exhumation of the Sesia Zone** "The IC shows several tectonic sheets, from several hundred meters to a few kilometres in thickness (Giuntoli and Engi, 2016), some of which may have moved independently (Rubatto et al., 2011;Regis et al., 2014) at some stages of the evolution. Some of the samples studied, while taken at most a few kilometres apart in the field, recorded similar P-T paths but at different times, as reflected by the three age groups identified. **This age difference may reflect relative mobility between such sheets, which are notoriously difficult to delimit in this terrain** (Giuntoli and Engi, 2016)." We propose that these differences in P-T-t  trajectories experienced by these samples may reflect different tectonic sheets and/or an interplay between tectonic mobility and several stages of  hydration at eclogite facies triggering crystallization of the main parageneses (including allanite and zircon) to explain such different P-Tt paths. As visible from Fig. 1b, FG12157, FG1249 and FG1347 are located in the Croix Courma Sheet, as defined by Giuntoli and Engi (2016), because in the field no marker highlighting further tectonic boundaries were found inside that sheet (see section **3.3 Criteria used to subdivide units** pg. 5 of Giuntoli and Engi 2016 for the criteria used). However, fieldwork alone is insufficient to identify all the tectonic boundaries, as distinctive field markers are scarce in gneiss-terrains. Giuntoli and Engi (2016, Pg. 25) point out: "The present study confirms the presence of several such units and documents their spatial relation, though more such sheets probably remain to be discovered. … We recognize how difficult it is to trace tectonic contacts, even major ones such as between the Internal and External Complexes, in areas where markers (e.g. distinctive marble trails) are missing and similar lithotypes are juxtaposed…".

Fieldwork used in conjunction with petrochronology enhances our ability to disentangle the evolution of complex tectonometamorphic complexes.

**Specific comments**

*There are a number of important references relevant for your study area which should be cited. P.2,L.10 and P.19,L.22: Angiboust et al. (2014). This paper proposes a vision significantly different than yours on the geodynamics and emplacement of the internal nappes (in line with previous works from Pognante, 1987 and*

*Polino et al., 1990). This model should not be neglected in your work and some words presenting these models and comparing them to your results are needed here (in particular in section 7.3). I also believe that the paper from Beltrando et al., 2010 (Gond. Res.) should be acknowledged in the geological setting section.*
Agreed. We will thus refer to *Angiboust et al. (2014)* and add a few sentences to section 7.3 comparing our results with this model, as requested. We also agree to acknowledge *Beltrando et al., 2010 (Gond. Res.)* in the revised version, as it is a useful reference.

*P.8: no Zrn found in the External Complex? Please state this explicitly.*
Zircon in the External Complex displays pre-Alpine magmatic oscillatory zoning, no Alpine overgrowth zones have been found. We will add this sentence in the revised version.

*P.17: why initial starting PT guess of 650°C? maybe this is the reason why your P-T estimates went astray...*
The initial starting guess is a technicality used for reasons discussed in Lanari et al. (2017, sections **5.2.3. Stage 2 – go fast mode** and **6.3. Automated strategy [1]: limitation of multiple minima and solution finding**). The specific purpose is not to miss a minimum in this part of the PT space. This function searches a solution around the starting guess and follows the gradient in the objective function; there cannot be two local minima at high pressure (see Fig. 8 in Lanari et al. 2017). We will add a clarification in the revised paper, but more importantly we reject Angiboust's conjecture that our P-T estimates "went astray".

*P.11, L.7: isochemical twice*
Deleted

*P.11, L.20: 6 vol.% of biotite is a lot for a biotite-free sample. I would not use phengite silica isopleths to constrain the peak with so much biotite predicted. I would rather consider this attempt as a fail and try with another sample or with a different microchemical domain. If you take the other intersect at 550°C (Fig.6a), the biotite amount would surely be lower – and thus closer from actual petrological observations.*
This objection is justified, at least in part, but we attribute the predicted biotite saturation to inadequacies in the available solution models, especially for alkali-amphibole. In the present case, the discrepancy has no drastic implications, since all of the phases observed in the (high-variance) assemblage are in fact part of the model assemblage that buffers the phengite composition. In this sample, garnet and phengite are the most abundant mineralogical phases, and their composition is matched perfectly by the models used. The intersect at 550°C is no valid alternative, because it is completely out of the range of the isopleth intersections for garnet and phengite, while these match perfectly at the accepted intersect at 1.65-1.75 GPa and 600-650 °C. We stress that the high temperatures are specific for the present samples; the exact same method (and with similar bulk rock compositions) yields 500-550°C for eclogitic micaschists of Mont-Emilius, in agreement with Angiboust's results there. So the modelling results appear to be robust even though they predict biotite as an additionally stable phase for one of the present samples.

*P.13, L.7: "chlorite IS retrograde and recordS"*
Corrected

*P.13, L.16: chloritoid*
Corrected

*P.13, L.25: I am very puzzled by the meaning of 0.6 +/- 2.0 GPa pressure estimates...*
Our mistake, the correct (asymmetric) uncertainty is *0.6 +/- 0.2 GPa*. Corrected

*P.19, L.28: in the same time range (75-60 Ma)? But 15 Ma is a lot of time! These two units may easily have been subducted diachronously (with the EC entering the subduction zone much later than the IC; see my comment above and the attached references).*
This is correct, but not at all in disagreement with what we state (pg. 17 lines 31-32): "Comparing P-T-t data for the IC and EC, we note that Group 2 (in the IC) and Group 4 (in the EC) recorded the same age data of ca. 65 Ma, but very different metamorphic conditions". We could modify the sentence at *P.19, L.28, stating* "at ca. 65 Ma" (instead of 75-60 Ma). We do not claim to know which of the two units entered the subduction zone earlier – we do not have data to constrain this. But we discuss that in the same time range

these two complexes were experiencing very different PT conditions (~1 GPa and 100-180 °C less in the EC, as stated on line 29 pg. 19).

*P.20, L.30: I see no evidence here for peak, eclogite-facies metamorphism at 55 Ma. This allanite/zrn ages could date the blueschist-facies overprint associated with exhumation and fluid ingression (Pognante 1987, Halama et al., 2014). I would advise to follow the same "petrochronological" strategy in the Tallorno shear zone and see what Aln and Zrn tell you.*
We disagree with this comment: As we stressed in our reply to comment #1, all the evidence we have from the dated samples indicates eclogite facies, none of it blueschist facies. Even though we have no reasons to doubt the results reported by Konrad-Schmolke from the Tallorno shear zone, it is an excellent idea to include samples from there in a petrochronological study using allanite and zircon. However, we have not extended our study to samples from there.

*Fig.1: FG1347 and FG1249 are very spatially close but have very different P-T estimates. Have you noticed any tectonic boundary between them? See my comment #3.*
Indeed, these two sample localities are not far apart, and we found no markers highlighting a tectonic boundary between them in the field. However, absence of evidence is not evidence of absence – especially in such a high-strain gneiss terrain – so we really cannot exclude the possibility of a tectonic boundary. Please, note our reply to comment #3 as well.

*Fig.2: External complex (not ENTERNAL)*
Mistake, corrected

*Fig.4: which ones are EC / IC ?*
Specified in the revised version to make the figure clearer

*Fig.5: sure about the reference FG1347? (or rather FG1247?) Please provide spot location where the ablation holes have been made. This is very important to understand the meaning of your Zrn ages.*
Yes, the sample is FG1347, the error was in table 5 and we corrected it, in the revised version. We will update our figure to show the spot locations, as requested.

*Fig.6: EC/IC should be given (in the title, close to the sample number)*
Good advice, we will update the figure in the revised version

*Fig.9: Th-isochron diagram for FG1420: why so much uncertainty? Lead loss or fluid-rock interaction?*
Neither one of these reasons, the large uncertainty is due to the high common lead content of the sample.

*Table 1: please provide EC garnet composition as well*
OK, modified in the revised version

*Table 2: why you give the average composition? If so, you should mention the associated standard deviation*
These values are based on the area selected for each growth zone (using X-ray compositional maps in XMapTools). We can add the associated standard deviation in the revised version.

*Table 3: please give totals for chlorite composition*
Ok, added in the revised version

*Table 4: please give totals for phengite composition*
Ok, added in the revised version

*I hope these comments helped. There is a still a long way until we really understand how the Sesia zone formed. This contribution does not solve all the problems but it provides some elements of the puzzle and raises important questions for future works.*

We thank Samuel **Angiboust** for his many constructive comments.

Francesco Giuntoli, Pierre Lanari, Marco Burn, Barbara Eva Kunz and Martin Engi

**Below are available the modified figures and the Concordia diagrams that will be added in the revised manuscript, according to our replies to the two referees.**

[Figure]

**Figure 1: (a) Simplified tectonic map of the Western Alps (modified from Manzotti et al., 2014). (b) Tectonic sketch of the study area (modified from Giuntoli and Engi, 2016) with sample locations and P-T-t data (this study). (c) Cross section**

through the study area (location show in a) with **projection of the studied samples**. Foliation traces: violet indicates the eclogite facies foliation (S2) of the IC, dark green indicates the composite epidote blueschist-greenschist facies foliation (S2+S3) of the EC, dark green indicates the greenschist facies mylonitic foliation (S4) at the contact IC-EC; **BSZ Barmet Shear Zone, PLO Piemonte-Liguria Oceanic unit** (modified from Giuntoli and Engi, 2016).

[Figure]

**Figure 5: CL-images of zircon textures in the samples from the Internal Complex. Typically detrital cores show more or less resorption and are followed up by one to five metamorphic rim generations of different CL responses. The scale bar in all images is 50 μm. Solid circles correspond to 32μm LA-ICP-MS spots, while dashed circles are 16μm spot sizes. The dates are individual $^{206}$Pb/$^{238}$U spots analysis given in Ma.**

**S10 Concordia diagrams of the zircon Alpine age data**

Concordia plots of individual $^{206}Pb/^{238}U$ spots analysis of Alpine dates given in Ma (using the software Isoplot- Ludwig, K.R., 2003. Isoplot/Ex version 3.0. A Geochronological Toolkit for Microsoft Excel. Berkeley Geochronological Centre Spec. Pub., Berkeley 70.).

[Figure]

data-point error ellipses are 2σ

Here we reply to the comments of **Anonymous Referee #2**. We reported in Italic the **Anonymous Referee #2**

Note: in the document "Reply to Angiboust-Referee1" are available the modified figures and the Concordia diagrams, according to our replies to the two referees.

**Reviewer questions**

1. *Does the paper address relevant scientific questions within the scope of SE? Yes although the significance and importance of the question could be better highlighted in the abstract, introduction and conclusions.*

See our replies to the specific comments and **Annotations** (below)

2. *Does the paper present novel concepts, ideas, tools, or data? I am not an expert in Alpine Geology so it is difficult to fully assess this as regards the topic. Methodologically the paper does not use new methods, however it does provide a solid dataset of allanite data- a relatively underused and novel geochronometer. The paper also uses the relatively new field of petrochronology, which with some edits to the discussion could be more effectively used to link the age and P-T data.*

See our replies to the specific comments and **Annotations** (below)

3. *Are substantial conclusions reached? The paper is able to draw together a relatively large P-T-t dataset to explain the tectonic evolution of the area, however I think that the wider novelty and significance of the paper could be better highlighted, particularly for non-experts.*

We will highlight better the wider novelty and significance of the paper in the revised version, according to the specific comments and annotations (see below)

4. *Are the scientific methods and assumptions valid and clearly outlined? Yes the methods are outlined, there are a few minor improvements for the presentation of the geochronology methods that could be done. The authors do not adequately outline the limitations of their study, particularly due to the fact that allanite crystals were not analysed in-situ but separated from the rocks.*

Preliminary analysis of allanite for U-Th-Pb by LA-ICP-MS was indeed done *in situ*, on polished thin sections. However, to obtain more material for dating, allanite grains were subsequently separated and mounted in 1 inch resin mounts and polished.

There are several raisons why allanite grains were separated for dating:

- Grain mounts are more efficient as they allow selecting more spots for analysis in suitable areas.
- Grains mounts produce more likely equatorial cuts because polishing is optimized to obtain the largest sections possible at a given grain size. This is essential for LA-ICP-MS analysis of grains with several growth zones: if sectioning is not near-equatorial, one is more likely to drill through different growth zones.

We do consider petrographic control before dating to be essential. All mounts were imaged at the SEM with the BSE detector to document allanite textures. For each sample, both in the thin section and in the mounts, the allanite grains displayed the same textures and growth zones, allowing us to link the microtextural features found in thin sections to those seen in grain mounts. Yet grain mounts allowed us to obtain at least an order of magnitude more spot analyses per sample than thin sections. We note that U-Th-Pb ages from grain mounts analyses agree within uncertainty with the spot analyses made in thin sections.

We will add the details in this paragraph to the revised manuscript.

5. *Are the results sufficient to support the interpretations and conclusions? Yes largely, although the conclusions about the external zone are based on only one sample.*

This is correct for the PT data, but regarding age dating two samples were analysed (FG12107, FG1420, Table 5). We are aware of this limitation, which is due to the difficulty of finding samples of suitable composition in the External Complex, as we stated on pg. 4 lines 13-14; this also explains why there are no P-T-t data available in the literature from this important complex in the central Sesia Zone (pg.4 line 4). The dominant rock type is orthogneiss devoid of suitable minerals to obtain the Alpine P-T-t path, as discussed in Giuntoli and Engi (2016). We will put more emphasis on this point in the revised manuscript.

6. *Is the description of experiments and calculations sufficiently complete and precise to allow their reproduction by fellow scientists (traceability of results)? yes*

7. *Do the authors give proper credit to related work and clearly indicate their own new/original contribution? Yes. Authors cite a lot of work that is 'in preparation' and 'submitted' which is not that useful, the authors may consider limiting the scope of the paper to mainly discuss the results presented in this manuscript.*

All the citations "submitted" are referred to our companion paper (Giuntoli, F., Lanari, P., and Engi, M.: Deeply subducted continental fragments: I. Fracturing, dissolution precipitation and diffusion processes recorded by garnet textures of the central Sesia Zone (Western Italian Alps), Solid Earth, submitted.) that is currently at the review stage. The citations "in preparation" refer to Engi, M., Giuntoli, F., Lanari, P., Burn, M., Kunz, B. E., and Bouvier, A.-S.: Brittle deformation and rehydration of lower continental crust during subduction trigger pervasive eclogite formation. Implications on continental recycling, Geochemistry, Geophysics, Geosystems ($G^3$). At this stage this manuscript is still under review in $G^3$. We expect an editorial decision shortly and will update this reference or delete it if need be

8. *Does the title clearly reflect the contents of the paper? Yes*
9. *Does the abstract provide a concise and complete summary? Yes, but I suggest adding something about the methods employed and the wider significance of the research question.*

We will improve our abstract in the revised version, adding what the referee requested here

10. *Is the overall presentation well structured and clear? Some improvements could be made by restructuring the discussion.*

See our replies to the specific comments and **Annotations** (below)

11. *Is the language fluent and precise? Yes*
12. *Are mathematical formulae, symbols, abbreviations, and units correctly defined and used?*
13. *Should any parts of the paper (text, formulae, figures, tables) be clarified, reduced, combined, or eliminated? The paper could be shortened by making the text more concise, particularly by restructuring the discussion*

See our replies to the specific comments and **Annotations** (below)

14. *Are the number and quality of references appropriate? Yes*
15. *Is the amount and quality of supplementary material appropriate? Yes, the authors may consider including concordia diagrams of the zircon data.*

Thanks for this suggestion, we will include Concordia plots in the Supplementary Material.

**Good points**

- *The paper does a very good and thorough job exploring the P-T conditions of formation of the samples with a very solid investigation of the P-T conditions using multiple thermobarometry methods. The authors are clear in the limitations of the P-T work and it is clear that this is the strongest part of the manuscript, particularly the consideration of garnet growth.*
- *This paper is appropriate for the journal, although the wider significance of the findings could be better discussed for a non-Alpine audience.*

See reply to comment #3 of the **Reviewer questions**

- *The authors employ sound methodology to produce a solid dataset to support the interpretations and conclusions. However, some of the interpretation and discussion of the data could be expanded to better discuss the tectonic implications and the potential impact of the study*

See our replies to the specific comments and **Annotations** (below)

**Suggestions for improvement**

*1. Impact of the research The manuscript needs to more clearly and simply state the importance, novelty and impact of the research question and results for the fields of Alpine geology, HP metamorphism and petrochronology in the abstract, introduction and discussion/ conclusions.*

As stated in our replies to the Reviewer questions, we will improve this section in the revised version, according to the referee specific comments and annotations

*2. Petrochronology The paper makes a good effort to highlight the need and methods for careful Petrochronological analysis of samples in order to adequately link ages to metamorphic processes. However there seems to be a discord between the aims and ideals of the authors and the reality of the data they present and the discussion of that data. This is not an unrepairable flaw- the authors just need to think about how to better organise their discussion to more adequately link the allanite ages with the careful P-T work that they have undertaken. This is possible but the manuscript needs to be more clearly written so that that reader can easily link together the ages, P-T data and interpretation*

- *The authors should explain why the allanite was separated from the rock rather than analysing the grains in-situ in thinsections- it would have been much easier to link the ages to the textures and therefore P-T data if they had been analysed in situ. The authors could think about maybe doing some extra analyses of the different textures observed to make sure of the ages of the interpreted textures, or if that extra work is*

*not feasible, I recommend that the different inclusions in allanite are very clearly explained so that the reader can easily link the geochron and P-T work- perhaps with a table?*

See reply to comment #4 of the Reviewer Questions. The link of age data and PT-data is based on critical mineral inclusions in allanite. These are documented in detail in section **4.4 Allanite textures and their microstructural relations** (Pg. 7), Fig. 4, Table 5 and Table S3, where all the chemical analyses of phengite and garnet included in allanite are presented and compared with analyses of the minerals defining the eclogite facies fabrics. We will strengthen this essential point to make the link more explicit in the text.

- *The authors may consider moving the section describing the allanite textures closer to the geochronology results so that they can be more effectively discussed with respect to the data.*

Good idea, we can pair up these two sections.

- *It would be useful to have a petrochronology discussion section separate from the main discussion- here you could carefully, clearly and systematically describe how the allanite ages, textures and P-T conditions fit together for each sample.*

Ok, we will add a sub-heading to separate the petrochronology discussion from the main discussion and reiterate how we tied the ages to PT-data.

- *Did the authors collect any trace element data either from the allanite or the zircon, monazite or major phases such as garnet that could be used to link the geochronology to the P-T work. Could allanite be included in the pseudosection to show where these crystals were growing in the P-T path?*

No trace element data were collected in this study for allanite, zircon and monazite, as these would be of limited utility in linking the age data to specific parts of the P-T path. Given the complex and multiple growth zones in garnet, we relied on petrographic and microstructural observations to link the allanite age data (t) to the PT data, which were all derived from the assemblage of major phases. Unfortunately, thermodynamic models for allanite and monazite so far are quite preliminary and available for but a couple of endmembers (Engi 2017), hence allanite and monazite were not included in isochemical phase diagrams (pseudosections).

*3. Development of discussion*

- *As discussed above, it would be useful if you could separate out the petrochronological and geological discussions. If you clearly took the reader through step by step how all the allanite data is linked to the P-T data and then in a separate section explain how that data fits with previously published data. You could then discuss the wider geological implications of the data.*

See our replies to comment #2 of **Suggestions for improvement** above. Furthermore, in section **4.4 Allanite textures and their microstructural relations** we report for each sample what data and observations were used to link the allanite growth zones to the main mineralogical phases, and we summarize this in section **7.1 Linking equilibrium conditions with time constraints.**

- *It would be useful to present a diagram- perhaps a schematic cross section or cartoon explaining the spatial correlation between units and age- what is the relationship in time and space between the IC and EC and could you show this in a figure?*

We modify the cross section of Fig. 1c, adding the sample locations and P-T-t data, as suggested by the reviewer.

- *There are lots of references throughout the manuscript to other manuscripts in preparations or submitted by the authors, I am not sure the journal policy on this, but I would recommend that the authors restrict the discussion to based on evidence presented in this manuscript rather than lots of other data that is not evident to the reader. This is particularly true in the section about fluids in the crust- if the authors would like to discuss this they should present more evidence for the interaction of fluids in the discussion, particularly of the accessory phase textures.*

See our reply to comment #7 of the Reviewer questions

***Other suggestions for improvement***

*1. Make sure you outline the reproducibility of your secondary reference material and explain if the quoted ages include propagated uncertainties taking into account the long term reproducibility of these secondary standards.*

We will provide the ages of our secondary standards in a separate table in the supplementary material. We do not propagated the systematic uncertainties ($S_{sys}$), because the value appear to be quite variable depending the reference material used (see **Burn, et al., J. Anal. At. Spectrom., 2017, 32, 1359-1377.**, reference given in section **6.1.1 Allanite geochronology).** This is especially true for U-Pb system as all the magmatic allanite used as standard have excess $^{206}$Pb cause by the presence of $^{230}$Th. However we will clarify in the revised

manuscript that the uncertainties given in the text and figures do not include the systematic uncertainties as defined by **M. S. A. Horstwood et al., Geostand. Geoanal. Res., 2016, 40, 311–332.** The low MSWD values (~0.5) obtained on the Th-Pb ages clearly support this assumption; we slightly overestimate the total uncertainties without propagating $S_{sys}$.

*2. Quote how many analyses make up each age in the results section.*

Ok, we can add such information, even though it is evident in Figs. 9-10 and Table S9 in the Supplement. For Zircon Dating (pg. 15 lines 29–32) we will add in the revised version: Detrital cores 27 analyses; Permian 53 analyses; Alpine 34 analyses.

*3. The discussion of the External complex interpretations is only based on one sample- perhaps consider more clearly stating the limitations of your study- and perhaps how viable interpretations based on one sample are.*

See our reply to comment #5 of the Reviewer Questions.

*4. Add grid references of samples somewhere in the manuscript or supplementary data*

Such a table is already present in the supplement (**Table S8 Location of the studied samples**), as stated on pg.5 line 19 of the submitted manuscript.

*5. Data tables- where is the allanite data presented in a data table?*

They are presented in Table 5 and in Figs. 9, 10.

*6. Include the concordia plots of the zircon data.*

Yes, we will include them. See our reply to comment #15 of the Reviewer Questions

**Annotations on attached PDF**.

- *It would be useful if you could highlight up front in the abstract what the research question is and why this study is of importance.*

In our opinion this is essentially given in the abstract: we introduce the problem, we state why the study area is suitable to solve it and we say why this study is of importance (Lines 6-11 pg. 1). But we will add a more general statement about the research question up front.

- *Add a sentence here of what you did- methods*

Ok, we can add such sentence in the revised version

- *What is the signficance of this relatively long time period?*

At pg. 17 lines 18-20 we wrote " The three growth stages captured by our samples are at ~73 Ma, ~65 Ma, and ~56 Ma. The different P-T-t paths of Group1, 2 and 3 are interpreted to represent different continental sheets (Giuntoli and Engi, 2016) that experienced similar PT conditions but at different times (further discussed in section 7.3 Assembly and exhumation of the Sesia Zone). " and at Pg. 18 lines 26-30 "The IC shows several tectonic sheets, from several hundred meters to a few kilometres in thickness (Giuntoli and Engi, 2016), some of which may have moved independently (Rubatto et al., 2011; Regis et al., 2014) at some stages of the evolution. Some of the samples studied, while taken at most a few kilometres apart in the field, recorded similar P-T paths at different times, as reflected by the three age groups identified. This age difference may reflect relative mobility between such sheets, which are notoriously difficult to delimit in this terrain (Giuntoli and Engi, 2016)." See also our reply to Angiboust-Referee#1 comments

- *Highlight why these findings are significant.*

Ok, we will add a sentence or two in the abstract

- *An upfront selling point of this paper: Exhumed HP rocks provide an opportunity to understand processes happening deep under collisional orogens like the Himalaya today. Sesia is a window into those deep earth processes. Make sure that you make it very clear what the purpose and wider significance of the research is upfront.*

We agree with this suggestion: this message is located at line 26 of pg. 1. We trust that the present structure is transparent for the reader.

- *expand on what is not understood*

This is an introductory sentence; more on it follows (pg. 2 lines 1-14).

- *consider using another word*

Changed to "on the contrary"

- *Is this what you mean?*

Yes, ablated refers to "ablative subduction" (Tao, W. C., & O'Connell, R. J. (1992). Ablative subduction: A two-sided alternative to the conventional subduction model. Journal of Geophysical Research: Solid Earth, 97(B6), 8877-8904.). We will add this citation in the revised manuscript.

- *elaborate on this link?*

Modified for the sake of clarity in "suggest that fluid influx triggers deformation and mineral reactions in deeply subducted high-grade (granulite and amphibolite facies) domains."

- *Highlight the advantages over a modelling study?*

Ok, we will add one or two sentences about it.

- *Has this been introduced?*

Yes, the reference was given few lines later, but we can give this reference again here.

- *How is this related to the IZ?*

As we wrote in lines 8-9 at pg. 3 "The Insubric line is a major fault system of Oligocene-Neogene age that separates the SZ from the Southern Alpine," Furthermore, as we wrote at line 10 pg.3 "An important complex of the Southern Alps is the Ivrea Zone,"

- *Do all the geochronological/ thermochronological datasets record eclogite facies metamorphism?*

They did not, and that is exactly our point here (pg. 4 lines 7-8): "…but apart from those detailed above, none of the datasets have been linked in detail to petrogenetic conditions."

- *Did you consider undertaking Zr-in-rutile thermometry?*

Here we did not use Zr-in-rutile thermometry. However, Kunz et al. (2017) report data for one of our samples (FG 1249 from the Internal Complex); they show Zr-in-rutile thermometry (Table 3 of that study), and the maximum temperature obtained (640°C) matches the temperature we got for this sample using thermodynamic modeling (Fig. 6e).

- *Can you have a high pressure greenschist foliation?*

Yes, in the greenschist facies field you can have high pressure greenschist facies (e.g. 0.6 GPa) or low pressure (0.3 GPa)

- *of what?*

Of phengite, changed sentence in "In the EC, two distinct generations of phengite are distinguished based on their microtextural position" for the sake of clarity

- *You may consider moving the allanite texture descriptions to closer to the geochronology section*

Ok, we can move this section.

- *which other minerals?*

These include phengite, garnet, paragonite and rutile. We can specify these in the revised version

- *Were these dated?*

No, these were not dated.

- *Again you may consider moving this section down to be with the rest of the geochron.*

Ok, we can move this section.

- *Were the different rims resolvable when dated?*

We were able to distinguish between detrital cores, Permian rims (very sparsely developed), and Alpine rims. However, within the Alpine rims it was not possible to make absolute age distinctions between the rim generations. This is due to relative large uncertainties of the ages, as the narrow width of the rims only allowed small spot size during LA-ICP-MS measurements, which increases the analytical uncertainty.

- *Is there a reference to explain this?*

This is a textural description used by e.g.: Vavra G, Gebauer D, Schmidt R, Compston W (1996) Multiple zircon growth and recrystallization during polyphase Late Carboniferous to Triassic metamorphism in granulites of the Ivrea Zone (Southern Alps): an ion microprobe (SHRIMP) study. Contrib Mineral Petrol 122:337–358. and Root DB, Hacker BR, Mattinson JM, Wooden JL (2004) Zircon geochronology and ca. 400 Ma exhumation of Norwegian ultrahigh-pressure rocks: an ion microprobe and chemical abrasion study. Earth Planet Sci Lett 228:325–341. We will add these two citations.

- *Consider rewording this sentence*

Rephrased: "No resorption was allowed in sample FG1420, as garnet textures in the compositional maps show no evidence for corrosion".

- *What were these P-T guesses based on?*

The initial starting guess is a technicality used for reasons discussed in Lanari et al. ( 2017, sections **5.2.3. Stage 2 – go fast mode** and **6.3. Automated strategy [1]: limitation of multiple minima and solution finding**), The specific purpose is not to miss a minimum in this part of the PT space. This function searches a solution around the starting guess and follows the gradient in the objective function; there cannot be two local minima at high pressure (see Fig. 8 in Lanari et al. 2017). We will add a clarification in the revised paper (this reply is taken from our reply to Angiboust's Specific Comment p. 17)

- *Did any of the minerals involved in the metamorphic reactions contain Fe3+?*

In the studied rocks, some of the HP minerals do contain minor amounts of $Fe^{3+}$, but as we stated (pg. 10 line 1): "$Fe^{3+}$ was ignored because of the lack of analytical data and suitable ferric end-members in solid solution models"

- *Are any images shown for these textures?*

No, these are not shown in the manuscript because they add no particular information

- *Excellent very good approach!*
- Thanks
- *You may wish to make a 'results' heading to make navigation easier.*

This is a good suggestion, will be adopted in the revised version

- *Were the garnet isomodes useful for constraining the P-T?*

For constraining P-T the garnet isopleth intersections were used. The garnet modal abundances predicted in those P-T conditions were matching the observed values.

- *Why did you not analyse the allanite in-situ? That would have been a useful way of linking the dates to the textural context?*

See reply to comment#4 of the Reviewer questions.

- *Report the reproducibility of the secondary standards and state whether the quoted uncertainties include propogated errors to take into account the reproducibility of the secondary standard.*

See our reply to comment#1 of other suggestion for improvement

- *Why was iolite not used for the allanite data above?*

Simply because it is not possible to apply a non matrix-matched standardization in Iolite. See Burn et al. 2017 for detailed explanations.

- *Outline how many analyses contribute towards each of these ages.*

See our reply to comment#4 of other suggestions for improvement

- *Consistent with Stacey and Kramers 1975 common Pb values*

We commented on this a few lines below (14-18, pg. 15) "These values are close to the predicted values of model lead evolution of Stacey and Kramers (1975) for this time range (Fig.1 in Burn, 2016). The exception is sample FG1347, in which the Tera-Wasserburg diagram shows a $^{207}Pb/^{206}Pb$ intercept of 0.787 ± 0.04 (MSWD on the regression of 2.5) and displays a $^{208}Pb/^{206}Pb$ intercept on the $^{206}Pbc$-isochron diagram of 1.98 ± 0.082 (MSWD on the regression of 0.4). These values differ from the predicted values of Stacey and Kramers (1975)."

- *Is this shown in a figure anywhere? It would be useful to include in a concordia diagram (even in supplementary material).*

We will add such figure with Concordia plots in the supplement of the revised version.

- *typical of metamorphic zircon (i.e. Rubatto)*

True.

- *But it says in section 6.1.1 that the grains were separated- that is not in-situ.*

This is correct, it is *"in-situ"* only within allanite and zircon grains. We must modify our wording to state that the LA-ICP-MS spot location and diameter were chosen so as to resolve single growth zones of the mineral, rather than bulk grain analyses.

- *I would argue that as your allanite data was not collected in-situ in thin sections it is much more difficult to texturally link the allanite age data and the textures.*

Yes and no, we rely mostly on inclusions (of phengite, garnet, rutile) in allanite, which are readily visible in separate grains. See also our reply to comment#4 of Reviewer questions

- *Perhaps split into sub-headings of Pre-Alpine and Alpine*

Ok, we can do this in the revised version

- *Perhaps direct to relevant part of figure 11?*

Here we want to refer to all of it

- *Is there any chemical evidence that these garnet cores are HT?*

This conclusion is derived from the thermodynamic modeling results, as in this portion of the Alps granulite and amphibolite facies conditions were never reached during the Alpine history. Also, these conditions fit perfectly well with the ones of C; Lardeaux and Spalla, 1991;Rebay and Spalla, 2001, as we noted in lines 26-30 pg.16.

- *Make sure all acronyms are explained.*

We will add the missing ones in the revised version.

- *Why is this age not on figure 11?*

Because we do not have PT for this stage, and it is based on one single date.

- *State which samples these age groups are recorded in.*

This is stated, e.g. line 5 pg. 17 "based on samples FG1324 and FG1249 respectively"

- *What are the textural constraints for these ages?*

See reply to comment#2 of Suggestion for improvements

- *You need to explain how you have linked the ages to these metamorphic conditions*

See reply to comment#2 of Suggestion for improvements

- *How are these groups spatially organised- are they recorded in different samples*

Please, see lines 5-8 of pg. 17 and fig. 1b for sample location

- *State this before.*

It is stated, see lines 5-8 of pg. 17

- *It may be simpler here to separate the discussion into a two stage 'Petrochronological discussion' linking the ages and P-T conditions and then a second wider geological discussion thinking about the wider implications and citing relevant literature.*

Ok, we can do this in the revised version

- *Make sure the organisation is logical, I am finding the discussion a little hard to follow with the jumping between Petrochron discussion, discussion of preAlpine then Alpine, then preAlpine ages, and links to the wider context all seemingly jumbled up. I would recommend separating this out and trying to restructure to make this discussion flow a bit better and make it more logical for the reader to follow.*

Section **7.2 P-T-t paths of samples** first presents the data for the IC and then the data for the EC. In the revised version, we can use two sub-heading levels to make this clearer.

- *Why are they associated with those conditions?*

See our reply to suggestion for improvement, comment#3.

- *Explain these 4 groups in the same bullet point fashion as above- or show both in a table.*

Referee misunderstanding: we called these two samples "Group 4"

- *Try to show this in a schematic diagram? How the IC and EC are related in space and time in the crust.*

As stated earlier, we will modify the cross section in Fig. 1c, adding the sample locations and P-T-t data, as suggested by the referee.

- *Show this in a schematic diagram?*

See reply to comment above

- *This mention of fluids comes a little out of the blue- could you maybe explain earlier on why fluids may be important in your observed metamorphic textures, allanite-monazite-zircon textures etc., Try to base the discussion a little more on your manuscript rather than relying so much on lots of other submitted and in preparation manuscripts (even if that means simplifying the discussion a little).*

Ok, we will introduce the role of fluid earlier on in the revised version, as requested by the referee.

- *And what textures within the accessory phases support this coeval crystallisation with fluids?*

We linked several lines of evidence: 1. Fluid is necessary to mobilise the REE, Th and U elements to form allanite. 2. Fabric evolution (pg. 7, lines 9-10): "allanite prisms are elongate in the eclogite foliation, showing mutual intergrowth relations with other minerals defining this main foliation.", and 3. Age data  (pg. 19 lines 8-9): "As allanite and zircon ages from these samples are identical within analytical uncertainties, it appears that this fluid influx also triggered the coeval crystallization of several accessory phases."

- *Link to the figure and again try and make this discussion a little more logical, you could organise this*
    1. *Petrochronological discussion of linking your allanite ages to P-T constrains and textures.*
    2. *Putting that all together to define a P-T path for your samples and linking that to other contraints.*
    3. *Being arm wavy and linking to the wider significance.*

We will restructure the discussion section as follow:

7 Discussion

   7.1 Linking equilibrium conditions with time constraints: petrochronology

   Here we will expand this section, adding a discussion of linking age data to P-T constrains and textures for each sample, as suggested by the referee.

   7.2 P-T-t paths of samples

   For the sake of clarity, we will subdivide this chapter into subchapters, as suggested by the referee:

      7.2.1 Pre-Alpine-IC

      7.2.2 Pre-Alpine-EC

      7.2.3 Alpine-IC

Here we will expand on the wider significance of our findings, as requested by the referee.

- *Should this not be in the conclusions? The discussion could be restructured here to systematically discuss the HP metamrphism and then in a new section the timing, conditions and evidence for juxtaposition of the different blocks and the tectonic model that helps to explain that.*

This paragraph serves as a summary of the discussion before the conclusions

- *Make sure it is really clear why your paper is significant and clearly explain (for a non-expert) what is new and exciting about this.*

We can add the following paragraph in the revised manuscript in section "7.4 Implication for the subduction, assembly and exhumation of continental fragments"

"Our work highlights that subducted continental terranes can be composed of several complexes that experienced major differences in their subduction histories (i.e. ~1 GPa and 100-180 °C for the Sesia Zone). Furthermore, these complexes can have been juxtaposed by tectonic contacts during the latest metamorphic stages (i.e. greenschist facies conditions) before being jointly exhumed to the surface. These complexes can be lithologically heterogeneous and may comprise several tectono-metamorphic subunits (from a few hundred metres to several kilometres in thickness). These may experience similar P-T paths but at different times (5-10 Ma apart). Differences in the P-T-t trajectories would thus reflect different tectonic sheets and attest to tectonic mobility in the subduction zone and/or several stages of internal deformation plus hydration at eclogite facies that triggered a pervasive HP-fabric and -assemblage (including datable accessory minerals).

To unravel such complex histories in subducted continental terranes, carefully established field relations, followed by microstructural and petrochronological analysis need to be combined to

- map and identify the primary and secondary tectono-metamorphic contacts;
- characterize the different fabrics and the mineral phases defining them;
- quantify the differences existing in P, T and t for each complex or subunit.

Finally, the heterogeneity of subducted continental terranes highlighted by this study should be considered when comparing results to numerical and analogue models that aim to investigate the mechanisms responsible for the subduction and exhumation of the continental crust."

- *Why is this significant/ new/ important?*

As comment above

- *Should you also have a bullet point explaining the EC?*

Ok, we can add such a sentence in the revised manuscript

**Figures**

- *Make sure the stars are obvious*

We modify the figure to make the stars more visible in the revised version of the manuscript.

- *Consider putting the stars on the cross section?*

Ok, we are happy do this in the revised figure

- *Clearly highlight how the allanite relates to each of these textures?*

In these microphotographs allanite grains are not visible at this resolution because they are tiny. If needed, we could provide two small-scale microphotographs with allanite crystals and their relation with the main foliation.

- *These pseudosections may be a little bit small, perhaps consider splitting them across multiple pages?*

Ok, we can expand these in the revised manuscript

- *Clearly explain what the red, blue and green ellipses mean.*

It is explained in the legend below each Equilibrium phase diagrams: these are the modeled garnet growth zones (Rim1, 2, 3)

- *Is this all allanite data? Where is the zircon data?*

We will introduce the Concordia diagrams in the revised version

- *The slope of these elipses looks a little odd- perhaps something to do with the correlation of uncertainties?*

That's correct. The correlation of the uncertainty is included in the plots, as described in Burn et al. 2017. That is the way it should be done to obtain the best isochrones.

- *Perhaps explain in another figure how these different parts of the HP IC are linked spatially? If there is accretion over time can you show this in a cross section explaining how the samples relate to one another in time and space?*

This is in fact visible from Fig. 1, but we could remind the reader here.

- *What is this age based on? Intercept TW age?*

Yes, we will specify that in the caption.

- *Again state what age this is- i.e. 238-206, 235-207?*

Zircon $^{238}U/^{206}Pb$ dates are given as the minimum and maximum range of individual analyses. We will add this specification in the revised version.

- *It may be useful to add another table explaining the 3 groups in the IC and the four groups in the EC and their relative age and P-T conditions.*

Misunderstanding: in the EC there is one group (Group 4). This table already exists (Table 5). We can improve this table adding a column where we specify to which group the samples belong, according to the referee's request.

We thank the **Anonymous-Referee #2** for so many detailed, constructive comments.

Francesco Giuntoli, Pierre Lanari, Marco Burn, Barbara Eva Kunz and Martin Engi

**Deeply subducted continental fragments: II. Insight from petrochronology in the central Sesia Zone (Western Italian Alps)**

Francesco Giuntoli[1], Pierre Lanari[1], Marco Burn[1], Barbara Eva Kunz[1]& Martin Engi[1]

[1]Institute of Geological Sciences, University of Bern, Baltzerstrasse 1+3, 3012 CH-Bern

5    *Correspondence to*: Francesco Giuntoli (francesco.giuntoli@plymouth.ac.uk)

**Abstract.** Subducted continental terranes commonly comprise an assembly of subunits that reflect the different tectonometamorphic histories they experienced in the subduction zone. Our challenge is to unravel how, when and in which part of the subduction zone these subunits were juxtaposed. Petrochronology offers powerful tools to decipher pressure-temperature-time (P-T-t) histories of metamorphic rocks that preserve a record of several stages of transformation. A major issue is that the driving forces for re-equilibration at high pressure are not well understood. For example, continental granulite terrains subducted to mantle depths frequently show only partial and localized eclogitization. The Sesia Zone (NW Italy) is exceptional because it comprises several continental subunits in which eclogitic rocks predominate and high-pressure (HP) assemblages almost completely replaced the Permian granulite protoliths. This field-based study comprises both main complexes of the Sesia terrane, covering some of the recently recognized tectonic subunits involved in its assembly, hence our data constrain the HP-tectonics that formed the Sesia Zone. We used a petrochronological approach consisting of petrographic and microstructural analysis linked with thermodynamic modeling and U-Th-Pb age dating to reconstruct the P-T-t trajectories of these tectonic subunits. Our study documents when and at what conditions re-equilibration took place. Results constrain the main stages of mineral growth and deformation, associated with fluid influx that occured in the subduction channel. In the Internal Complex (IC), pulses of fluid percolated at eclogite facies conditions, between 77 and 55 Ma with the HP-conditions reaching ~2 GPa and 600-670 °C. By contrast the External Complex (EC) records a lower pressure peak of ~0.8 GPa for 500 °C, at ~63 Ma. The juxtaposition of the two complexes occurred during exhumation, probably at ~0.8GPa and 350°C; the timing is constrained between 46 and 38 Ma. Mean vertical exhumation velocities are constrained between 0.9 and 5.1 mm/year for the IC, up to its juxtaposition with the EC. Exhumation to the surface occurred before 32 Ma, as constrained by the overlying Biella Volcanic Suite, at a mean vertical velocity between 1.6 and 4 mm/year. These findings constrain the processes responsible for the assembly and exhumation of high pressure continental subunits, thus adding to our understanding of how continental terranes behave during subduction.

**1 Introduction**

The behaviour of continental crust subducted to high-pressure (HP) conditions is far from fully understood (Kylander-Clark et al., 2008;Rubatto and Hermann, 2001;Brun and Faccenna, 2008;Malusà et al., 2011;Angiboust et al., 2016). Seismic tomography beneath collisional orogens shows that large slab parts reached mantle depths (e.g. Lippitsch et al., 2003;Replumaz et al., 2010;Zhao et al., 2015), but the source of such remnants is hard to assess because the rocks cannot be directly investigated. However, where orogens contain exhumed continental HP-rocks, these offer opportunities to investigate some of the tectono-metamorphic processes involved, notably those responsible for the return flow of continental fragments back to the surface.

In recent studies on the Western Alps, the combination of tectonic and numerical modeling studies (e.g. Yamato et al., 2008;Faccenda et al., 2009) with the reconstruction of pressure-temperature-time (P-T-t) paths (e.g. Regis et al., 2014;Rubatto et al., 2011) has led to two possible end-members scenarios for the subduction of continental domains to HP conditions. Either the units essentially experienced tectonic slicing, followed by accretion to overlying continental units, thus assembling complexes composed of different tectonometamorphic slices (e.g. Regis et al., 2014;Manzotti et al., 2014;Vitale Brovarone et al., 2013;Angiboust et al., 2014). Alternatively, units were eroded or ablated in the subduction channel, which lost coherence and experienced substantial mixing, leading to diverse and complex P-T-t paths (e.g. Tao and O'Connell, 1992;Stöckhert and Gerya, 2005;Warren et al., 2008;Keppie et al., 2009;Roda et al., 2012). The distinction between these two end-members scenarios is important to understand the evolution of deeply subducted continental domains and possibly their paleogeographic provenance.

Crucial questions related to continental HP-units in orogens include: How rapidly were they subducted and to what depth? When and how fast were they exhumed? What P-T-t trajectories did they experience? When and how did fluids affect these continental fragments? To shed light on these questions, the sequence of metamorphic stages recorded in suitable samples needs to be analysed in detail (Engi et al., 2017). P-T-t paths promise insight into details of these fundamental tectonic processes. Petrochronological studies in the Western Alps have helped to constrain exhumation rates of Alpine HP and UHP continental domains. Calculated rates vary from a few mm (e.g. Zucali et al., 2002;Regis et al., 2014 for the internal part of the Sesia Zone) to a few centimetres per year (e.g. Rubatto and Hermann, 2001 for Dora Maira UHP massif). In detail, some studies found that, after a first phase of rapid exhumation up to lower crustal levels, exhumation rates markedly decreased to a few millimetres per year (Rubatto and Hermann, 2001;Yamato et al., 2008).

Generally speaking, how much of the subduction history is recorded in a sample is related to what processes triggered mineral recrystallization or equilibration. Several studies (e.g. Erambert and Austrheim, 1993;Austrheim, 1987;Rubie, 1986;Oliver, 1996;Etheridge et al., 1983;Engi et al., in review;Pennacchioni, 1996) proposed that fluid influx triggers deformation and mineral reactions in deeply subducted high-grade (granulite and amphibolite facies) domains. It is of interest to know when this happened in the P-T-t evolution, notably whether it occurred early, during subduction or only late, upon exhumation (e.g. Konrad-Schmolke et al., 2011).

We present a field-based study emphasizing P-T-t data and discuss their implications on such first order questions. Datasets from a well defined multidimensional analysis, i.e. pressure-temperature-time-deformation-space, are essential as a reference frame for numerical modelling studies. Our petrochronological approach highlights the complex and heterogeneous tectonometamorphic evolution in a polydeformed continental terrane. A key requirement in this approach is to establish reliable links between age data (t) and the P-T conditions of mineral formation (e.g. Schenk, 1989;Buick and Holland, 1989;Scott and St-Onge, 1995;Liati and Gebauer, 1999;Rubatto and Hermann, 2001;Foster et al., 2004;Janots et al., 2009;Gasser et al., 2012;Donaldson et al., 2013;Rubatto et al., 2011;Regis et al., 2014;Mottram et al., 2014). In this study, mutual inclusions relationships of the main mineralogical phases in the datable accessory minerals were used, along with microstructural analyses, to bracket mineral age data to P-T conditions.  The study area is the central Sesia Zone, located in the Western Alps (Valle d'Aosta, Italy). P-T-t data are reconstructed for the eclogite facies IC and, for the first time, for the epidote-blueschist facies EC. These data allow us to constrain the juxtaposition of the two complexes, which occurred under HP greenschist facies conditions, and to determine exhumation rates for the IC and for the assembly of the Sesia Zone.

**2 Geological setting**

[revised manuscript text omitted]

**3 Sampling strategy and petrochronological approach**

To document the polyphase history of the Sesia Zone, we reconstructed detailed P-T-t paths for five samples taken in the IC and two samples in the EC. Of over a hundred samples checked, a very small percentage fulfilled the requirements for such a study. In the EC, in particular, suitable material to quantify P-T-t conditions by present methods turned out to be very rare. This is mostly due to the fact that orthogneiss, the dominant rock type in the EC, is lacking in minerals suitable to obtain the Alpine P-T-t path, as discussed by Giuntoli and Engi (2016). Nevertheless, the samples analysed provide constraints to derive a P-T-t path for the EC as well, allowing us to determine when and at what conditions the two complexes were juxtaposed.

All samples were taken in key areas of the mapped structures and were collected oriented, in order to keep the link between the meso- and microstructural evidence. Samples were carefully studied by optical and (where needed) scanning electron microscopy (SEM) to reconstruct their microstructural and metamorphic evolution. Once a relative chronology was

established, selected growth zones of mineral phases were analysed by electron probe micro-analyser (EPMA) as a basis to perform thermodynamic modelling. P-T data were linked to fabric elements, using textural criteria. Geochronology was performed by LA-ICP-MS, targeting specific growth zones of accessory phases that were separated after noting their microstructural and geochemical context. It turned out to be critical to analyse each growth zone separately to link the age (t) to a specific metamorphic stage (P-T). Observations and P-T-t data derived from each sample are then compared and correlated within the sample series and then related to observations made in the field data and in microscopy. This detailed petrochronological approach (e.g. Engi et al., 2017) is particularly effective when applied to a study area that has been mapped and structurally characterized in detail, as in the present case.

**4 Petrography and mineral chemistry**

**4.1 Methods**

**4.1.1 SEM**

Back-scattered electron images (BSE) were acquired using the Zeiss EVO50 SEM at the Institute of Geological Sciences (University of Bern) using an accelerating voltage of 15 to 25 KeV, a beam current of 500 pA and a working distance of 10 mm. Cathodoluminescence (CL) pictures where obtained with the same operative conditions, but with 10 KeV accelerating voltage and a working distance of 9.5 mm.

**4.1.2 EPMA analyses**

EPMA analyses were performed using a JEOL JXA-8200 superprobe at the Institute of Geological Sciences (University of Bern). Point mode analyses and X-ray compositional maps were acquired both using wavelength dispersive spectrometers (WDS). For X-ray mapping the procedure described in Lanari et al. (2013) was followed. It consists in measuring point mode analyses first and then acquiring X-ray compositional maps on the same area. For point analyses, analytical conditions were 15 KeV accelerating voltage, 10 to 20 nA specimen current, 40 s dwell times (including $2\times10$ s of background measurement) and a beam ø from 1 to 5 µm. Lower current and higher beam size were used for minerals containing Ca, Na and K such as phengite and plagioclase. Nine oxide compositions were measured, using synthetic and natural standards: wollastonite / orthoclase / almandine ($SiO_2$), anorthite / almandine ($Al_2O_3$), anorthite (CaO), almandine (FeO), forsterite / spinel (MgO), orthoclase / phlogopite ($K_2O$), albite ($Na_2O$), ilmenite ($TiO_2$), and tephroite (MnO). For X-ray maps, analytical conditions were 15 KeV accelerating voltage, 100 nA specimen current and dwell times of 150-250 ms. Nine elements (Si, Ti, Al, Fe, Mn, Mg, Na, Ca and K) were measured at the specific wavelength in two passes. Intensity X-ray maps were standardized to concentration maps of oxide weight percentage using spot analyses as internal standard. X-ray maps were processed using XMapTools 2.2.1 (Lanari et al., 2014).

**4.2 Results: Sample description**

Five samples were analysed from the IC and two samples from the EC. These were collected from internal (SE) to external areas (NW) of the Sesia Zone. A brief account is given here, with characteristic images shown in Fig. 2. Supplement S1 contains detailed descriptions and GPS locations.

The samples of the IC (FG1324, FG1315, FG12157, FG1347, and FG1249) are micaschists with eclogite facies assemblages comprising quartz, phengite, garnet, ± paragonite ± glaucophane ± omphacite ± chloritoid, with accessory allanite ± rutile. The main fabric in all of these samples is an eclogite facies foliation (Fig. 2a). Evidence of several deformation stages occurring before or after the main eclogite facies foliation is preserved in several samples as microlithons, commonly of phengite, omphacite, glaucophane or chloritoid oriented at high angles relative to the main foliation, which wraps around them or is overgrown by them (Fig. 2b). Further evidence of several metamorphic stages occurring at eclogite facies conditions is reflected in growth zones of garnet (Giuntoli et al., in review). Pre-Alpine relics include garnet cores (Fig. 2c) and zircon (cores ± first rims; chapters 6.4 and 6.5).

Retrograde stages of blueschist or greenschist facies assemblages related to decompression are locally present in samples. The blueschist facies stage produced pleochroic crossite rims around glaucophane (Fig. 2d). The greenschist facies stage produced symplectites of actinolite ± albite ± chlorite around glaucophane and omphacite, chlorite at the expense of garnet, epidote/clinozoisite rims around allanite, and titanite rims formed around rutile (Fig. 2e-f).

Sample FG1420, collected in the EC, is a garnet orthogneiss that shows a HP greenschist foliation marked by phengite, chlorite, and titanite; the foliation wraps garnet porphyroblasts that preserve a relic internal foliation (Fig. 2g). Permian magmatic relics of pleochroic allanite are surrounded by an Alpine corona of epidote grains (Fig. 2h). Some hundred meters to the north, another sample (FG12107) of the EC was collected. This is a leucogneiss characterized by the same metamorphic fabric and parageneses as the previous sample, except that garnet and magmatic allanite are missing.

**4.3 Results: Growth zones of garnet and phengite**

Garnet and phengite display features in the IC samples that differ from those in the EC samples. To highlight and describe these, mostly two samples are compared in the following paragraphs: FG1249 (IC) and FG1420 (EC). A more complete account of garnet textures and mineral inclusions in the IC is shown in Giuntoli et al. (in review).

In the IC samples, garnet consists of a core followed by several rims with a grain size up to several millimeters (Fig. 3a, b). The compositional map of the grossular end-member fraction ($X_{Grs}$) in sample FG1249 shows a porphyroclastic core ($Alm_{72}Prp_{18}Grs_5Sps_5$; Table 1) with internal fractures sealed by garnet of higher $X_{Grs}$ (Fig. 3b). A first rim (rim1-$Alm_{76}Prp_{15}Grs_9$) overgrows the core and displays higher grossular contents. This rim1 is followed by rim2, which again records an increase of $X_{Grs}$ ($Alm_{62}Prp_{20}Grs_{18}$). Rim2 resorbs parts of rim1 externally, as well as parts of rim1 internally and core. Rim3 is peripheral and shows the highest Ca contents ($Alm_{58}Prp_{19}Grs_{23}$).

Sample FG1315 is characterized by a porphyroclastic core ($Alm_{69}Prp_{28}Grs_4$) with lobate edges and resorption features (details in Giuntoli et al. submitted and Engi et al. in prep.) surrounded by several rim generations: rim1 ($Alm_{61}Prp_{21}Grs_{19}$), rim2 ($Alm_{65}Prp_{24}Grs_{11}$) and rim3 ($Alm_{70}Prp_{24}Grs_6$). Atoll garnets, a few hundred microns in size, are observed in this sample. The shells of the atoll garnet have similar zoning patterns and compositions as the rim generations just described. In sample FG12157 the garnet core ($Alm_{70}Prp_{26}Grs_4$) is rimmed by two growth zones: rim1 ($Alm_{64}Prp_{20}Grs_{16}$) and rim2 ($Alm_{59}Prp_{24}Grs_{17}$). In sample FG1347, the garnet core ($Alm_{69}Prp_{28}Grs_3$) is enclosed by three rims (rim1-$Alm_{66}Prp_{23}Grs_{11}$, rim2- $Alm_{68}Prp_{26}Grs_6$, rim3-$Alm_{70}Prp_{26}Grs_4$). The exception is sample FG1324, in which garnet shows a single growth zone of homogeneous composition ($Alm_{70}Prp_{18}Grs_{11}Sps_1$).

In the EC, sample FG1420 shows garnet with completely different features. As shown in Fig. 3d, the $X_{Sps}$ map highlights concentric zoning (values of $Alm_{54}Prp_2Grs_{36}Sps_8$ for the core, $Alm_{57}Prp_3Grs_{37}Sps_4$ for rim1, $Alm_{60}Prp_3Grs_{35}Sps_2$ for rim2 and $Alm_{63}Prp_4Grs_{32}Sps_1$ for rim3), with no visible resorption features (further compositional end-member maps are shown in Supplement S2).

To link the growth zones of garnet to the main assemblage observed in the mineral matrix, microstructural relations, overprinting criteria and mutual inclusions were employed, based on optical microscopy, SEM, and compositional maps. In particular, garnet in sample FG1249 contains inclusions of paragonite, phengite, and quartz between rim1 and rim2 (Fig. 3a). Rutile inclusions of few microns are present in rim2 and 3. Late chlorite fractures dissect the entire garnet. Garnet in sample FG1420 is wrapped by the main external foliation and includes an internal foliation marked by quartz, epidote, and titanite (Fig. 3c).

Phengite in IC samples displays a uniform composition except along grain boundaries, where lower Si and $X_{Mg}$ contents are found, indicating retrograde overprinting (e.g. Fig. 3e; Group1 Si ~3.36 apfu, $X_{Mg}$ ~0.83; Group2 Si ~3.3 apfu, $X_{Mg}$ ~0.68 in sample FG1249; Table 2).

In the EC, two distinct generations of phengite are distinguished based on their microtextural position: the first one describes the main foliation and is characterized by high Si values (Fig. 3f; Group1 Si ~3.4 apfu, $X_{Mg}$ ~0.61 in sample FG1420). The second phengite generation (Group2 Si ~3.32 apfu, $X_{Mg}$ ~0.61) rims the first one and occurs in fold hinges that deform the main foliation.

**5 Thermobarometry**

**5.1 Methods**

**5.1.1 Whole rock major element compositions**

Major element compositions were analysed by X-ray fluorescence (XRF) spectrometry at the University of Lausanne (Switzerland). Representative quantities of samples were crushed and then pulverized in a tungsten carbide mill. The powder

was dried for two hours at 105°C. Loss of ignition was then determined by weight difference after heating to 1050°C for 3 hours.

**5.1.2 Garnet thermobarometry using GrtMod**

Garnet growth zones were carefully selected after detailed microstructural and compositional analysis of the high-resolution X-ray maps to extract representative compositions that are used as input for modeling. Following the strategy proposed by Lanari et al. (2017), the composition of each growth zone was obtained directly from the quantitative maps by sampling chemically homogeneous areas amongst different garnet grains. Each composition was then assigned to a specific growth zone that is assumed to be uniform in the following. The regularity of the chemical zoning observed in garnet (Giuntoli et al., in review) supports the grain boundary equilibrium model adopted here (Lanari and Engi, 2017). Minor heterogeneities observed in each growth zone (<0.01 in XAlm and XGrs; < 0.005 in XPrp, XSps) may be due to kinetic effects during growth, they do not affect the results of the equilibrium model.

To model the complex garnet textures adequately, fractionation and resorption processes must be taken into account in approximating the evolution of the reactive bulk composition. The latter is strongly affected by fractionation (i.e. the removal of refractive garnet, e.g. Evans, 2004;Konrad-Schmolke et al., 2008) and by resorption, which can shift the reactive bulk composition back toward the garnet composition (Lanari and Engi, 2017). The program GRTMOD (Lanari et al., 2017) was specially developed to deal with samples in which garnet experienced a complex history involving several stages or growth, resorption and/or pseudomorphic replacement. In essence, the program refines the reactive bulk composition used in free energy minimization at each stage predicted if previously grown garnet was preserved or only partly dissolved. GRTMOD uses an iterative approach that refines the P-T conditions for successive garnet growth zones. For each inversion (i.e. a single growth stage), a solution was deemed acceptable if the least squares residual value (the cost function $C_0$ used by Lanari et al., 2017) was <0.05, reflecting a sufficiently close match between the modeled and observed garnet compositions. In the IC samples, resorption and fractionation were constrained according to the volumetric proportion of each growth zone, as estimated from the thin section and the compositional maps. No resorption was permitted in the program for sample FG1420, as garnet textures in the compositional maps show no evidence of this process. To model the rims generation in the IC samples the "go fast mode" function (Lanari et al., 2017) was used, with an initial starting P-T guess of 650 °C and 1.6 GPa. The initial starting guess is an essential technicality used for reasons discussed by Lanari et al. (2017, sections 5.2.3 and 6.3); the specific purpose is to avoid local minima in optimization. This function searches a solution around the starting guess and follows the gradient in the objective function; there cannot be local minima at high pressure for this range of bulk rock composition (see Fig. 8 in Lanari et al., 2017). The MnO component was used in the thermodynamic computations of the relatively rich Mn-garnet in sample FG1420. In sample FG1249, MnO was used to model the garnet core but was ignored in the models of the following rims because the concentration found in garnet is low (< 1 wt% MnO; Table1). In the remaining samples, MnO was ignored (< 1 wt%), and the system considered in modelling was simplified to $SiO_2$-$TiO_2$-

Al$_2$O$_3$-FeO-MgO-CaO-Na$_2$O-K$_2$O-H$_2$O. The thermodynamic database used was the same as to compute the isochemical phase diagrams (see below).

**5.1.3 Isochemical phase diagrams (pseudosections)**

Isochemical equilibrium phase diagrams were computed using the Gibbs free energy minimization algorithm
THERIAK/DOMINO (de Capitani and Petrakakis, 2010;De Capitani and Brown, 1987). The thermodynamic database of
Berman (1988) with subsequent updates collected in JUN92.bs (distributed with Theriak-Domino 03.01.2012; Supplement
S3) was used, together with the following solution models: Berman (1990) for garnet; Fuhrman and Lindsley (1988) for
feldspar; Meyre et al. (1997) for omphacite; Keller et al. (2005) for white mica, and ideal mixing models for amphibole
(Mäder and Berman, 1992;Mäder et al., 1994), epidote, and chlorite (Hunziker, 2003). All Gibbs free energy minimizations
were carried out assuming an excess in pure H$_2$O fluid. The amount of H$_2$O component predicted at high-pressure is in line
with the measured LOI (1.4-2.7 wt-%) in the present-day samples. Note that for the pre-Alpine HT computations no melt
model was used. Fe$^{3+}$ was ignored because of the lack of analytical data and suitable ferric end-members in solid solution
models.

**5.1.4 Chlorite and white mica multi-equilibrium**

To constrain the P-T conditions of retrograde stages, multi-equilibrium computations of the high-variance assemblages
involving chlorite and white mica were carried out, using the standard state properties and solid solution models of Vidal et
al. (2005; 2006) for chlorite, Dubacq et al. (2010) for phengite, and the program CHLMICAEQUI (Lanari, 2012). The activity
of H$_2$O was set to unity. Three methods were successively employed:

(1) *Chlorite+Quartz+H$_2$O thermometry*: The chlorite formation temperature and XFe$^{3+}$ were estimated at a fixed
    pressure of 1 GPa from the combination of four equilibria involving five chlorite end-members, quartz and H$_2$O
    (Lanari et al., 2012; Vidal et al., 2016).

(2) *White-Mica+Quartz+H$_2$O barometry*: A divariant P-T equilibrium line was estimated for each white mica analysis
    (assuming XFe$^{3+}$ = 0) from the convergence of three equilibria involving five phengite end-members, quartz and
    H$_2$O (Dubacq et al., 2010).

(3) *Chlorite+White-Mica+Quartz+H$_2$O thermobarometry*: P and T of formation for each chlorite and white mica
    couple, as well as their respective XFe$^{3+}$, were estimated by minimizing the square root of the sum of $(\Delta G_{reaction})^2$
    for 6 equilibria (see Supplement S4).

For the sake of clarity, only 64 equilibria (excluding the Pyrophyllite·1H$_2$O end-member) are shown in the P-T
diagrams. The starting guess for T and P was taken from the result of Chlorite+Quartz+H$_2$O thermometry and White-
Mica+Quartz+H$_2$O barometry. This multi-equilibrium approach relies on the assumption of local thermodynamic
equilibrium between the selected chlorite and white mica at the P-T conditions of convergence. Chlorite and white mica

couples were chosen where microtextural evidence suggested equilibrium, notably where sharp contacts were observed between these sheet silicates.

**5.2 Bulk rock and reactive bulk composition**

For samples FG1324 and FG1420 the original bulk rock compositions obtained by XRF were used to compute isochemical phase diagrams. In samples FG1315, FG12157, FG1347 and FG1249 however, the unmodified bulk rock composition cannot be used for modeling because a significantly high volume fraction of garnet is present (5 – 10 vol%), including a pre-Alpine core and Alpine rims generations. To compute equilibrium diagrams properly, the reactive bulk rock composition was approximated using the program GRTMOD (see section 5.1.2). Each isochemical phase diagram is thus valid for a single P-T stage only. To select the reactive bulk composition of this specific stage, a link must be established between the particular garnet generation that formed in equilibrium with the mineral phases present in the matrix. We established this link using petrographic observations, including textural equilibrium criteria, compositional zoning (visible in compositional maps) and inclusions relationships. Specifically, we determined that the garnet growth zones that coexisted with the mineral matrix are as follows: garnet rim3 for sample FG1315, rim1 for sample FG12157, rim3 for sample FG1347, and rim2 for sample FG1249. The corresponding reactive bulk compositions used for the modeling are provided in the Supplement S5.

**5.3 Results: Garnet thermobarometry and phase diagrams**

Fig. 4 shows isochemical P-T phase diagrams for each sample. The plots show the results of garnet thermobarometry (GRTMOD), garnet isopleths ($X_{Grs}$, $X_{Alm}$, and $X_{Prp}$ in sample FG1324), and $X_{Mg}$ and Si (in apfu) isopleths for phengite. For sample FG1420, results of chlorite – white mica thermobarometry are also displayed. A summary of mineral compositional data for the main phases, the modelling method used, the XRF analyses of major elements of each sample, and details of the GrtMod results are available as Supplement S5, S6, S7 and S8 respectively. Each sample from the IC and EC is presented separately below.

**5.3.1 IC - FG1324 Omphacite, garnet, glaucophane and rutile micaschist**

[revised manuscript text omitted]

Two phengite generations are presents (chapter 4.3; Fig. 3f): phengite describing the main foliation, with higher silica content, displays Si apfu and $X_{Mg}$ isoplets iersection at ~1.4 GPa, 550 °C. These conditions are not substantiated by the mineral assemblage, which is predicted to contain omphacite and rutile, but neither phase was observed in thin section. Also, at these P-T conditions further garnet growth is predicted, with a modal increase from 6.5 to >7.5 vol%, but is no garnet is observed with a composition compatible with these P-T conditions. We suspect that phengite grew at lower P-T conditions, as the appropriate Si apfu values intersect the P-T results derived from white mica + quartz + $H_2O$ barometry at 0.6-0.8 GPa, 350-400 °C (Fig. 4f; more details in the next chapter).

The second generation of phengite, post-dating the main foliation, shows Si values for which the isopleths intersect with the results derived from white mica + quartz + $H_2O$ barometry at 0.55-0.75 GPa, 300-350 °C.

**5.4 Results: Multi-equilibrium thermobarometry**

**5.4.1 IC - FG1315 Garnet, epidote and rutile quartz-micaschist**

Chlorite in this sample is retrograde and records formation temperatures decreasing from 450 °C to 300 °C (Fig. 5a). White mica + quartz + $H_2O$ barometry suggest pressures comprised between 1.5 and 0.4 GPa for the temperature range of chlorite. Chlorite and white mica grains in textural equilibrium were used to constrain the equilibrium conditions at 0.8 ± 0.2 GPa and 340 ± 50 °C for the retrograde stage (Fig. 5b; Tables 3 and 4).

**5.4.2 IC - FG12157 Garnet, glaucophane, epidote and rutile micaschist**

Chlorite records formation temperatures decreasing from 430 °C to 310 °C (Fig. 5c). White mica + quartz + $H_2O$ barometry finds pressures between 0.02 and 1 GPa, for the temperature range shown by chlorite. Chlorite and white mica grains in textural equilibrium are used to approximate equilibrium conditions at 0.54 ± 0.2 GPa and 394 ± 50 °C for the retrograde stage (Fig. 5d).

**5.4.3 IC - FG1347 Chloritoid, garnet and rutile micaschist**

Chlorite registers temperatures from 370 °C to 250 °C (Fig. 5e). White mica + quartz + $H_2O$ barometry indicates pressures between 0.02 and 1 GPa for the temperature range of chlorite. Chlorite and white mica grains record 0.78 ± 0.2 GPa and 341 ± 50 °C for the retrograde stage (Fig. 5f).

**5.4.4 EC - FG1420 Garnet orthogneiss**

The chlorite and white mica multi-equilibrium technique was used to constrain the equilibrium conditions of three successive stages of retrogression using couples linked to different microstructural positions that developed after the main foliation. These show equilibrium conditions at $0.87 \pm 0.2$ GPa, $354 \pm 50$ °C in fold hinges; in pressure shadows at $0.6 \pm 0.2$ GPa, $331 \pm 50$ °C; and in static overgrowths over the main foliation at $0.42 \pm 0.2$ GPa, $231 \pm 50$ °C (Fig. 6).

**6 Texture and geochronology of allanite and zircon**

**6.1 Methods**

**6.1.1 Allanite geochronology**

[revised manuscript text omitted]

micas have the same composition as those defining the eclogite foliation (Fig. 7b, c, d). In the case of FG1249 and FG1347, allanite also shows intergrowths with both white micas (Fig. 7c).

In the EC, allanite is rare and, where present, is typically magmatic; it appears dark brown and pleochroic in the optical microscope, with a grain size of some millimetres (Fig. 2h; Giuntoli and Engi, 2016). In only two samples metamorphic allanite was found, preserved in the core of epidote crystals (FG1420, FG12107). The metamorphic allanite has a grain size less than 50 µm, is colourless or pale yellow in polarized light, with low interference colour and undulose extinction in crossed polarized light. Sample FG1420 shows both magmatic and metamorphic allanites (Fig. 7g, h). Allanite includes paragonite, with phengite and titanite occurring both in the epidote rim and at the allanite-epidote boundary (Fig. 7g). Very few tiny monazite grains (few µm) are found in the core of metamorphic allanite. Sample FG12107 also shows similar epidote crystals preserving metamorphic allanite in their core, as in sample FG1420. Magmatic allanite preserved in sample FG1420 occurs as mm-size grains that are fractured and appear much brighter in BSE pictures than metamorphic allanite (Figs. 2h, 7h). Epidote crystals form satellites around magmatic allanite, suggesting partial breakdown (Fig. 7h). Note that these epidote crystals retain a BSE-bright core of newly grown (Alpine) metamorphic allanite.

**6.3 Results: Allanite U-Th-Pb dating**

For both IC and EC, only allanite cores were successfully dated. The rims showed too high common lead (Pb$_c$) contents, its correction (Gregory et al., 2007;Burn et al., 2017) would lead to large uncertainties in the age calculation.

In the IC and EC, the Tera-Wasserburg and $^{232}$Th/$^{206}$Pb$_c$ -$^{208}$Pb/$^{206}$Pb$_c$ isochron diagrams are concordant, within the range of uncertainty, in all the analyzed samples (Figs. 8. 9; every ellipse is a single spot measurement). Ages range between 77 and 56 Ma for the allanite cores of the IC. In the EC, the magmatic allanite cores in sample FG1420 yield ages of ~290 Ma, the metamorphic cores of 73.7±8.2 Ma. In sample FG12107, the metamorphic allanite cores were dated to 62.8±3.3 Ma. A summary of the Alpine allanite age data from each sample is listed in Table 5.

In detail, the IC Tera-Wasserburg diagrams show $^{207}$Pb/$^{206}$Pb y-intercepts of 0.84 ± 0.01, 0.823 ± 0.018 and 0.83 ± 0.004 for samples FG1324, FG1315 and FG12157 respectively, with MSWD on the regression comprised between 1.2 and 2.1 (Fig. 8). $^{206}$Pb$_c$-isochron diagrams display a $^{208}$Pb/$^{206}$Pb y-intercept of 2.081 ± 0.027, 2.077 ± 0.066 and 2.085 ± 0.028 for samples FG1324, FG1315 and FG12157 respectively, with MSWD on the regression comprised between 0.46 and 0.98. These values are close to the predicted values of Stacey and Kramers (1975) for model lead evolution of this time range (Fig.1 in Burn, 2016). The exception is sample FG1347, in which the Tera-Wasserburg diagram shows a $^{207}$Pb/$^{206}$Pb y-intercept at 0.787 ± 0.04 (MSWD on the regression of 2.5) and displays a $^{208}$Pb/$^{206}$Pb y-intercept on the $^{206}$Pb$_c$-isochron diagram at 1.98 ± 0.082 (MSWD on the regression of 0.4). These values deviate from the values predicted by Stacey and Kramers (1975), probably indicating local inheritance.

In the EC, data for magmatic allanite in sample FG1420 shows $^{207}$Pb/$^{206}$Pb y-intercept at 0.85 ± 0.11 in a Tera-Wasserburg diagram and displays a $^{208}$Pb/$^{206}$Pb y-intercept at 2.09 ± 2.7 in a $^{206}$Pb$_c$-isochron diagram (Fig. 9). These large uncertainties reflect few (8) spots analyses. Metamorphic allanites show $^{207}$Pb/$^{206}$Pb y-intercepts of 0.825 ± 0.01 and 0.811 ± 0.014 in

Tera-Wasserburg diagrams (MSWD on the regression of 0.7 and 1.2) and displays $^{208}$Pb/$^{206}$Pb y-intercepts of 2.061 ± 0.029 and 2.03 ± 0.031 in $^{206}$Pb$_c$-isochron diagrams (MSWD on the regression of 0.4) for sample FG1420 and FG12107, respectively. These values are close to the predicted values of Stacey and Kramers (1975) for these time ranges.

**6.4 Results: Zircon textures**

5   Internal textures of zircon from the IC reveal complex zoning in CL-images (Fig. 10), showing detrital cores and several phases of resorption and (metamorphic) overgrowth. Zircon cores commonly preserve a variety of internal textures, most commonly oscillatory zoning (Fig. 10e, f) which is typical of zircon grown form melt. Many cores are affected by resorption, obliterating earlier features, but in some grains show sharp boundaries between core and rims, and these preserve features of sediment transport such as broken, rounded or pitted surfaces (Fig. 10b). The number of rims varies between and within

10   samples, from two rims in sample FG1257 to a maximum of five in FG1315. Most zircon grains show a first metamorphic rim with a light grey to bright CL-response, followed by a rim with dark CL-appearance. The third rim typically is again medium grey to bright white in CL. In sample FG1315, a forth (dark CL) and fifth (light grey) rim follow, and FG1347 occasionally shows a CL-dark forth rim. The internal textures of the different rims are not always clearly distinguishable, either because of limited width or, in case of very dark or bright CL-response, limited contrast. The second metamorphic rim

15   in FG12157 is either uniform or shows fir-tree zoning (Vavra et al., 1996;Root et al., 2004), in sample FG1347 metamorphic rims are often too thin to distinguish or no texture is recognizable in the bright-CL third rim. The first two rims in sample FG1315 have cloudy textures, the third rim with the bright-CL shows no further internal textures but sometimes has inclusions and the two outermost rims commonly are uniform or cloudy in texture. The innermost and outermost metamorphic rims in sample FG1249 and FG1324 are bright and featureless, the second dark rim in sample FG1249 shows

20   sector zoning and many inclusions. The third (medium grey) rim in FG1249 has a cloudy texture.
Zircon in the External Complex displays pre-Alpine magmatic oscillatory zoning; rarely a bright CL rim, 5-10 µm in width, surrounds the core.

**6.5 Results: Zircon U-Pb dating**

The range of Alpine $^{206}$Pb/$^{238}$U zircon dates for each sample is summarized in Table 5. Details on the pre-Alpine dates are

25   available in Kunz et al. (2017). Supplement S10 gives an overview of pre-Alpine dates as well as individual analyses of Alpine dates. Concordia plots of individual spots analyses of Alpine zircon are available in Supplement S11. For the five samples we obtained 27 analyses of detrital cores, 53 of Permian rims, and 34 of Alpine rims. Detrital zircon cores in samples from the IC range from ~793 to 353 Ma and partially overlap with the first rim in sample FG1324 and FG1249 (Fig. 10), for which the dates range from ~450–420 Ma. In most samples, at least two rims are found yielding Carboniferous to

30   Triassic dates from ~313 to 222 Ma, as discussed in detail by Kunz et al. (2017); only one sample (FG1324) shows none of these. Alpine rims, with a range between 77 and 56 Ma, were measured in all samples except FG1249, where the third rim was too thin (10µm) to be analysed. No correlation was found between rim types and ages within samples or amongst them,

and it was not possible to make absolute age distinctions among Alpine rim generations. This is due to relatively large uncertainty in the ages, as the narrow rims only allowed small spots for LA-ICP-MS measurements, decreasing their analytical precision.

Th/U ratios in detrital zircon cores range between 0.15 and >3, the rim generation with dates between 450–420 Ma have low Th/U ratios in sample FG1324 (0.006–0.14) whereas those in sample FG 1249 are between 0.1 and 0.5. The Carboniferous to Triassic rims show Th/U ratios between 0.002– 0.4. The Alpine metamorphic rims show low Th/U ratios between 0.001– 0.01.

**7 Discussion**

**7.1 Linking time constraints with equilibrium conditions (Petrochronology)**

Preliminary analysis of allanite for U-Th-Pb by LA-ICP-MS was performed *in situ* (i.e. using LA-ICPMS spot analyses), on polished thin sections. However, to obtain more material for dating, allanite grains were subsequently separated, mounted on resin mounts and polished. The main reasons why allanite grains were separated for dating are the following:

- Grain mounts are more efficient as they enable selecting more spots to analyze a single growth zone (at least one order of magnitude more spot analyses per sample than in thin sections).

- Grains mounts produce more likely equatorial cuts because polishing is optimized to obtain the largest sections possible at a given grain size. This is essential for LA-ICP-MS analysis of grains with several growth zones: if sectioning is not near-equatorial, one is more likely to drill through different growth zones.

We do consider petrographic control before dating to be essential: all mounts were imaged at the SEM with the BSE detector to document allanite textures. For each sample, both in the thin section and in the mounts, the allanite grains displayed the same textural patterns, growth zones and minerals inclusions at the sample scale (in separates and thin section). The microtextural features observed in thin sections were readily linked to those seen in grain mounts (compare Figs. 7a and 7f). Furthermore, we note that U-Th-Pb ages from grain mounts analyses agree within uncertainty with the spot analyses made in thin sections.

The fundamental task to reconstruct the P-T-t paths is to establish a strong link between P-T conditions, deriving from thermodynamic modelling, and t data, deriving from age dating. Several criteria were used to achieve this goal, notably: (i) textural evidence, (ii) the presence and composition of distinctive minerals as inclusions, and (iii) the presence of intergrowths of allanite with other phases.

As described in section 6.2, allanite grains of the IC are intergrown with the main mineral phases that describe the eclogite facies foliation (Fig. 7; Table 5). Furthermore, the compositions of mineral inclusions in allanite matched to those found in the matrix and to those predicted by thermodynamic modeling (representative chemical analyses in Supplement S5). Based on these observations (and details given in section 6.2 for each sample), we infer growth of allanite coeval with the minerals marking the eclogite facies foliation in all of the samples analyzed from the IC. This link between age data and P-T

conditions is shown by the red ellipses on the equilibrium phase diagrams shown in Fig. 4 and on the P-T-t paths shown in Fig. 11b and c (colour-coded for each sample). Additionally, in four samples of the IC, the ages obtained for allanite cores and Alpine zircon rims overlap within uncertainty, suggesting coeval growth of these two accessory minerals.

In the EC, metamorphic allanite is rare and preserved only in the core of epidote (section 6.2; Fig. 7 g, h). Epidote is intergrown with white mica, chlorite and titanite, marking the retrograde greenschist overprint, which is constrained by Chlorite+White-Mica+Quartz+$H_2O$ thermobarometry (~0.9 GPa, 350 °C to 0.4 GPa, 230 °C, see Fig. 6). According to these results, metamorphic allanite growth in the EC predates the retrograde greenschist overprint (red ellipse in Fig. 4f and purple ellipse in Fig. 11b-c).

**7.2 P-T-t paths of samples**

**7.2.1 Pre-Alpine conditions in the IC**

The history of the studied samples is summarized in P-T-t diagrams (Fig. 11) and compared with relevant data from the literature. The first metamorphic stage recorded in our samples from the IC is of Permian age, it reflects granulite facies conditions. Porphyroclastic garnet cores (present in samples FG1315, FG12157, FG1347 and FG1249) are the only major mineral relics of this HT stage. Among accessory minerals, late Carboniferous to Upper Triassic zircon rims (between 313 and 222 Ma) are preserved, and monazite relics in allanite most probably are Paleozoic as well. The ages of the Permian zircon cores fit well with those reflecting HT metamorphism in the Ivrea Zone and Adria-derived units in the Western Alps (Vavra et al., 1999;Vavra et al., 1996;Kunz et al., 2017;Ewing et al., 2013), thus supporting the long held view that the Sesia Zone is closely related to the Ivrea Zone (e.g. Compagnoni et al., 1977). The single Upper Triassic date (222 ± 13 Ma) in sample FG1315 is similar to ages reported from the Ivrea Zone by Vavra et al. (1999), where they have been related to fluid alteration. The P-T conditions for the pre-Alpine HT stage are constrained between 0.6-0.8 GPa and 700-900 °C from the garnet cores of the IC samples. These results confirm earlier data for the IC (0.6-0.9 GPa, ~850 °C; Lardeaux and Spalla, 1991;Rebay and Spalla, 2001; Fig. 11). A retrograde pre-Alpine path through amphibolite facies conditions is evident in sample FG1249, where a first garnet rim generation yields ~0.6 GPa and 600 °C (Fig. 11b), again in fair agreement with previous data (0.3-0.5 GPa, 570-670 °C; Lardeaux and Spalla, 1991;Rebay and Spalla, 2001). A sketch summarizing the chronology and thermodynamic modelling of garnet in sample FG1249 is shown in Giuntoli et al. (in review).

**7.2.2 Pre-Alpine conditions in the EC**

In the EC, sample FG1420 shows pre-Alpine ages of ~290 Ma for the magmatic allanite grains. These age data probably reflect magmatic crystallization ages of the granitoids; they are in good agreement with several age data of magmatic zircon and monazite for the Sesia-Dent Blanche nappe (Fig. 1 in Bussy et al., 1998; Table 2 and Fig. 5 in Kunz et al., 2017).

**7.2.3 Alpine conditions in the IC**

For the Alpine history, our samples from the IC indicate higher P and T than in the EC, based on detailed garnet compositional modelling (Lanari et al. 2017). The IC recorded a range of maximum pressures between 1.6 and 2 GPa at temperatures of 600-670 °C. Age constraints based on allanite dating of five samples from the IC fall into three groups:

- Group 1 with ages of ~73 Ma (from 77.2 ± 7.3 Ma to 72.4 ± 1.4 Ma, based on samples FG1324 and FG1249 respectively)

- Group 2 at ~65 Ma (from 65.4 ± 3.5Ma to 64.5 ± 4.3Ma, based on samples FG1315 and FG12157)

- Group 3 at 55.7 ± 4.5Ma (sample FG1347).

Metamorphic conditions for Group 1 show 1.6-1.75 GPa and 580-650 °C for the earliest stage. In Group 2, samples FG1315 and FG12157 yield not only the same age, within analytical uncertainty, but also similar metamorphic conditions; the pressure difference of 0.1-0.2 GPa is within the uncertainty of the model, as indicated by error bars in Fig. 11b. Notably, the same P-T conditions were derived from garnet rim1 of FG12157 in Group 2 and from garnet rim3 of sample FG1249 in Group 1. Summarizing, Groups 1 and 2 experienced similar P-T conditions but Group 2 attained these some 5-10 Ma later. Furthermore, the outermost garnet rim in sample FG12157 preserves evidence of a retrograde stage at 1.4 GPa and 650 °C. Sample FG1347 (Group 3) shows the youngest allanite ages for a similar pressure range as Groups 1 and 2, but lower temperature conditions (~580-600°C). It thus appears that the samples from the IC reflect several stages of allanite growth, probably because rocks of slightly different bulk composition produced allanite by different metamorphic reactions (Engi, 2017). Three growth stages captured in our samples are at ~73 Ma, ~65 Ma, and ~56 Ma.  The different P-T-t paths of Groups 1, 2 and 3 are interpreted to represent different tectonic sheets (Giuntoli and Engi, 2016) that experienced similar P-T conditions but at different times (further discussed in section 7.3).

Regis et al. (2014; Druer slice) and Konrad-Schmolke and Halama (2014) suggested similar P-T paths for different parts of the IC (Fig. 11c), with the highest pressure reached at ~85 Ma. Halama et al. (2014) constrained the development of the retrograde blueschist facies Tallorno Shear Zone (Konrad-Schmolke et al., 2011) at 65.0 ± 3.0 Ma using Ar-Ar data on phengite. This shear zone was related to external fluid influx occurring approximately at 1.35 GPa and 550 °C (Konrad-Schmolke and Halama, 2014). These two P-T paths are similar to Group 3 of the present study, but the latter are up to 30 Ma younger. Groups 1 and 2 consistently display similar P-T paths, but temperatures are 50 – 100°C higher than those determined by Regis et al. (2014) and Konrad-Schmolke and Halama (2014). In our samples from the IC, we detected no evidence of pressure cycling such as documented by Rubatto et al. (2011) further south, in the Fondo slice (Regis et al., 2014).

**7.2.4 Alpine conditions in the EC**

In sample FG1420, metamorphic allanites yield ages of 73.7 ± 8.2 Ma associated with metamorphic conditions of 0.8 GPa and 350 - 500°C, a stage constrained by the growth of garnet and phengite, the mica marking the main foliation. Sample FG12107 yields an allanite growth age of 62.8 ± 3.3 Ma (Fig. 11c highlighted by the star). Chlorite and white mica data from
5 this sample give thermobarometric results in agreement with FG1420. The proximity of the two samples in the field, their similarity in P-T conditions and in textural features of allanite lead us to suggest that allanite grew at the same metamorphic stage in these two samples. Their (nominal) ages seem discrepant, but considering the overlapping (2σ) uncertainties, we view them as a single age group (Group 4: ~63±3 Ma).

**7.2.5 Comparison of P-T-t data for the IC and EC**

10 Comparing P-T-t data for the IC and EC, we note that Group 2 (in the IC) and Group 4 (in the EC) recorded the same age data of ca. 65 Ma, but very different metamorphic conditions. This implies that the IC and EC at that time were at completely different structural position in the subduction zone: A difference of 1 GPa translates to a ~30 km vertical offset between the two complexes, assuming lithostatic pressures. Tectonic overpressure cannot be ruled out, but it cannot explain the observed difference in pressures, since Group 4 (at 0.8 GPa) also displays ~150 °C lower temperatures compared to
15 Group 2 (at 1.6-1.8 GPa). We conclude that the IC resided in a substantially deeper and hotter part of the subduction system than the EC ~65 Ma ago.

The retrograde P-T trajectories in the IC and EC are similar, based on our data for chlorite – white mica equilibria: ~0.8 GPa and 340 °C for samples FG1315, FG1347 (IC) and sample FG1420 (EC; Fig. 11b). Sample FG12157 recorded a low-pressure stage at slightly higher temperature (0.54 GPa and 393 °C) compared to the two other samples of the IC. The EC
20 records two further retrograde stages at 0.60 GPa, 330 °C and at 0.42 GPa, 220°C. However, these retrograde stages have yet to be precisely dated. Inger et al. (1996) and Reddy et al. (1999) constrained the greenschist facies overprint of the EC between 46 and 38 Ma (Rb-Sr using phengite). In the light of our data, we interpret these ages to reflect the greenschist facies conditions documented in both complexes (constrained between ~0.8 GPa and 400 °C to 0.4 GPa and 300 °C). The allanite core ages in Group 4 would then be related to an earlier deformation event, as the epidote rim marks the greenschist
25 overprint (Figs. 7g, 11c). Zircon fission track ages for the Sesia Zone range from 42 to 26 Ma (Hurford and Hunziker, 1985;Hurford et al., 1989;Hurford et al., 1991;Berger et al., 2012;Wagner and Reimer, 1972;Kapferer, 2010), suggesting that the terrain crossed the ~250°C isotherm in this age range during exhumation. For the most internal section of the Sesia terrain, the overlying Biella Volcanic Suite dated at 32.5 Ma (Kapferer et al., 2012;Kapferer, 2010) provide an age constraint for the final exhumation to the surface.

**7.3 Tectonic assembly and exhumation of the Sesia Zone**

The IC shows several tectonic sheets, from several hundred meters to a few kilometres in thickness (Giuntoli and Engi, 2016), some of which moved independently in the subduction channel (Rubatto et al., 2011;Regis et al., 2014) at some stages of the evolution. Some of the samples we studied, while taken at most a few kilometres apart in the field, recorded similar P-T paths at different times, as reflected by the three age groups identified (Fig 1b-c). This age difference may reflect relative tectonic mobility at eclogite facies conditions between such sheets of basement-derived rocks, which are notoriously difficult to delimit in this terrain (Giuntoli and Engi, 2016). In addition to tectonic mobility, the combined effects of deformation and repeated hydration must be considered as triggers of recrystallization that promoted chemical equilibration to eclogite facies assemblages (including allanite and zircon, as discussed below).

The differences in the P-T-conditions (at eclogite facies) documented above for the IC samples do not indicate trajectories as tortuous as those suggested by numerical models for ablative subduction (e.g. Stöckhert and Gerya, 2005;Roda et al., 2012). Our P-T-t data as well as the nature and geometry of tectonic boundaries mapped by Giuntoli and Engi (2016) rather indicate progressive accretion at depths of 50-60 km, probably during final subduction-early exhumation stages, juxtaposing a series of continental sheets and forming a coherent complex (the IC Complex; Vitale Brovarone et al., 2013;Regis et al., 2014). Eclogite facies conditions in the IC evidently prevailed for an extended period, at least ~15 Ma.

In the IC, the main deformation stages and mineral reactions were related to pulses of external fluid passing through the rocks, as reported by Engi et al. (in review) and Giuntoli et al. (in review). Repeated fluid influx occurred at eclogite facies conditions, as shown by resorption and growth features in garnet and zircon, with hydrous fluid pervasively rehydrating rocks that had previously been almost completely dehydrated by Upper Paleozoic metamorphism (Engi et al., in review). Thus Permian kinzigites (granulite facies metapelites) were transformed back to micaschists during Alpine subduction. As allanite and zircon ages from these samples are identical within analytical uncertainties, it appears that this fluid influx also triggered the coeval growth of accessory phases at eclogite facies conditions. Later and more localized fluid influx has also been documented at blueschist facies conditions (Konrad-Schmolke et al., 2011), and this probably continued to greenschist facies conditions, as reflected by the partial overprint of the main eclogite assemblages, which is frequently observed in the IC. Our samples in the IC indicate higher temperatures than have been previously reported (Fig. 11). This discrepancy may reflect a regional thermal gradient, simply higher temperatures in more external areas (NW) of the IC, but we suspect it to be a direct effect of the (re)hydration of pre-Alpine HT rocks, which is an exothermic process (e.g. Walther and Orville, 1982;Peacock, 1987;Lyubetskaya and Ague, 2009). Walther and Orville (1982) analysed the thermal effects of (de-)hydration reactions during regional metamorphism and found them to be substantial. When applied to the present situation of (re-)hydration, heat capacity data indicate that heating the HT Permian protolith requires ~1.0 kJ/K per kg of leucogneiss. The enthalpy released upon partial hydration of this protolith (producing the water content typical of these micaschists, 1.5 wt-% $H_2O$) adds ~77 kJ/kg in enthalpy. Such hydration should thus result in a temperature increase of some 80 °. This estimate lends credibility to the P-T estimates obtained here, which indeed are 20-80 °C higher than some maximum

temperatures recently reported from other parts of the Sesia Zone, e.g. 575°C (Konrad-Schmolke and Halama, 2014), 570-630°C for the Druer Slice (Regis et al., 2014), or >600°C for the Ivozio Complex (Zucali and Spalla, 2011). In addition, Zr-in-rutile data reported by Kunz et al. (2017, their Table 3) gave 640 °C for one of the present samples (FG1249), confirming our results by an entirely independent method.

A counterclockwise P-T path is proposed here for the EC. This trajectory is based on the results of garnet and chlorite – white mica modeling. The path as shown in Fig. 11 implies that, at nearly isobaric conditions, this area of the EC experienced cooling by 100-150 degrees, possibly related to the entry of cold material into the subduction channel, as already proposed by Pognante (1989) for the southern Sesia Zone. Piemonte-Liguria oceanic units are a likely source of such material, which would be in line with age data of 58-40 Ma for the HP metamorphism in the Zermatt-Saas Zone (e.g. Rubatto et al., 1998;de Meyer et al., 2014;Weber et al., 2015) and with tectonic-kinematic models for the evolution of the Sesia–Dent Blanche nappes (Manzotti et al., 2014;Angiboust et al., 2014).

Summarizing, the major differences between the eclogite facies conditions recorded in the IC and the epidote blueschist facies condition in the EC are now quantitatively constrained by P-T-t data presented in this study (Fig. 11). Tectonic juxtaposition of these two complexes appears to have occurred during exhumation probably at 0.7-0.9 GPa and 350-400 °C, since these are the first joint P-T conditions found in both complexes, around 46-38 Ma (age data from Inger et al., 1996 and Reddy et al., 1999). These data confirm the proposition by Williams and Compagnoni (1983) of a first order tectonic contact (Barmet Shear Zone in Giuntoli and Engi, 2016; Fig. 1b-c) between the two complexes. Our data are the first to quantify the P-T discontinuity at that contact, i.e. ~1 GPa and 100-180 °C. This age range is younger compared to age range proposed in the tectonic model by Angiboust et al. (2014) for the juxtaposition of the two complexes (70-57 Ma, based on data of Babist et al., 2006, Konrad-Schmolke et al., 2011, and Halama et al., 2014). Whereas Angiboust et al. (2014) considered the Tallorno-Chiusella Shear Zone (Babist et al., 2006;Konrad-Schmolke et al., 2011) to represent the contact between the IC and EC, more recent mapping (Giuntoli and Engi, 2016) shows that this contact is located several kilometres further NW, in the Barmet Shear Zone.

The data shown in Fig. 11 allow us to derive average exhumation rates for the IC (Table 6). Using Group 1 and Group 3 as the two anchors, i.e. the oldest and the youngest groups, a first calculation considers Stage1, the interval from the highest pressures recorded in these two groups (~1.6 GPa at ~73 Ma and ~2 GPa at ~55 Ma, respectively) to the greenschist conditions upon juxtaposition with the EC (~0.6 GPa at ~38 Ma, age from Inger et al., 1996). The mean vertical exhumation velocity obtained ranges from 0.9 to 2.7 mm/year respectively (Table 6). If we take the juxtaposition to be at ~46 Ma (Reddy et al., 1999) for the same P conditions, values for the mean vertical exhumation velocity increase to 1.2 and 5.1 mm/year, respectively. Plate convergence between Adria and Europe was ~15 mm/year in the time span of 68-38 Ma (Handy et al., 2010). In the above calculations we considered subduction angles of 90°, 60° and 45°, a thickness of 20 km for the upper crust, 10 km for the lower crust; their densities were taken as 2.7 and 3.0, respectively, and 3.3 g/cm$^3$ for the upper mantle.

For Stage2, the final stage of exhumation of the IC and EC, we obtained mean vertical exhumation velocities using the juxtaposition conditions (~0.6 GPa) and ages quoted for the end of Stage1, and 32.5 Ma for the arrival at the surface. That

age of the Biella Volcanic Suite (Kapferer, 2010;Kapferer et al., 2012) is supported by zircon fission track ages (Berger et al., 2012). Depending on the time of juxtaposition adopted, at 38 or 46 Ma (Table 6), a mean vertical exhumation velocity of 4 or 1.6 mm/year results. For Stage2, only the vertical exhumation velocity was computed, and an average density of 2.7 g/cm$^3$ was assumed. The rate of convergence in the 35-20 Ma period was ~13 mm/year (Handy et al., 2010). Data for Stage2 are taken as an approximate range of exhumation rates, since the last part of the exhumation was not uniform throughout the Sesia Zone; owing to brittle structures local differences may be important (e.g. Berger et al., 2012;Malusà et al., 2006). Furthermore, the values given are minimum exhumation velocities, as the emplacement of the Biella Volcanic Suite may have postdated the arrival of the Sesia Zone at the surface.

The Stage1 exhumation rates we obtained for the IC agree with exhumation velocities proposed for the same complex by Zucali et al. (2002), Regis et al. (2014) and the mean exhumation rates for the Sesia Zone proposed by Roda et al. (2012). However, our rates are up to an order of magnitude lower than those proposed by Rubatto et al. (2011) and the maximum values of Roda et al. (2012). Mean exhumation velocities for the final exhumation stage of the Sesia Zone (Stage2) are in the same range (1.6-4 mm/year) as those for Stage1 (0.9-5.1 mm/year). Our data show no decrease in exhumation velocity from early exhumation (i.e. mantle to deep crustal positions) to late stages (i.e. crustal levels to the surface), such as has been proposed by Rubatto and Hermann (2001) for ultra-high pressure terranes.

**7.4 Implication for the subduction, assembly and exhumation of continental fragments**

Our work highlights that subducted continental terranes can be composed of several complexes that experienced major differences in their subduction histories (i.e. ~1 GPa in pressure and 100-180 °C in temperature for the Sesia Zone). These complexes can have been tectonically juxtaposed during late metamorphic stages (i.e. greenschist facies conditions) before being jointly exhumed to the surface. Complexes can be lithologically heterogeneous and may comprise several tectono-metamorphic subunits (from a few hundred metres to several kilometres in thickness). These may experience similar P-T paths but at different times (5-10 Ma apart). Differences in the P-T-t trajectories would thus reflect different tectonic sheets and attest to tectonic mobility in the subduction zone and/or several stages of internal deformation plus hydration at eclogite facies that triggered a pervasive HP-fabric and -assemblage (including datable accessory minerals).

To unravel such complex histories in subducted continental terranes, carefully established field relations, followed by microstructural and petrochronological analysis need to be combined to

- map and identify the major and secondary tectono-metamorphic contacts;
- characterize the different fabrics and the mineral phases defining them;
- quantify the differences existing in P, T and t for each complex or subunit.

Finally, the heterogeneity of subducted continental terranes highlighted by this study should be considered when comparing results to numerical and analogue models that aim to investigate the mechanisms responsible for the subduction and exhumation of the continental crust.

**8 Conclusions**

The present paper provides P-T-t data for the central the Sesia Zone, documenting in particular when and at what conditions the two main complexes of this HP-terrain were juxtaposed during the Alpine orogenic cycle.

In particular our data indicate that:

- In the Internal Complex the main stages of mineral growth, and probably the attendant deformation are related to eclogite facies condition occurring between 77 and 55 Ma and triggered by repeated influx of external fluid. It is during this subduction phase that most of the Permian HT-assemblages were replaced, leaving only sparse relics of essentially dry granulites.

- The Internal Complex encompasses three groups of the samples studied, probably reflecting different tectonic sheets. These experienced similar internal deformation and P-T paths, but at different times, reflecting minor (km-scale) adjustments in the subduction channel at 50-60 km depth and were juxtaposed probably during final subduction-early exhumation stages.

- Comparing the two main complexes, diverse metamorphic evolutions emerge between 77 and 55 Ma, with conditions of 1.6-2 GPa and 600-670 °C in the Internal Complex, whereas only 0.7-0.9 GPa and ~500°C appear to have been reached in the External Complex.

- The two complexes were juxtaposed between 46 and 38 Ma (Inger et al., 1996;Reddy et al., 1999) at ~0.8GPa and 350°C, so at mid-crustal levels during exhumation.

- The different tectonic sheets of the Internal Complex were initially exhumed with mean vertical velocities of 0.9 – 5.1 mm/year during Stage1, from the highest pressures recorded to the greenschist facies conditions attained upon juxtaposition with the External Complex.

- During Stage2, the final exhumation of the entire Sesia Zone occurred with mean vertical velocities between 1.6 and 4 mm/year, but the late stages of exhumation in the area can display local differences (Malusà et al., 2006;Berger et al., 2012).

This case study shows that subducted HP continental terranes can be composed of various complexes, characterized by major differences in their tectonometamorphic histories, juxtaposed during exhumation by tectonic contacts operating at greenshist facies conditions. Such complexes include several subunits, which in the present case were some hundred metres to few kilometres in thickness, that acted as fragments in the subduction zone, recording slightly different P-T-t paths. The development of the main fabrics at eclogite facies conditions appears to have been triggered by external fluid influx. These processes could be dated based on the coeval growth of allanite and zircon. Petrochronology proved to be a powerful tool to quantify processes and unravel the metamorphic evolution in a complex geological setting, essentially because the detailed analytical work at the microscale could be linked to solid field evidence.

**Data availability**

Original data underlying the material presented are available by contacting the authors.

**Competing interests.**

The authors declare that they have no conflict of interest.

5  **Acknowledgments**

This work was financially supported by the Swiss National Science Foundation (Project 200020-146175). We thank Roberto Compagnoni, Daniele Regis, and Jörg Hermann for fruitful discussions. We acknowledge constructive comments and suggestions from Samuel Angiboust and an anonymous referee, and we are grateful to Patrice Rey for editorial handling.

[Figure]

Figure 1: (a) Simplified tectonic map of the Western Alps (modified from Manzotti et al., 2014). (b) Tectonic sketch of the study area (modified from Giuntoli and Engi, 2016) with sample locations and P-T-t data (this study). (c) Cross section through the study area (location show in a) with projection of the studied samples. Foliation traces: violet indicates the eclogite facies foliation (S2) of the IC, dark green indicates the composite epidote blueschist-HP greenschist facies foliation (S2+S3) of the EC, dark green indicates the greenschist facies mylonitic foliation (S4) at the contact IC-EC; BSZ Barmet Shear Zone, PLO Piemonte-Liguria Oceanic unit (modified from Giuntoli and Engi, 2016).

[Figure]

**Figure 2: Thin section photos illustrating the metamorphic evolution of the studied samples. End-member mineral abbreviations used throughout text and figures are from Whitney and Evans (2010). (a) Eclogite foliation ($S_{ecl}$) marked by subparallel phengite, omphacite and glaucophane (plane-polarized light). (b) Eclogite foliation marked by the preferred orientation of chloritoid and phengite. Note the chloritoid crystal (left) is oriented perpendicular to $S_{ecl}$ and overgrows it (plane-polarized light). (c) Garnet crystal with a bright pre-Alpine core and darker Alpine rims visible due to fine rutile inclusions (plane-polarized light). (d) Glaucophane crystal with a core displaying pale blue absorption colour, rimmed by darker blue crossite that marks a retrograde blueschist stage (plane-polarized light). (e) Colorless to pale green amphibole growing at the expense of omphacite during a retrograde greenschist stage (plane-polarized light). (f) Open folds with chlorite crystallizing in the hinge zone, marking a retrograde greenschist stage (plane-polarized light). (g) Garnet porphyroclast wrapped by the main greenschist foliation ($S_{gr}$); note the inner foliation inside the garnet ($S_{int}$; crossed-polarized light). (h) Brown magmatic allanite surrounded by finer grain epidote crystals (plane-polarized light).**

[Figure]

**Figure 3: (a) Mineral phases in sample FG1249 (compare with Fig. 2c) . (b) $X_{Grs}$ map highlights the porphyroclastic cores showing fractures sealed by higher $X_{Grs}$ garnet and three rims (Further garnet end-members maps for the samples are available in Giuntoli et al. submitted). (c) Mineral phases in sample FG1420 (compare with Fig. 2g). (d) Garnet displaying concentric zoning related to a decrease in $X_{Sps}$. (Further garnet end-members maps for the samples are available in the Supplement S2). (e) Phengite groups in sample FG1249 (details in text). (f) Phengite groups in sample FG1420 (details in text). Note larger phengite flakes of Group2 localized in fold hinges.**

[Figure]

[Figure]

**Figure 4: Equilibrium phase diagrams of the studied samples computed with Theriak/Domino, assuming a free H$_2$O fluid, between 1.3-2 GPa and 400-700°C for (a), (b), (c), (d), (e) and between 0.3-1.5 GPa and 200-600°C for (f). The error bars departing from filled ellipses show the P-T uncertainty related to the analytical error of the garnet composition.**

[Figure]

**Figure 5: Multi-equilibrium thermobarometry results for the IC samples. (a), (c), and (e): intersection of the results deriving from** *Chlorite+Quartz+H₂O thermometry* **with** *White-Mica+Quartz+H₂O barometry.* **(b), (d), and (f): results of** *Chlorite+White-Mica+Quartz+H₂O thermobarometry.*

[Figure]

**Figure 6:** *Chlorite+White-Mica+Quartz+H₂O thermobarometry* **results for the EC sample.**

[Figure]

**Figure 7: Back-scattered electron photos illustrating some of the allanite crystals in the studied samples (a-h: thin sections; b-g: grain mounts). (a) Allanite grain elongate in the eclogite foliation with phengite and paragonite. It is rimmed by epidote and a simplectite of paragonite and chlorite.** (b) Allanite displaying several growth zones and a dark outermost rim of epidote. Note the phengite inclusion and laser ablation pits (32 µm). (c) Allanite displaying several growth zones, the innermost of which are intergrown with paragonite. The tiny dark inclusions are fine grained phengite and paragonite. (d) Monazite with lobate edges preserved at the core of an allanite grain. (e) Allanite intergrown with garnet and wrapped by a darker epidote rim. (f) Allanite including garnet, rutile and phengite. (g) Allanite preserved at the core of epidote. Inclusions of paragonite and phengite are present at the boundary allanite-epidote rim. (h) Magmatic allanite with epidote crystals as satellites; some of the latter display cores of metamorphic allanite.

**AGE DATA-Internal Complex**

**Tera-Wasserburg diagram  Sample  Th–isochron diagram**

[Figure]

**FG 13 24**

initial $^{207}Pb/^{206}Pb$:0.8398 ± 0.0097
Intercept Age:77.1693 ± 7.2294
MSWD:2.2428

isochron age: 77.7694 ± 11.9377
ini $^{208/206}Pb$: 2.0815 ± 0.0274
MSWD: 0.4580

**FG 13 15**

initial $^{207}Pb/^{206}Pb$:0.8229 ± 0.0181
Intercept Age:65.0300 ± 3.6123
MSWD:2.1307

isochron age: 65.8526 ± 4.4563
ini $^{208/206}Pb$: 2.0774 ± 0.0663
MSWD: 0.9825

**FG 12 157**

initial $^{207}Pb/^{206}Pb$:0.8290 ± 0.0088
Intercept Age:64.5110 ± 4.3763
MSWD:1.2168

isochron age: 64.5327 ± 6.3285
ini $^{208/206}Pb$: 2.0851 ± 0.0280
MSWD: 0.5501

**FG 13 47**

initial $^{207}Pb/^{206}Pb$: 0.7870 ± 0.0399
Intercept Age:55.6831 ± 4.4989
MSWD:2.5376

isochron age: 59.3781 ± 5.0689
ini $^{208/206}Pb$: 1.9796 ± 0.0817
MSWD: 0.39698

**FG 12 49**

initial $^{207}Pb/^{206}Pb$:0.8301 ± 0.0036
Intercept Age:72.5305 ± 1.4113
MSWD:1.7768

isochron age: 75.1703 ± 2.9426
ini $^{208/206}Pb$: 2.0771 ± 0.0137
MSWD: 0.5126

[revised manuscript text omitted]

11    **Table 1: Representative analyses of garnet of the samples**

| Sample Ph | IC-FG1324 | | | IC-FG1315 | | IC-FG12157 | | IC-FG1347 | | IC-FG1249 | | EC-FG1420 | | | |
|---|---|---|---|---|---|---|---|---|---|---|---|---|---|---|---|
| | Spot analysis (wt%) | | | Average composition (wt%) | σ | Average composition (wt%) | σ | Average composition (wt%) | σ | Average composition (wt%) | σ | Average comp. (wt%) High Si | σ | Average comp. (wt%) Low Si | σ |
| wt% | | | | | | | | | | | | | | | |
| SiO2 | 51.34 | 49.24 | 51.99 | 50.89 | 0.55 | 49.61 | 0.58 | 49.43 | 0.62 | 49.57 | 0.79 | 50.82 | 0.87 | 48.54 | 0.86 |
| TiO2 | 0.41 | 0.53 | 0.53 | 0.26 | 0.15 | 0.31 | 0.06 | 0.23 | 0.05 | 0.24 | 0.08 | 0.15 | 0.04 | 0.13 | 0.03 |
| Al2O3 | 26.67 | 26.82 | 26.83 | 28.15 | 0.40 | 27.58 | 0.35 | 29.78 | 0.50 | 27.92 | 0.52 | 27.45 | 0.69 | 30.17 | 0.76 |
| FeO | 1.67 | 1.39 | 1.64 | 1.49 | 0.11 | 2.09 | 0.43 | 1.40 | 0.14 | 1.30 | 0.42 | 3.21 | 0.25 | 2.63 | 0.28 |
| MnO | 0.00 | 0.00 | 0.00 | 0.06 | 0.01 | 0.19 | 0.04 | 0.06 | 0.01 | 0.05 | 0.01 | 0.08 | 0.01 | 0.08 | 0.01 |
| MgO | 3.00 | 3.37 | 3.45 | 3.68 | 0.14 | 3.65 | 0.15 | 3.29 | 0.16 | 3.45 | 0.18 | 2.85 | 0.25 | 2.30 | 0.21 |
| CaO | 0.00 | 0.00 | 0.00 | 0.00 | 0.00 | 0.19 | 0.04 | 0.01 | 0.00 | 0.01 | 0.00 | 0.00 | 0.00 | 0.00 | 0.01 |
| Na2O | 0.62 | 0.86 | 0.74 | 0.77 | 0.09 | 0.66 | 0.08 | 0.71 | 0.11 | 0.87 | 0.14 | 0.44 | 0.11 | 0.32 | 0.08 |
| K2O | 10.16 | 10.14 | 10.33 | 10.24 | 0.17 | 9.97 | 0.19 | 10.35 | 0.21 | 9.97 | 0.26 | 10.56 | 0.26 | 10.77 | 0.25 |
| Total | 93.87 | 92.35 | 95.51 | 95.54 | - | 94.26 | - | 95.25 | - | 93.37 | - | 95.56 | - | 94.93 | - |

**Formulae based on 11 oxygens**

| | | | | | | | | | | | | | | | |
|---|---|---|---|---|---|---|---|---|---|---|---|---|---|---|---|
| Si | 3.46 | 3.38 | 3.44 | 3.37 | 0.02 | 3.35 | 0.02 | 3.29 | 0.03 | 3.36 | 0.03 | 3.40 | 0.04 | 3.27 | 0.04 |
| Ti | 0.02 | 0.03 | 0.03 | 0.01 | 0.00 | 0.02 | 0.00 | 0.01 | 0.00 | 0.01 | 0.00 | 0.01 | 0.00 | 0.01 | 0.00 |
| Al | 2.12 | 2.17 | 2.09 | 2.20 | 0.01 | 2.19 | 0.03 | 2.34 | 0.03 | 2.23 | 0.04 | 2.16 | 0.05 | 2.39 | 0.05 |
| Fe | 0.09 | 0.08 | 0.09 | 0.08 | 0.00 | 0.12 | 0.02 | 0.08 | 0.01 | 0.07 | 0.02 | 0.18 | 0.01 | 0.15 | 0.02 |
| Mn | 0.00 | 0.00 | 0.00 | 0.00 | 0.00 | 0.01 | 0.00 | 0.00 | 0.00 | 0.00 | 0.00 | 0.00 | 0.00 | 0.01 | 0.00 |
| Mg | 0.30 | 0.35 | 0.34 | 0.36 | 0.01 | 0.37 | 0.02 | 0.33 | 0.02 | 0.35 | 0.02 | 0.28 | 0.03 | 0.23 | 0.02 |
| Ca | 0.00 | 0.00 | 0.00 | 0.00 | 0.00 | 0.01 | 0.00 | 0.00 | 0.00 | 0.00 | 0.00 | 0.00 | 0.00 | 0.00 | 0.00 |
| Na | 0.08 | 0.11 | 0.10 | 0.10 | 0.01 | 0.09 | 0.01 | 0.09 | 0.01 | 0.11 | 0.02 | 0.06 | 0.01 | 0.04 | 0.01 |
| K | 0.87 | 0.89 | 0.87 | 0.87 | 0.01 | 0.86 | 0.02 | 0.88 | 0.02 | 0.86 | 0.02 | 0.90 | 0.02 | 0.93 | 0.02 |
| ∑ cations | 6.94 | 7.01 | 6.97 | 7.00 | - | 7.01 | - | 7.02 | - | 7.00 | - | 6.99 | - | 7.02 | - |
| XMg | 0.76 | 0.81 | 0.79 | 0.82 | 0.012 | 0.76 | 0.04 | 0.81 | 0.02 | 0.83 | 0.05 | 0.61 | 0.03 | 0.61 | 0.03 |

12   **Table 2: Representative analyses of phengite of the samples. σ is the standard deviation associated to the average composition analysis**

[revised manuscript text omitted]

16   **Table 5: Constraints linking age data to pressure and temperature based on mineral inclusions in allanite; uncertainties**

17   **of age data are 2σ. Allanite age data based on Tera-Wasserburg intercepts. Zircon [206]Pb/[238]U dates are given as the**

18   **minimum and maximum range of individual analysis. Groups are discussed in section 7.**

| | | Age at ~0.6GPa | vertical velocity mm/year | along sub. inter. 60° mm/year | along sub. inter. 45° mm/year |
|---|---|---|---|---|---|
| Stage1 | Group 1 | ~38 Ma | 0.9 | 1.1 | 1.3 |
| | Group 1 | ~46 Ma | 1.2 | 1.4 | 1.7 |
| | Group 3 | ~38 Ma | 2.7 | 3.1 | 3.8 |
| | Group 3 | ~46 Ma | 5.1 | 5.9 | 7.2 |
| Stage2 | INT-EXT | ~38 Ma | 4 | - | - |
| | INT-EXT | ~46 Ma | 1.6 | - | - |

21 **Table 6: Exhumation velocities for Groups 1 and 3 (IC) from the highest pressures recorded to the greenschist**
22 **juxtaposition with the EC (Stage1) and for the late exhumation to the surface of the Sesia Zone (INT-EXT; Stage 2).**